# Mitigating the Impact of Increased Drought-Flood Abrupt Alternation Events under Climate Change: The Role of Reservoirs in the Lancang-Mekong River Basin

Keer Zhang[1], Zilong Zhao[1], Fuqiang Tian[1,2]

[1]Department of Hydraulic Engineering & State Key Laboratory of Hydroscience and Engineering, Tsinghua University, Beijing 100084, China

[2]Southwest United Graduate School, Kunming 650091, China

*Correspondence to*: Fuqiang Tian (tianfq@mail.tsinghua.edu.cn)

**Abstract.** The Lancang-Mekong River (LMR) Basin is highly vulnerable to extreme hydrological events, including Drought-Flood Abrupt Alternation (DFAA). The efficacy of potential mitigation measures, such as reservoir operations, on DFAA under climate change remains poorly understood. This study investigates these dynamics using five Global Climate Models (GCMs) from the Coupled Model Intercomparison Project Phase 6 (CMIP6). It employs the Revised Short-cycle Drought-Flood Abrupt Alteration Index (R-SDFAI), along with the Tsinghua Representative Elementary Watershed (THREW) model integrated with the developed reservoir module. The findings reveal that DFAA in the LMR Basin is primarily dominated by DTF (drought to flood), with probabilities of DTF exceeding those of FTD (flood to drought) at mild, moderate, and severe intensity levels. The increase in DTF probability for future periods is also significantly higher than that of FTD. Mild DTF and mild FTD account for 58% to 90% and 75% to 100% of their total probability in the future, making the mild-intensity events the most frequent DFAA. Reservoirs play a significant role in reducing DTF risks during both dry and wet seasons, though their effectiveness in controlling FTD risks, particularly during the dry season, is relatively weaker. Furthermore, there is a positive correlation between the reservoir's capacity to mitigate total DFAA risk and its total storage. Reservoirs display a stronger ability to regulate high-intensity FTD and high-frequency DTF events, and significantly reduce the monthly duration of DFAA. These insights provide valuable guidance for the effective management of water resources cooperatives across the LMR Basin.

**Keywords.** Drought-Flood Abrupt Alternation; Climate change; Reservoir operation; Lancang-Mekong River Basin.

**1. Introduction**

Flood and drought are two of the most frequent natural disasters in the world (Adikari and Yoshitani, 2009; ADREM et al., 2024). Drought-Flood Abrupt Alternation (DFAA), which is defined as the rapid transition between flood and drought conditions within a region (Xiong and Yang, 2025), has received growing attention in recent years (Chen et al., 2025; Wu et al., 2023; Zhang et al., 2012; Shan et al., 2018; Song et al., 2023). DFAA specifically consists of two types of rapid transition events: (1) drought to flood (DTF), where conditions shift quickly from drought to flood, and (2) flood to drought (FTD), where conditions rapidly change from flood to drought. Hazards arising from DFAA are more significant than those from floods and droughts. DFAA not only alters soil conditions and increases the potential for exceeding water quality standards (Bai et al., 2023; Yang et al., 2019) but also challenges food security and seriously affects agricultural production. Furthermore, DFAA events, particularly DTF events, are prone to triggering severe secondary natural hazards, primarily including flash floods, landslides, and mudslides (Wang et al., 2023).

It has been observed that the intensity and frequency of DFAA events demonstrate a global increasing trend (Yang et al., 2022; Chen et al., 2024). However, notable regional differences exist. Shan et al. (2018) observed that the scope of DFAA events in the Yangtze River mid-lower reaches has expanded since the 1960s, with both frequency and intensity increasing annually. Zhang et al. (2012) found that although droughts and floods have increased in the Huai River Basin, DFAA events have become less frequent. Looking ahead, Zhao et al. (2022) projected that the Han River Basin will experience an upward trend in both DFAA frequency and intensity, whereas Yang et al. (2019) reported a projected decline in the frequency of DFAA events in the Hetao region.

The Lancang-Mekong River (LMR), as a significant international river in Southeast Asia, profoundly affects key sectors such as hydropower, agriculture, fisheries, and transport (Morovati et al., 2024). At the same time, the LMR Basin is a high-incidence area for floods and droughts (Liu et al., 2020; MRC, 2020). Notably, wet-season droughts account for about 40% of annual droughts (Tian et al., 2020), while the region is also prone to large floods during the dry season (e.g., May 2006, May 2007, December 2016) (Tellman et al., 2021). The existence of these wet-season droughts and dry-season floods establishes the necessary conditions for DFAA in the LMR Basin.

Continued global warming is expected to further intensify both extreme wet and dry climate patterns

(IPCC, 2023), contributing to increased vulnerability to DFAA in the future (Yang et al., 2022; Wang et
al., 2023; Chen et al., 2025). There is a strong tendency toward more intense floods and droughts in
Southeast Asia (IPCC WG1, 2021) and specifically in the LMR Basin (Wang et al., 2021; Li et al., 2021;
Dong et al., 2022; Hoang et al., 2016). This heightens concerns about DFAA patterns in the LMR Basin,
emphasizing the need for improved water security, sustainable management, and early disaster
forecasting and prevention systems.
The hydrological regime of the LMR Basin is shaped mainly by climate change and human activities
(LMC and MRC, 2023). Despite the severe impacts of climate change, human activities such as reservoir
operation can help adapt the hydrological regime to these changes (Zhang et al., 2023; Khadka et al.,
2023; Sridhar et al., 2019; Lu et al., 2014; Gunawardana et al., 2021). Researches highlight that reservoirs
play a crucial role in reducing flood damage during the wet season and in minimizing low-flow
occurrences (Arias et al., 2014; Räsänen et al., 2012; Dang and Pokhrel, 2024). To evaluate reservoir
impacts under the changing climate, integration of a reservoir module within hydrological models is a
widely adopted practice. For example, Wang et al. (2017b) demonstrated that reservoir operation can
reduce flood intensity and frequency, while Yun et al. (2021a; 2021b) showed that careful reservoir
management can relieve both extreme drought and wet events, though with some trade-offs in
hydroelectric benefits. Collectively, these studies indicate that reservoirs offer practical adaptation
solutions to address climate change impacts.
It is essential to consider how human activities, especially reservoir operations, can help manage DFAA
under climate change. This consideration supports effective water resource management and the
sustainable development of the basin system. However, little research to date has focused on this aspect
for the LMR Basin. The statistics, reports, and studies on DFAA in the LMR Basin remain scarce,
particularly concerning the mitigating role of reservoirs under the changing climate. In response, this
study develops a reservoir module for hydrological modeling, examines the trends of DFAA in the LMR
Basin under climate change, and assesses how reservoirs can help basin states adapt to changing
conditions. This work aims to advance knowledge on DFAA and support regional water resources
management and sustainability.
**2. Methodology**

 **2.1 Study area**

The LMR originates from the Tibetan Plateau in China and flows through China, Myanmar, Laos, Thailand, Cambodia, and Vietnam before entering the South China Sea at the Mekong Delta. The LMR is approximately 4900 km long with a basin area of 812,400 km² (He, 1995). Its annual runoff is about 446 billion m³ (MRC, 2023). The LMR Basin is characterized by steep slopes and rapid flows in the upstream. The downstream features shallow slopes and slow, mixed flows. The wet and dry seasons in the LMR Basin extend from June to November and from December to May, respectively (LMC and MRC, 2023). These are mainly influenced by the southwestern and northeastern monsoons. The distribution of the hydrology system and mainstream hydrological stations in the LMR Basin is detailed in Fig. 1a.

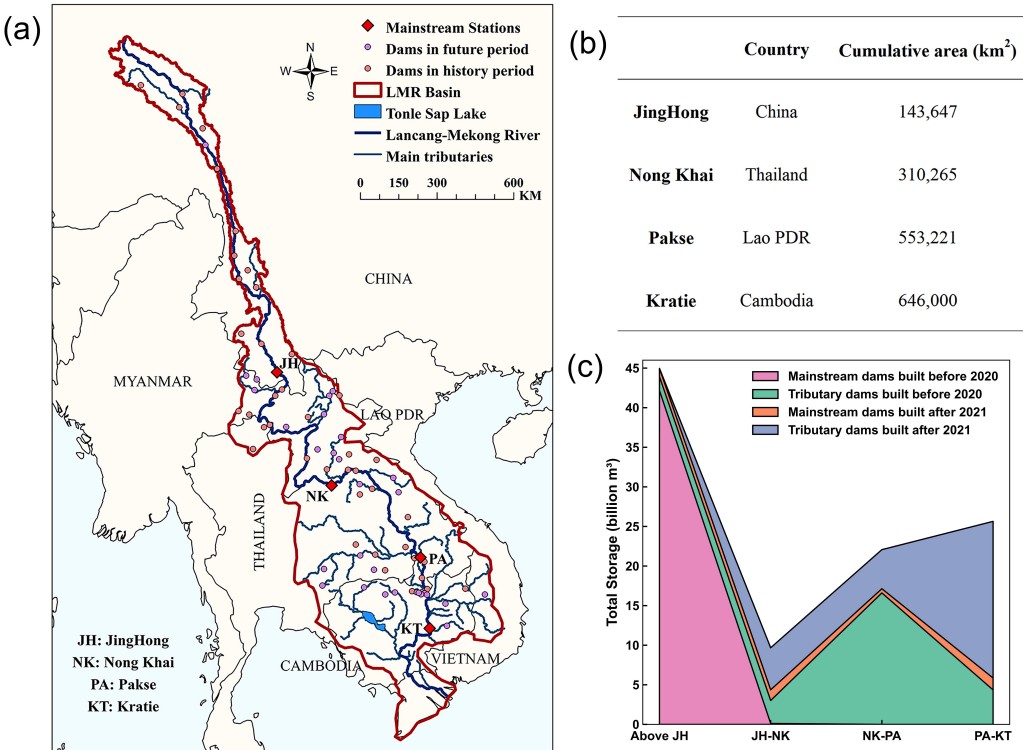

**Figure 1: Hydrology of the LMR Basin. (a) Map of rivers and reservoirs, (b) Information on four main hydrological stations, and (c) distribution of reservoir storage. Here, JH, NK, PA, and KT denote JingHong, Nong Khai, Pakse, and Kratie stations, respectively.**

The LMR Basin nourishes approximately 65 million people (Sabo et al., 2017; Luo et al., 2023). The basin states rely on the river system to develop economic industries, including capture fisheries, irrigation agriculture, and hydropower. The LMR Basin has the largest freshwater capture fishery in the world (MRC, 2010; MRC, 2019). Its irrigation area is estimated at around 4.3 million hectares (Do et al., 2020), with the Mekong Delta regarded as Southeast Asia's food basket. The LMR Basin is one of the most

active regions for hydropower in the world (MRC, 2019; Williams, 2019). It harbors about 235,000
GWhyr$^{-1}$ of hydroelectric potential in its mainstream and tributaries (Do et al., 2020; Schmitt et al., 2018).
The LMR Basin is also heavily impacted by floods and droughts. During the past two decades, the LMR
Basin has experienced several severe droughts (2004-2005, 2009-2010, 2015-2016, and 2019-2020) and
floods (Liu et al., 2020; Tian et al., 2020; MRC, 2020). These disasters affect crop cultivation and
fisheries harvesting, leading to the loss of property and lives in riparian countries. In 2013 and 2018,
floods heavily affected the lower basin, specifically Cambodia, Vietnam, Laos, and Thailand. These
floods covered 22.3 and 6.47 thousand km$^2$, respectively (Tellman et al., 2021).
**2.2 Data collection**
This study utilizes CMIP6 (Sixth Phase of Coupled Model Inter-comparison Project) data as the
meteorological input to analyze DFAA. Three SSP (Shared Socioeconomic Pathways) scenarios, namely
SSP1-2.6, SSP2-4.5, and SSP5-8.5, are considered to characterize the low-, medium-, and high-emission
scenarios, respectively. Five GCMs (Global Climate Models) with wide utilization and proven
performance in the LMR Basin are applied in this study (Li et al., 2021; Yun et al., 2021a; Yun et al.,
2021b), i.e., GFDL-ESM4, IPSL-CM6A-LR, MPI-ESM1-2-HR, MRI-ESM2-0, and UKESM1-0-LL.
The detailed information for these five GCMs is shown in Table 1 (Eyring et al., 2016; Gidden et al.,
2019; Cui et al., 2023). CMIP6 data span from 1980 to 2100. This study accordingly considers three
research periods: the history period from 1980 to 2014 (consistent with CMIP6), the near future period
from 2021 to 2060, and the far future period from 2061 to 2100.
In this study, the daily observed runoff data at four major mainstream hydrological stations from 1980 to
2020 are used to calibrate and validate the hydrological model. These data are derived from the China
Meteorological Administration (CMA) and the Mekong River Commission (MRC). The hydrological
stations from upstream to downstream are sequentially JingHong, Nong Khai, Pakse, and Kratie, whose
locations and basic information are shown in Figs. 1a and 1b. This study uses the ERA5_Land data as
the meteorological input for calibrating and validating the hydrological model, and as the correction
dataset for correcting the raw CMIP6 data. ERA5_Land data cover the period from 1980 to 2020, with a
spatial resolution of 0.1°, and contain precipitation, temperature, and potential evapotranspiration. Soil
data are obtained from the Global Soil Database (GSD) provided by the Food and Agriculture
Organization of the United Nations (FAO) with a spatial resolution of 10 km x 10 km. Normalized
Vegetation Index (NDVI), Leaf Area Index (LAI), and Snow Cover data are obtained from MODIS
(Moderate-resolution Imaging Spectroradiometer) with a spatial resolution of 500 m x 500 m and a
temporal resolution of 16 days.
Reservoir data are sourced from MRC and Mekong Region Futures Institute (MERFI) (MERFI, 2024).
This study utilizes 122 reservoirs, which simultaneously contain information on location, storage, and
operation years, including 24 reservoirs in the Lancang Basin and 98 reservoirs in the Mekong Basin.
The earliest and latest operation years for them are 1965 and 2035. The location and storage distribution
of these reservoirs are shown in Figs. 1a and 1c.
**Table 1: Details of 5 GCMs applied in this study.**

| Model Name | Modeling Center | Realization | Resolution (Lon×Lat) |
|---|---|---|---|
| GFDL-ESM4 | National Oceanic and Atmospheric Administration Geophysical Fluid Dynamics Laboratory, United States | r1i1p1f1 | 1.25°×1° |
| IPSL-CM6A-LR | Institute Pierre Simon Laplace, France | r1i1p1f1 | 2.5°×1.25874° |
| MPI-ESM1-2-HR | Max Planck Institute for Meteorology, Germany | r1i1p1f1 | 0.9375°×0.9375° |
| MRI-ESM2-0 | Meteorological Research Institute, Japan | r1i1p1f1 | 1. 125°×1. 125° |
| UKESM1-0-LL | Met Office Hadley Centre, UK | r1i1p1f2 | 1.875°×1.25° |

**2.3 Bias correction method for CMIP6 data**
The raw CMIP6 data require correction for more accurate modelling (Hoang et al., 2016; Mishra et al.,
2020; Sun et al., 2023). The uncorrected raw CMIP6 data misestimate the temperature and precipitation
in the LMR Basin, especially overestimating the precipitation (Cui et al., 2023; Lange, 2019; Lange,
2021). ERA5_Land data are used as correction data in this study to address bias in raw CMIP6 data.
This study interpolates the data from the five GCMs of CMIP6, which have different spatial resolutions,
to 0.1° (consistent with ERA5_Land) using the bilinear interpolation spatial resolution method. The
interpolated CMIP6 data are bias-corrected for each GCM according to an N-dimensional probability
density function transform of the multivariate bias correction approach (abbreviated as MBCn) (Cannon,
2016; Cannon, 2018). The MBCn method is trained based on the difference between precipitation and
temperature data from ERA5_Land and CMIP6 over the history period (1980-2014), and then applied to
the future period (i.e., 2021-2100) to correct the CMIP6 data for each GCM.
The MBCn method considers the multivariate dependency structure of meteorological data and enables
the simultaneous correction of temperature and precipitation data. Random orthogonal rotation and
quantile delta mapping are the two most critical formulas of the MBCn method (Cannon, 2018), as
illustrated in Eqs. (1) and (2).
$$\begin{cases} \widetilde{X}_T^{[l]} = X_T^{[l]} R^{[l]} \\ \widetilde{X}_S^{[l]} = X_S^{[l]} R^{[l]} \\ \widetilde{X}_P^{[l]} = X_P^{[l]} R^{[l]} \end{cases} \tag{1}$$

Eq. (1) displays the process of random orthogonal rotation. It outlines the process of transforming
historical observations $X_T^{[l]}$, historical climate model simulations $X_S^{[l]}$, and climate model projections
$X_P^{[l]}$ using a random orthogonal rotation matrix $R^{[l]}$ during the $l$-th iteration. The rotated data are
represented as $\widetilde{X}_T^{[l]}$, $\widetilde{X}_S^{[l]}$, and $\widetilde{X}_P^{[l]}$. This procedure is pivotal for MBCn's multivariate joint distribution
correction, as it transforms the original variable space into new random orientations. In contrast to
conventional univariate correction approaches, MBCn employs a random orthogonal matrix to mix
variables, thereby breaking their independence.
$$\begin{cases} \Delta^{(n)[l]}(i) = \tilde{x}_P^{(n)[l]}(i) - F_S^{(n)[l]^{-1}}(F_P^{(n)[l]}(\tilde{x}_P^{(n)[l]}(i))) \\ \hat{x}_P^{(n)[l]}(i) = F_T^{(n)[l]^{-1}}(F_P^{(n)[l]}(\tilde{x}_P^{(n)[l]}(i))) + \Delta^{(n)[l]}(i) \end{cases} \tag{2}$$

Eq. (2) exhibits the quantile delta mapping, which defines how quantile delta mapping is applied to the
$n$-th dimension of the rotated climate model projection data $\tilde{x}_P^{(n)[l]}(i)$ within the rotated space of the $l$-
th iteration. Here, $\Delta^{(n)[l]}(i)$ represents the quantile difference between the historical climate model
simulations and climate model projections in the $l$-th iteration and the $n$-th dimension. $F_P^{(n)[l]}$ denotes
the empirical cumulative distribution function for the rotated climate model projection data in the $n$-th
dimension. $F_T^{(n)[l]^{-1}}$ and $F_S^{(n)[l]^{-1}}$ denote inverse Functions of the empirical cumulative distribution
functions for the rotated historical observation data and historical climate model simulation data in the
$n$-th dimension. This step preserves the trend of the climate model projection data throughout the
correction process. The number of iterations is typically set to 10-30.
The MBCn algorithm performs multivariate joint distribution bias correction by iteratively applying
random orthogonal rotation and quantile delta mapping, while preserving the projected signals in the
climate model. The rotation operation breaks dependencies between variables, enabling the quantile delta
mapping of a single variable to indirectly adjust multivariate correlations. The quantile delta mapping
ensures the transmission of absolute or relative trends by computing quantile differences between the
historical and projected periods of the climate model. The MBCn method has been reported to increase
correction precision and accuracy compared to univariate and other multivariate bias correction
algorithms (Cannon, 2018).
In addition, this study utilized the method proposed by Van Pelt et al. (2009) to compute daily potential
evapotranspiration data for five GCMs under three SSP scenarios, based on daily temperature. The
computational approach is outlined in Eq. (3).
$PET = [1 + \alpha_0(T - \overline{T_0})]\overline{PET_0}$                 (3)
Where, $\overline{T_0}$ and $\overline{PET_0}$ correspond to the daily air temperature (°C) and daily potential
evapotranspiration (mm day$^{-1}$) in the history period sourced from ERA5_Land dataset. $T$ signifies the
corrected daily air temperature (°C) from CMIP6 dataset. The parameter $\alpha_0$ is determined by the
relationship between daily potential evapotranspiration and daily temperature in ERA5_Land data during
the history period.
**2.4 Hydrological model coupled with reservoir module**
The THREW (Tsinghua Representative Elementary Watershed) hydrological model is applied in this
study for runoff simulation. It utilizes the Representative Elementary Watershed (REW) approach for
spatial division, and further subdivides the REW into eight distinct hydrological zones: vegetated zone,
bare soil zone, glacier covered zone, snow covered zone, sub-stream-network zone, main channel reach,
saturated zone, and unsaturated zone (Tian et al., 2006; Mou et al., 2008).
The model is built upon scale-coordinated equilibrium equations, geometrical relationships, and
constitutive relationships, and enables comprehensive simulation of complex hydrological processes
from mountain to ocean. The fundamental balance equations in the THREW model are listed in Eqs. (4)
to (6).
$\frac{d}{dt}(\overline{\rho_\alpha^j}\epsilon_\alpha^j y^j \omega^j) = \sum_P e_\alpha^{jP} + \sum_{\beta \neq \alpha} e_{\alpha\beta}^j$           (4)
Eq. (4) demonstrates the general form of the mass conservation equation at the REW scale. $\frac{d}{dt}$ denotes
the time derivative. $\overline{\rho_\alpha^j}$ refers to the time-averaged density of phase $\alpha$ in sub-region $j$, in kg·m$^{-3}$. $\epsilon_\alpha^j$
means the volume fraction of phase $\alpha$ within sub-region $j$. $y^j$ indicates the time-averaged thickness of
sub-region $j$, in m. $\omega^j$ means the time-averaged fraction of REW horizontal area occupied by sub-region
$j$. $e_\alpha^{jP}$ denotes the net mass exchange flux of phase $\alpha$ in sub-region $j$ through interface $P$ (e.g., with
atmosphere, groundwater, neighboring REWs), in kg·m$^{-2}$·s$^{-1}$, where a positive value indicates the inflow
to sub-region $j$. $e_{\alpha\beta}^{j}$ refers to the phase transition rate between phase $\alpha$ and phase $\beta$ within sub-region
$j$, in kg·m$^{-2}$·s$^{-1}$, where a positive value indicates phase $\alpha$ gains mass from phase $\beta$. Sub-region here
refers to the eight zones within each REW.
$$\left(\overline{\rho_{\alpha}^{J}}\epsilon_{\alpha}^{j}y^{j}\omega^{j}\right)\frac{d\overline{v_{\alpha}^{J}}}{dt} = \overline{g_{\alpha}^{J}}\overline{\rho_{\alpha}^{J}}\epsilon_{\alpha}^{j}y^{j}\omega^{j} + \sum_{P}T_{\alpha}^{jP} + \sum_{\beta\neq\alpha}T_{\alpha\beta}^{j} \qquad (5)$$
Eq. (5) presents the general form of the momentum conservation equation at the REW scale. $\overline{v_{\alpha}^{J}}$
indicates the time-averaged velocity vector of phase $\alpha$ in sub-region $j$, in m·s$^{-1}$. $\overline{g_{\alpha}^{J}}$ denotes the time-
averaged gravity vector of phase $\alpha$ in sub-region j, in m·s$^{-2}$. $T_{\alpha}^{jP}$ means the force vector (pressure,
friction, seepage) exerted on phase $\alpha$ in sub-region j by interface $P$, in N·s$^{-2}$, representing the
momentum exchange. $T_{\alpha\beta}^{j}$ refers to the interfacial force vector between phase $\alpha$ and phase $\beta$ within
sub-region j, in N·s$^{-2}$, including drag and capillarity.
$$\left(\epsilon_{\alpha}^{j}y^{j}\omega^{j}c_{\alpha}^{j}\right)\frac{d\overline{\theta_{\alpha}^{J}}}{dt} = \overline{h_{\alpha}^{J}}\overline{\rho_{\alpha}^{J}}\epsilon_{\alpha}^{j}y^{j}\omega^{j} + \sum_{P}Q_{\alpha}^{jP} + \sum_{\beta\neq\alpha}Q_{\alpha\beta}^{j} \qquad (6)$$
Eq. (6) exhibits the general form of the heat conservation equation at the REW scale. $c_{\alpha}^{j}$ means the
specific heat capacity (constant volume) of phase $\alpha$ in sub-region j, in J·kg$^{-1}$·K$^{-1}$. $\theta_{\alpha}^{j}$ refers to the time-
averaged temperature of phase $\alpha$ in sub-region $j$, in K. $\overline{h_{\alpha}^{J}}$ denotes the heat generation rate per unit mass
within phase $\alpha$ in sub-region $j$, in W·kg$^{-1}$ (e.g., radioactive decay, negligible usually). $Q_{\alpha}^{jP}$ indicates
the heat exchange rate between phase $\alpha$ in sub-region $j$ and its environment via interface $P$, in W·m$^{-2}$,
with the positive value representing the heat gained by phase $\alpha$ in sub-basin $j$. $Q_{\alpha\beta}^{j}$ refers to the heat
exchange rate between phase $\alpha$ and phase $\beta$ within sub-region $j$, in W·m$^{-2}$, with a positive value
indicating that heat is gained by phase $\alpha$.
The THREW model employs an automatic calibration procedure to calibrate hydrological parameters
through parallel computation (Nan et al., 2021). The calibration period of the THREW model in the LMR
Basin is from 2000 to 2009, and the validation period is from 2010 to 2020. The calibration process
involves nine hydrological parameters. A compilation of their explanations and permissible value ranges
is given in Table 2. The Nash-Sutcliffe efficiency coefficient (NSE) indicator is adopted to calibrate the
objective function and evaluate simulation effectiveness at the daily scale, which is calculated according
to Eq. (7). The THREW model has been successfully applied to a number of basins with various climate
characteristics worldwide (Tian et al., 2012; Lu et al., 2021; Morovati et al., 2023; Cui et al., 2023; Zhang
et al., 2023).
$$NSE = 1 - \frac{\sum_{num=1}^{N}(Q_{obs}^{num} - Q_{sim}^{num})^2}{\sum_{num=1}^{N}(Q_{obs}^{num} - \overline{Q_{obs}})^2}$$     (7)
Where, $Q_{obs}^{num}$ is the daily observed runoff, $Q_{sim}^{num}$ is the daily simulated runoff, $\overline{Q_{obs}}$ is the average of
observed runoff, and $N$ is the total number of days.
**Table 2: Calibrated hydrological parameters and their ranges.**

| Parameter | Explanation | Range |
|---|---|---|
| kv | Fraction of potential transpiration rate over potential evaporation | 0-10 |
| nt | Roughness of slope | 0-2 |
| KKA | Exponential coefficient in subsurface runoff calculations | 0-100 |
| nr | Roughness of river channel | 0-1 |
| KKD | Linear coefficient in subsurface runoff calculation | 0-1 |
| B | Shape coefficient | 0-1 |
| WM | Average water storage capacity (m) | 0-5 |
| K | Storage factor in Muskingum Method | 0-1 |
| X | Flow ratio factor in Muskingum Method | 0-0.5 |

This study extends the THREW model by developing and integrating a reservoir management module.
This integration allows the expanded THREW model to use detailed information on 122 reservoirs in the
LMR Basin, with operational years ranging from 1965 to 2035. By specifying whether the module is
active, the model can simulate either natural runoff (without considering reservoirs) or dammed runoff
(with reservoirs included). This setup ensures a seamless interaction between the core model and the
reservoir operations framework.
Reservoir operation follows consistent rules across time and space, with each reservoir starting operation
according to its operational year. Strategies are adapted in response to inflow fluctuations and
administered on a daily scale. Each reservoir is assigned based on location. Cumulative multi-year sub-
basin storage is calculated as input for the reservoir module, which operates in two phases: initial and
normal. The normal phase is divided into general and emergency cases, both using the same operation
rules but differing constraints; the emergency case allows more flexibility. The module's flowchart is
illustrated in Fig. 2.
If a REW's cumulative multi-year storage changes within a year, it signals the start of a new reservoir's
operation, which follows initial phase rules. During the initial phase, the outlet flow matches the inlet if
it is below the minimum discharge constraint; otherwise, it meets the minimum discharge constraint. The
rules for the initial phase are described as Eqs. (8) to (9). Storage and discharge constraints are defined
in Eqs. (10) to (11) (Tennant, 1976; Yun et al., 2020). The initial phase ends when reservoir storage
exceeds the minimum constraint (Eq. (12)), then transitions to the normal phase.
$$Q_{out} = \begin{cases} Q_{in}, Q_{in} < Q_{min} \\ Q_{min}, Q_{in} \geq Q_{min} \end{cases} \tag{8}$$

$$S_t = S_{t-1} + Q_{in} - Q_{out} \tag{9}$$

$$S_{min} = 0.2 \times S_{total} \tag{10}$$

$$Q_{min} = 0.6 \times Q_{ave} \tag{11}$$

$$S_t \geq S_{min} \tag{12}$$

Where $Q_{out}$ is the outlet flow, $Q_{in}$ is the inlet flow, $Q_{min}$ is the minimum discharge constraint, $S_t$ is
the storage for time $t$, $S_{min}$ is the minimum storage constraint, $S_{total}$ is the total storage, and $Q_{ave}$ is
the average multi-year runoff during the calibration period (i.e., 2000-2009).

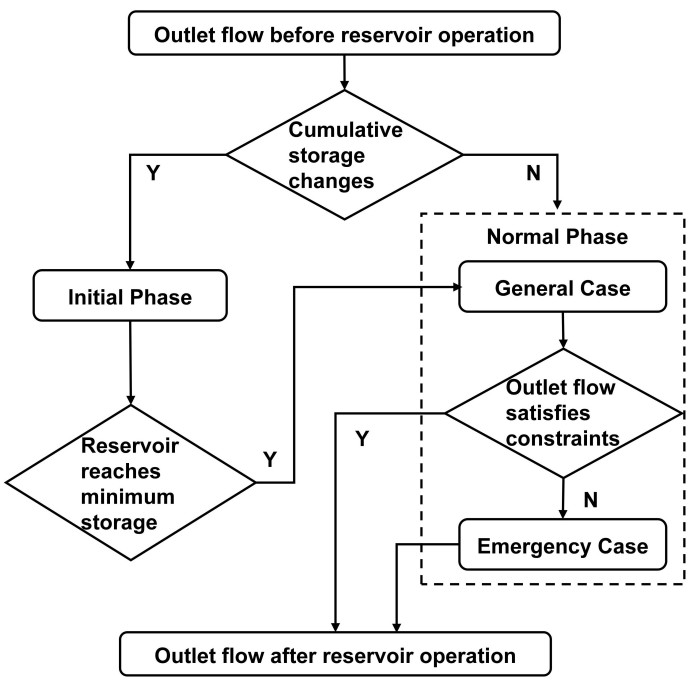


**Figure 2: Flowchart of the constructed reservoir module.**
The scheduling rule for the normal phase is the improved Standard Operation Policy hedging model
(SOP) (Wang et al., 2017a; Morris and Fan, 1998), as depicted in Eq. (9) and Eqs. (13) to (16). The SOP
operating policy is proven to effectively capture floods and droughts under reservoir regulation (Wang et
al., 2017a; Yun et al., 2020; 2021a; 2021b). Under the premise of water balance (Eq. (9)), constraints for
annual storage (Eq. (13)), outlet flow (Eq. (14)), wet season storage (Eq. (15)), and dry season storage
(Eq. (16)) are considered separately, where priority is given to the annual storage constraint (Eq. (13)).
$S_{min} \leq S_t \leq S_{max}$ (13)
$Q_{min} \leq Q_{out} \leq Q_{max}$ (14)
$min|S_c - S_t|, month = 6,7,8,9,10,11$ (15)
$min|S_n - S_t|, month = 12,1,2,3,4,5$ (16)
Where $Q_{max}$ is the maximum discharge constraint, $S_{max}$ is the maximum storage constraint, $S_c$ is the
storage corresponding to the flood control level, and $S_n$ is the storage corresponding to the normal water
level.
When in the normal phase, the reservoir first applies general case constraints (Eqs. (17) to (22)). If outlet
flow is not fully satisfied (Eq. (14)), constraints switch to the emergency case, and the reservoir is
rescheduled. Eq. (23) signals an emergency case start, which provides more flexible flow limits to avoid
extremes. Emergency case constraints are in Eqs. (24) to (25).
$Q_{max} = 2 \times Q_{ave}$ (17)
$Q_{min} = 0.6 \times Q_{ave}$ (18)
$S_c = S_{min} \times 1.2$ (19)
$S_n = S_{max} \times 0.8$ (20)
$S_{min} = 0.2 \times S_{total}$ (21)
$S_{max} = \begin{cases} 0.8 \times S_{total}, month = 6,7,8,9,10,11 \\ 1 \times S_{total}, month = 12,1,2,3,4,5 \end{cases}$ (22)
$Q_{min} \leq Q_{out}' \leq Q_{max}$ (23)
$Q_{min} = 0.3 \times Q_{ave}$ (24)
$S_{max} = 0.8 \times S_{total}$ (25)
Where $Q_{out}'$ is the outlet flow after the scheduling in the general case.
**2.5 Indicator for DFAA**
It is common practice to quantify DFAA incidents via indices. Long-cycle droughts-floods abrupt
alternation index (LDFAI), proposed by Wu et al. (2006), quantitatively characterizes long-term DFAA
during the wet season and has been widely adopted (Ren et al., 2023; Shi et al., 2021; Yang et al., 2022;
Yang et al., 2019). Building on this, Zhang et al. (2012) introduced the one-month interval SDFAI (short-
cycle droughts-floods abrupt alternation index), extending its application from precipitation to runoff and
characterizing short-term DFAA. SDFAI has since been applied in fields such as hydrology, meteorology,
ecology, and agriculture (Zhao et al., 2022; Lei et al., 2022; Yang et al., 2019; Zhang et al., 2019).
Song et al. (2023) proposed the Revised Short-cycle Drought-Flood Abrupt Alteration Index (R-SDFAI),
which extends the LDFAI and SDFAI time frame from only the flood season to the entire year, facilitating
multi-year DFAA analysis. R-SDFAI also addresses issues of over-identification, under-identification,
and misrepresentation of DFAA severity found in SDFAI. Therefore, this study uses R-SDFAI for DFAA
analysis, with the formulas outlined in Eqs. (26) to (31) (Song et al., 2023).
$$F_1 = S_{i+1} - S_i \tag{26}$$
$$F_2 = |S_{i+1}| + |S_i| \tag{27}$$
$$F = \left|\frac{F_1}{F_2}\right|^{|S_{i+1}+S_i|} \tag{28}$$
$$I = F \times min(|S_{i+1}|, |S_i|) \tag{29}$$
$$I' = \left(\frac{I}{0.5}\right)^{\frac{max(|S_{i+1}|,|S_i|)^2}{|F_1|+F_2}} \times \frac{I^{\frac{max(|S_{i+1}|,|S_i|)}{|F_1|+F_2}} + I^{\frac{min(|S_{i+1}|,|S_i|)}{|F_1|+F_2}}}{2} \tag{30}$$
$$R - SDFAI = sign(F_1) \times \left(\frac{I'}{I'_{0.5}} \times \frac{I}{0.5}\right)^{\left[\frac{max(|S_{i+1}|,|S_i|)}{|F_1|+F_2}\right]^{\left[1-\frac{max(|S_{i+1}|,|S_i|)}{|F_1|+F_2}\right]}} \tag{31}$$
Where, $S_i$ refers to the SRI in month $i$, $F1$ denotes the intensity of DFAA, $F2$ denotes the absolute
intensity of drought and flood, and $F$ is a weighting factor between 0 and 1. $I'_{0.5}$ refers to $I'$ when
$I$=0.5.
The calculation process of the SRI indicator utilized in this work is elucidated in Eqs. (32) to (37). The
runoff simulated by the THREW model for the LMR Basin conforms to a Gamma distribution, as detailed
in Appendix 1 of the Supplementary File. Hence, the Gamma distribution is adopted to derive the SRI
index. Eq. (32) gives the probability density function that satisfies the Gamma distribution for runoff $x$
at a given time period.
$$g(x) = \frac{1}{\beta^\alpha \Gamma(\alpha)} x^{\alpha-1} e^{-\frac{x}{\beta}}, x > 0 \tag{32}$$
Where, $\alpha > 0$ and $\beta > 0$ are respectively the shape and scale parameters. $\hat{\alpha}$ and $\hat{\beta}$ are the optimal
values of $\alpha$ and $\beta$, obtained according to the maximum likelihood estimation method, as illustrated in
Eqs. (33) to (35). $\Gamma(\alpha)$ is the gamma function, as given in Eq. (36).
$$\hat{\alpha} = \frac{1}{4A}(1 + \sqrt{1 + \frac{4A}{3}}) \tag{33}$$
$$\hat{\beta} = \frac{\bar{x}}{\hat{\alpha}} \tag{34}$$
$$A = ln(\bar{x}) - \frac{\sum ln(x_i)}{num} \tag{35}$$
$$\Gamma(\alpha) = \int_0^\infty y^{\alpha-1}e^y \, dy \tag{36}$$
Where, $x_i$ is the sample of runoff sequence, $\bar{x}$ is the average runoff, and $num$ is the length of the
runoff sequence.
Then the cumulative probability of runoff x is illustrated in Eq. (37).
$$G(x) = \int_0^x g(x) \, dx = \frac{1}{\hat{\beta}^{\hat{\alpha}}\Gamma(\hat{\alpha})} \int_0^x x^{\hat{\alpha}-1}e^{-\frac{x}{\hat{\beta}}} \, dx, x > 0 \tag{37}$$
**Table 3: The evaluation criteria and intensity classification for DFAA events.**

| Event | Intensity | Classification |
|---|---|---|
| | Mild | $1 \le$ R-SDFAI $< 1.44$ |
| DTF | Moderate | $1.44 \le$ R-SDFAI $< 1.88$ |
| | Severe | R-SDFAI $\ge 1.88$ |
| | Mild | $-1.44 <$ R-SDFAI $\le -1$ |
| FTD | Moderate | $-1.88 <$ R-SDFAI $\le -1.44$ |
| | Severe | R-SDFAI $\le -1.88$ |

The R-SDFAI index identifies DFAA events with a threshold of $\pm 1$ (Song et al., 2023), and further
categorizes DFAA events into three intensity levels—mild, moderate, and severe—using thresholds of
$\pm 1$, $\pm 1.44$, and $\pm 1.88$, as demonstrated in Table 3. This classification follows the criteria proposed by
Song et al. (2023). The underlying rationale involves using $\pm 0.5$, $\pm 1$, and $\pm 1.5$ as thresholds for the
SRI index to categorize extreme hydrological events into mild, moderate, and severe droughts and floods
(positive values indicate flood, while negative values indicate drought). The R-SDFAI index values of
$\pm 1$, $\pm 1.44$, and $\pm 1.88$ are calculated through the transitions between mild drought and mild flood,
moderate drought and moderate flood, and severe drought and severe flood. These thresholds serve as
the classification criteria for mild, moderate, and severe DFAA events. For a more detailed explanation
of this classification standard, please refer to Song et al. (2023). In this study, the frequency of DFAA
events is represented by their occurrence probabilities during history, near future, and far future periods,
while the intensity of DFAA is assessed through the probability of different intensity events.
**2.6 Scenario Setting**
This study examines two scenarios: dammed (with reservoir operations) and natural (without reservoir
operations). Meteorological data from five GCMs under three SSPs are downscaled to the REW scale
and used as input for the THREW model. The model, with the reservoir module, simulates runoff at key
hydrological stations for the history period (1980-2014), the near future (2021-2060), and the far future
(2061-2100). Both scenarios—with and without reservoir management—are examined. The R-SDFAI
indicator evaluates DFAA event probabilities for each period and for each scenario, using runoff
simulated by 5 GCMs and 3 SSPs.
This study adopts the difference in DFAA's probability between the natural scenario (without reservoir
operations) and the dammed scenario (with reservoir operations) to capture the reservoir's impact, as
shown in Eq. (38).
$P_{Impact\ of\ Reservoirs,i,e} = P_{Dammed,i,e} - P_{Natural,i,e}$         (38)
Where $P_{Impact\ of\ Reservoirs,i,e}$ represents the impact of reservoirs on the probability of event $e$ in period
$i$. $P_{Natural,i,e}$ denotes the probability of event $e$ under the natural scenario in period $i$, while $P_{Dammed,i,e}$
denotes the probability of event $e$ under the dammed scenario in period $i$. Period $i$ refers to near future or
far future. Event $e$ indicates DTF, FTD, or DFAA.
Eqs. (39) and (40) give the definitions of $P_{Natural,i,e}$ and $P_{Dammed,i,e}$ described above.
$P_{Natural,i,e} = \frac{M_{Natura,i,e}}{TM_i}$         (39)
$P_{Dammed,i,e} = \frac{M_{Dammed,i,e}}{TM_i}$         (40)
Where $M_{Natura,i,e}$ denotes the number of months in which event $e$ occurs in period $i$ under the natural
scenario. $M_{Dammed,i,e}$ denotes the number of months occurred event $e$ in period $i$ under the dammed
scenario. $TM_i$ refers to the total number of months in period $i$. Period $i$ refers to near future or far future.
Event $e$ indicates the DTF, FTD, or DFAA.
As each GCM possesses a unique structure and assumptions, projections of climate change by a single
GCM inherently possess uncertainties, which in turn introduce uncertainties in the simulation of
hydrological outcomes (Kingston et al., 2011; Thompson et al., 2014). Thus, averaging across multiple
GCMs is a crucial approach, as it minimizes model biases, eliminates outliers, reduces uncertainties, and
ensures more robust and universally applicable outcomes (Lauri et al., 2012; Hoang et al., 2016; Hecht
et al., 2019; Wang et al., 2024; Yun et al., 2021b). This method has been extensively employed in prior
studies (Dong et al., 2022; Li et al., 2021; Wang et al., 2022; Yun et al., 2021a). Therefore, this research
determines the average DFAA probability from five GCMs to lessen the uncertainty in their predictions
and assesses the fluctuation in these probabilities across the models to demonstrate their variability.
**3. Results**
**3.1 CMIP6 data bias correction performance**
From both regional and seasonal perspectives, the uncorrected raw CMIP6 data show significant
discrepancies with ERA5_Land data during the history period (1980-2014). When compared with
ERA5_Land data, the uncorrected raw CMIP6 data reveal an average annual precipitation bias of around
$\pm1800$ mm and an average daily temperature bias of approximately $\pm12$ ℃ (Figs. 3b and 3e). These
notable inconsistencies highlight that using uncorrected CMIP6 data for hydrological modeling would
incur considerable inaccuracies. However, CMIP6 data corrected by the MBCn method deviate from
ERA5_Land data by less than $\pm120$ mm of average annual precipitation and $\pm0.2$ ℃ of average daily
temperature (Figs. 3c and 3f). The bias correction greatly improves CMIP6 data accuracy in the LMR
Basin. The corrected CMIP6 data also match the seasonal cycle of ERA5_Land well for both
precipitation and temperature (Fig. 3g). Compared to the raw data, the corrected CMIP6 shows much
improved spatial and temporal accuracy, leading to more accurate and reasonable analyses for DFAA.
**3.2 Calibration and validation for the hydrological model**
The daily observed runoff and daily simulated runoff from the THREW model for the calibration period
(2000-2009) and validation period (2010-2020) are illustrated in Fig. 4, demonstrating the model's strong
performance. Importantly, since there was no massive reservoir construction in the LMR Basin before
and during the calibration period (Zhang et al., 2023), the THREW model without the reservoir module
is applied for calibration. Meanwhile, the addition of large-scale reservoirs during the validation period

allows validation of the THREW model configuration with the reservoir module, Notably, the THREW model captures runoff fluctuations between wet and dry seasons with high accuracy, achieving an NSE of at least 0.8 during both periods. This excellent simulation performance extends across both upstream and downstream regions, emphasizing the robustness of the model under observed conditions.

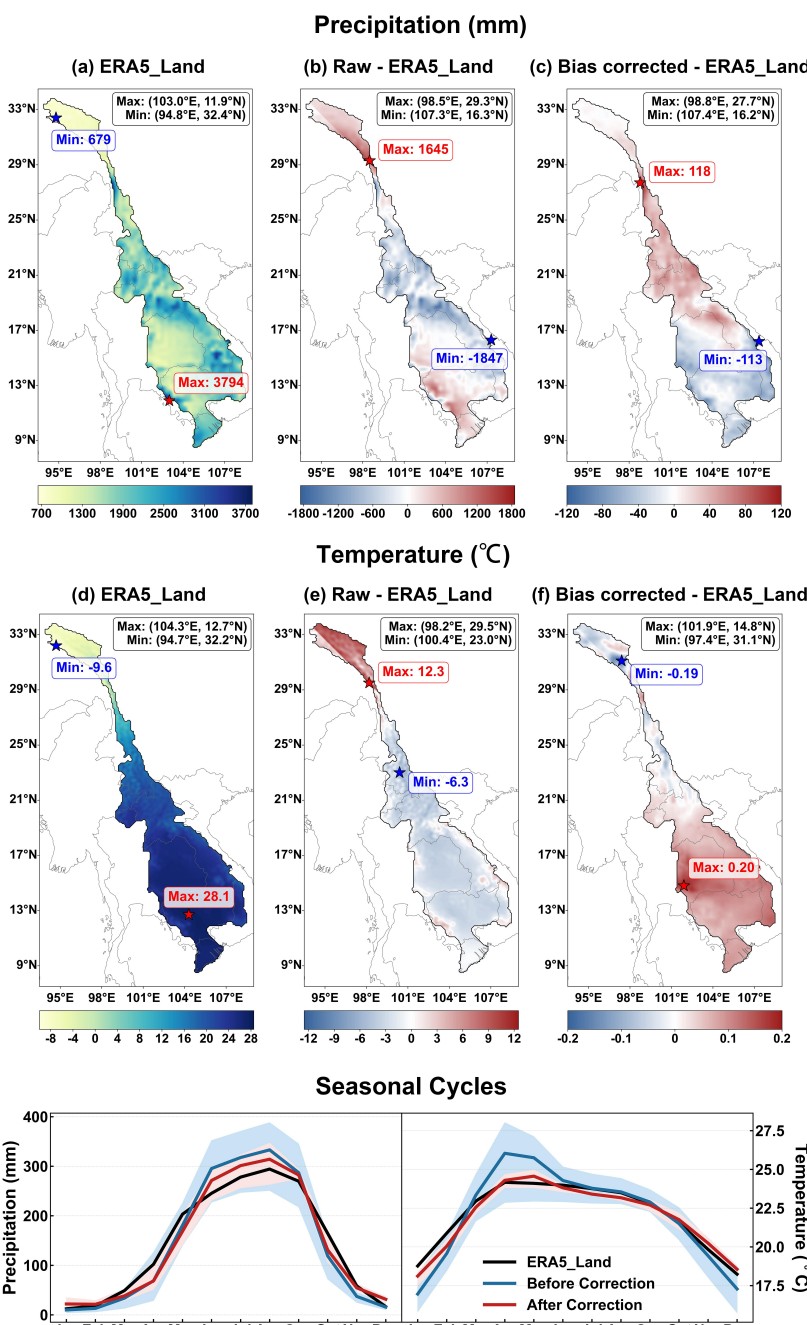

**Figure 3: Averaged meteorological data of 5 GCMs for the history period (1980-2014). Here, 5 GCMs are corrected separately. The red and blue star symbols respectively indicate the locations of the maximum and minimum values in (a) to (f). (a) to (c) present the spatial distribution of precipitation based on respectively ERA5_Land, raw CMIP6 (raw CMIP6 minus ERA5_Land) and bias-corrected CMIP6 (bias-corrected CMIP6 minus ERA5_Land). (d) to (f) illustrate the spatial distribution of temperature based on ERA5_Land,**

raw CMIP6 (raw CMIP6 minus ERA5_Land) and bias-corrected CMIP6 (bias-corrected CMIP6 minus
ERA5_Land). (g) shows seasonal cycles of temperature and precipitation from ERA5_Land, raw and bias-
corrected CMIP6, as well as their corresponding range.

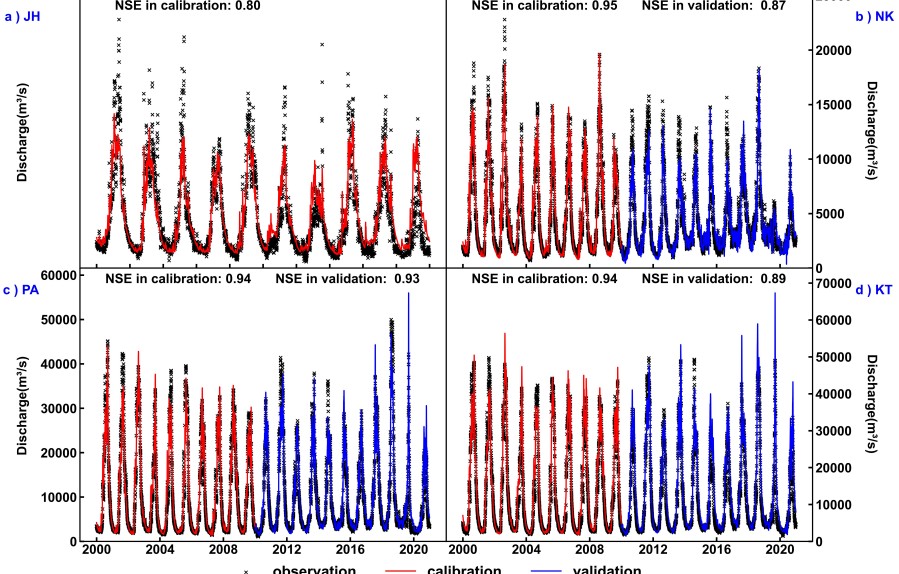

**Figure 4: Performance of the THREW model in calibration (2000-2009) and validation (2010-2020) periods.**
**Here, JH, NK, PA, and KT denote JingHong, Nong Khai, Pakse, and Kratie stations, respectively.**
**3.3 DFAA under the changing climate**
Under the natural scenario (without reservoir operations), DFAA in the LMR Basin is dominated by DTF,
that is, the risk of DTF is more critical than that of FTD (Table 4). The probability of FTD ranges from
0.7% to 2.1% in the history period, 0.6% to 2.0% in the near future, and 0.5% to 2.0% in the far future.
Conversely, DTF probabilities are higher, ranging from 1.6% to 2.3%, 1.2% to 3.2%, and 1.2% to 3.0%
respectively in these three periods.
**Table 4: The year-round DFAA probability averaged across five GCMs during each period under the natural**
**scenario.**

| Natural | Station | History | Near Future | | | Far Future | | |
|---|---|---|---|---|---|---|---|---|
| | | | SSP1-2.6 | SSP2-4.5 | SSP5-8.5 | SSP1-2.6 | SSP2-4.5 | SSP5-8.5 |
| DTF | JingHong | 1.67% | 2.04% | 1.71% | 1.63% | 1.67% | 1.75% | 1.21% |
| | Nong Khai | 1.52% | 1.71% | 2.08% | 1.17% | 1.96% | 2.25% | 1.71% |
| | Pakse | 2.24% | 2.38% | 3.13% | 1.83% | 2.67% | 2.75% | 2.04% |
| | Kratie | 2.33% | 3.17% | 2.83% | 2.08% | 3.04% | 2.92% | 2.54% |
| FTD | JingHong | 0.72% | 0.83% | 1.17% | 0.63% | 0.79% | 1.25% | 0.54% |
| | Nong Khai | 1.10% | 1.25% | 1.42% | 0.71% | 1.13% | 1.12% | 0.67% |
| | Pakse | 2.10% | 1.33% | 2.04% | 1.54% | 1.58% | 1.71% | 1.17% |
| | Kratie | 1.86% | 1.71% | 1.92% | 1.33% | 2.04% | 1.87% | 1.75% |

DFAA risk is substantially elevated during the wet season compared to the dry season (Table S1). For the
average of five GCMs, the probability of FTD in the wet season is 2 to 5.5 times higher than that in the
dry season in the history period. In the near and far future periods, this ratio ranges from 1.1 to 36 times
and 3.3 to 41 times, respectively. As for DTF, the probability in the wet season is correspondingly 1.7 to
5.7 times, 1.3 to 3.9 times, and 0.9 to 6.3 times higher than that in the dry season for history, near future,
and far future. Only JingHong station experiences a slightly higher probability of DTF in the dry season
(1.25%) than in the wet season (1.17%) for the far future.

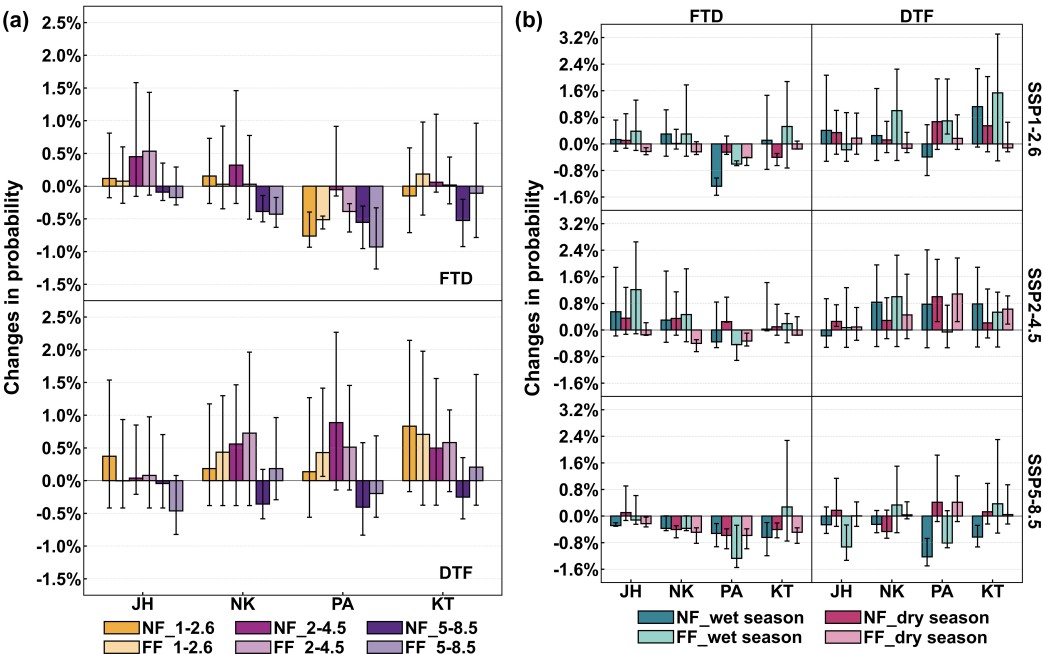


**Figure 5: DFAA under the natural scenario. (a) The annual change in DFAA probability averaged across five**
**GCMs and their ranges in the near and far future periods with respect to the history period under three SSPs.**
**(b) The seasonal change in DFAA probability averaged across five GCMs and their ranges in the near and**
**far future periods with respect to the history period during wet and dry seasons under three SSPs. Here, JH,**
**NK, PA, and KT respectively denote JingHong, Nong Khai, Pakse, and Kratie stations. NF and FF represent**
**the near future period and the far future period. 1-2.6, 2-4.5 and 5-8.5 respectively denote SSP1-2.6, SSP2-4.5,**
**and SSP 5-8.5 scenarios. Please note that this figure illustrates variations in DFAA events under climate**
**change. The annual and seasonal probabilities of DFAA under the natural scenario are presented in Table 4**
**and Table S1, respectively.**
DFAA risks show marked spatial variation, with annual probability consistently higher downstream than
upstream (Table 4). The annual probability of FTD ranges from 0.6% to 1.3% at JingHong station and
0.7% to 1.4% at Nong Khai station. These probabilities rise to 1.2% to 2.1% and 1.4% to 2.1% at Pakse
and Kratie stations, respectively. Similarly, the annual probability of DTF at JingHong and Nong Khai
stations is 1.2% to 2.1% and 1.2% to 2.3%. The probabilities at Pakse and Kratie stations range from 1.4%
to 3.2% and 3.1% to 3.2%, respectively. The DTF risk in the wet season and the FTD risk in both dry
and wet seasons are also higher downstream than upstream. Since the probability of FTD in the dry
season at Nong Khai, Pakse, and Kratie stations is limited, especially under the SSP5-8.5 scenario
(<0.2%), the risk of FTD in the dry season appears more notable upstream than downstream.
The annual DFAA probability increases under SSP1-2.6 and SSP2-4.5 scenarios (except for FTD at Pakse
station) and decreases under the SSP5-8.5 scenario (Fig. 5a). Such a pattern is attributable to the enhanced
tendency for flood and drought events in the LMR Basin to cluster rather than alternate under the SSP5-
8.5 scenario (Dong et al., 2022). Under the SSP5-8.5 scenario, the average probability of FTD across
five GCMs is 0.6% to 1.8%, while the probability of DTF ranges from 1.2% to 2.6%. Conversely, the
average probabilities of FTD and DTF under the SSP2-4.5 scenario range from 0.7% to 2.1% and 1.7%
to 3.2%, respectively.
The future growth in DTF is significantly greater than that in FTD. For the average probabilities across
five GCMs, relative to the history period, the future change in DTF probability at JingHong station is -
0.5% to 0.4%, at Nong Khai station is -0.4% to 0.7%, and at Pakse and Kratie stations, respectively, is -
0.5% to 0.9% and -0.2% to 0.8%. The future FTD probability change for JingHong is -0.2% to 0.5%,
while for Nong Khai, Pakse, and Kratie, the changes are -0.4% to 0.3%, -1% to -0.1%, and -0.6% to
0.2%, respectively. The maximum values from the five GCMs show a consistent trend, with increases in
DTF probability being significantly greater than those in FTD probability.
Upstream and downstream regions experience contrasting future risk increases, with FTD risks rising
more upstream and DTF risks rising more downstream (Fig. 5a). Under three climate scenarios, JingHong
Station experiences the maximum increase of 0.37% and 0.08% in DTF risks, respectively, in the near
and far future. Meanwhile, FTD risks at this station rise by 0.45% and 0.53%, respectively. Conversely,
Kratie Station exhibits the highest increase of 0.83% and 0.71% in DTF risks, alongside 0.06% and 0.02%
increases in FTD risks. The opposite trends of DFAA risk in upstream and downstream pose enhanced
challenges to the integrated management of the LMR Basin.
Future seasonal DFAA risks follow scenario-dependent trends: wet-season risks for both DTF and FTD
rise under SSP1-2.6 and SSP2-4.5 scenarios, and fall under the SSP5-8.5 scenario (Fig. 5b). This is
similar to the annual DFAA risk. The risk of FTD during the dry season decreases, with an upward trend
emerging only in the near future under the SSP2-4.5 scenario (average across five GCMs <0.4%,
maximum <1.3%). The risk of DTF during the dry season rises in most situations, except at Nong Khai

station in the near future under the SSP5-8.5 scenario, where it shows an average decrease of 0.46%

across five GCMs. The largest increase of dry-season risk of DTF is found at Pakse station under the

SSP2-4.5 scenario, with an average increase of 1.08% across five GCMs and a maximum increase of

2.08%.

Mild-intensity DFAA events constitute the majority of all DFAA occurrences (Fig. 6). The probability of

mild DTF varies across scenarios, with values ranging from 0.7% to 2.4%, which corresponds to 58% to

90% of the total DTF probability. Likewise, mild FTD probabilities range from 0.6% to 1.8% (Fig. 6),

comprising a larger share of the total FTD probability, specifically 75% to 100%. Mild DTF events

account for 2 to 13 times the possibility of moderate DTF events. This ratio escalates to 3 to 31 times for

FTD events. Notably, severe FTD events are extremely rare, often occurring at 0% probability. However,

severe DTF events are notable, with probabilities ranging from 0% to 0.38%, and in some instances,

accounting for up to 13% of total DTF probability.

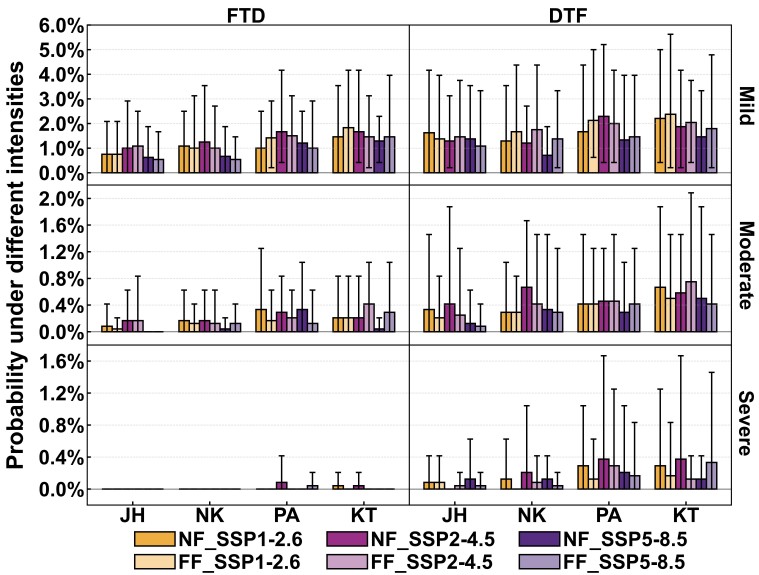

**Figure 6: Annual probability of DFAA at different intensities under the natural scenario, averaged across five GCMs and their ranges in the near future (2021-2060) and far future (2061-2100) periods under three SSPs. Here, JH, NK, PA, and KT respectively denote JingHong, Nong Khai, Pakse, and Kratie stations. NF and FF represent the near future period and the far future period. The specific value shown in this figure can be found in Table S2.**

The total probabilities of DTF events exceed that of FTD events (Fig. 5a), and this holds true for mild,

moderate, and severe intensity events (Fig. 6). The disparity between DTF and FTD events is not as

pronounced in mild intensity events, but it becomes significant in moderate intensity events. The

probabilities of moderate DTF range from 0.08% to 0.75%, whereas the probabilities of moderate FTD

range from 0.04% to 0.42% (Fig. 6). The marked disparity in severe intensity events is even more
pronounced by the extremely low probability of severe FTD.
Mild DTF probabilities are projected to increase in the far future, while moderate and severe DTF
probabilities are projected to decrease. Specifically, the probability of mild DTF rises to 1.1% to 2.4% in
the far future, compared to 0.7% to 2.3% in the near future. The probabilities of moderate and severe
DTF drop from an average of 0.42% and 0.19% in the near future to 0.38% and 0.12%, respectively, in
the far future. However, the probabilities of FTD events across all three intensity levels remain relatively
consistent between the near and far future.
**3.4 Reservoirs' impacts on DFAA**
Reservoirs exhibit extraordinary mitigation effects on DTF risk under the changing climate while
showing weaker effects in FTD risk (Fig. 7a). Nonetheless, the higher probability of DTF compared to
FTD (Fig. 5a) demonstrates that reservoirs contribute significantly to reducing overall DFAA risk. The
distinct controlling role of reservoirs on DTF risk versus FTD risk is associated with the consistency
between these two types of DFAA events and the logic of reservoir operation. Section 4.1 will delve into
the mechanistic details.
Reservoirs adequately reduce or only slightly increase the future DTF probability (-0.13% to 1%,
averaged across five GCMs. Throughout this section, a negative value indicates that reservoirs increase
the probability of DFAA, while positive values indicate a reduction. In most scenarios, the reservoir plays
a positive mitigating role across all GCMs (Fig. 7a). Reservoirs are expected to have better mitigation
effects in the near future at JingHong station. As for Nong Khai and Pakse stations, the reduction effect
of reservoirs on DTF is more pronounced in the far future under SSP1-2.6 and SSP2-4.5 scenarios, while
in the near future under the SSP5-8.5 scenario. The effect conversely, exhibits greater strength under
SSP1-2.6 and SSP5-8.5 scenarios in the near future, while it is stronger under the SSP2-4.5 scenario in
the far future at Kratie station. These findings are consistent across both the average of the GCMs and
their ranges.
Reservoirs are more effective in reducing FTD in the near future than in the far future at JingHong, Pakse,
and Kratie, while the effect at Nong Khai is slightly less in the far future (Fig. 7b). Reservoirs are most
effective under high emissions (SSP5-8.5), reducing FTD probability at all stations (0.13% to 0.42%,
GCM average). Under lower emissions (SSP1-2.6 and SSP2-4.5), mitigation is weaker (-0.33% to 0.38%,

 GCM average) at Nong Khai and Pakse, but notable at JingHong and Kratie, especially in certain future

periods. For example, under intermediate emissions (SSP2-4.5) in the far future at JingHong, reservoirs
lower the average probability by over 0.9% and maximum by nearly 1.8%.

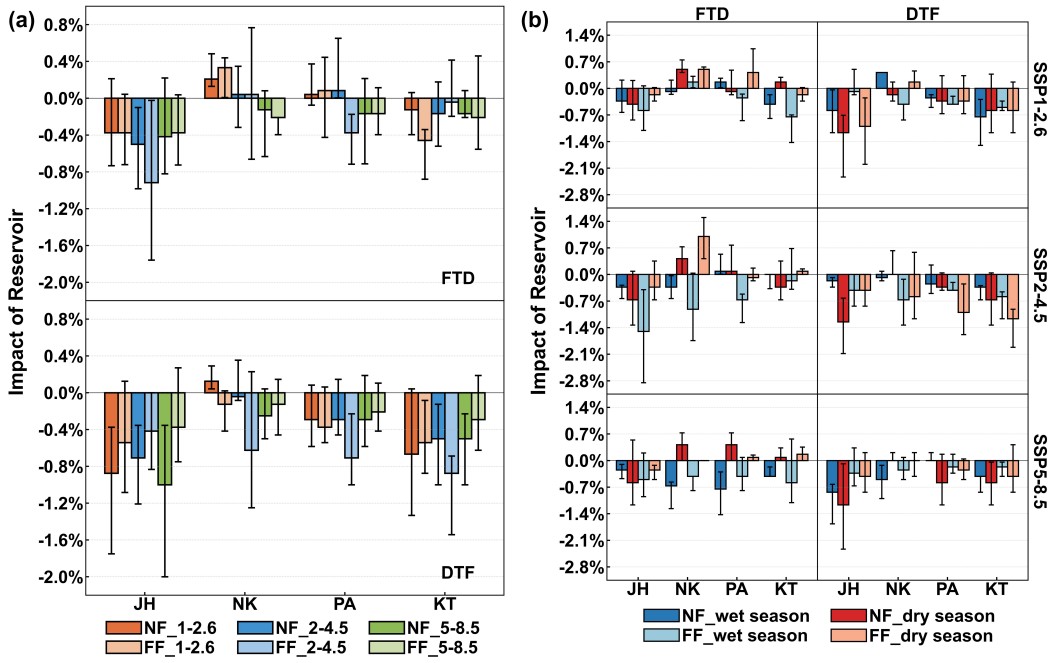


**Figure 7: Reservoir impacts on DFAA during the near future (2021-2060) and the far future (2061-2100)**
**under three SSPs. (a) The annual reservoir impacts averaged across five GCMs and their ranges. (b) The**
**seasonal reservoir impacts in wet and dry seasons averaged across five GCMs and their ranges. Here, JH, NK,**
**PA, and KT respectively denote JingHong, Nong Khai, Pakse, and Kratie stations. NF and FF represent the**
**near future period and the far future period. 1-2.6, 2-4.5 and 5-8.5 respectively denote SSP1-2.6, SSP2-4.5,**
**and SSP 5-8.5 scenarios. Please note that this figure illustrates the impact of reservoir operations on DFAA**
**events. The annual and seasonal probabilities of DFAA under the dammed scenario are presented in Table**
**S3.**

Reservoirs reduce FTD more in the wet season (-0.17% to 1.5%, GCM average) than in the dry season

(-1% to 0.67%), especially at Nong Khai, Pakse, and Kratie (Fig. 7b). Negative values mean a reservoir

increases FTD probability. In the wet season, reduction is notable (-0.17% to 0.92%), but in the dry

season, FTD probability increases (-1% to 0.33%). Seasonal differences in DTF mitigation are less

pronounced. Reservoirs slightly better reduce DTF in the dry season (-0.17% to 1.25%) than in the wet

season (-0.42% to 0.83%). Reservoirs mitigate DTF more effectively than FTD in both seasons, aligning

with the annual DFAA.

Reservoirs effectively manage DFAA events, which are predominantly characterized by mild intensity.

They decrease the probability of mild DTF by -0.1% to 0.9% (Fig. 8), whereas the probability of such

events is 0.7% to 2.4% under the natural scenario (Fig. 6), indicating that reservoirs decrease their
likelihood by -0.12 to 0.64 times. Reservoir reduces the probability of mild FTD by -0.4% to 0.8% (Fig.
8). They increase the probability of mild FTD at the Nong Khai station under the SSP1-2.6 scenario.
Since the probability of mild FTD is 0.6% to 1.8% under the natural scenario (Fig. 6), reservoir operation
reduces their probability by -0.38 to 0.69 times.

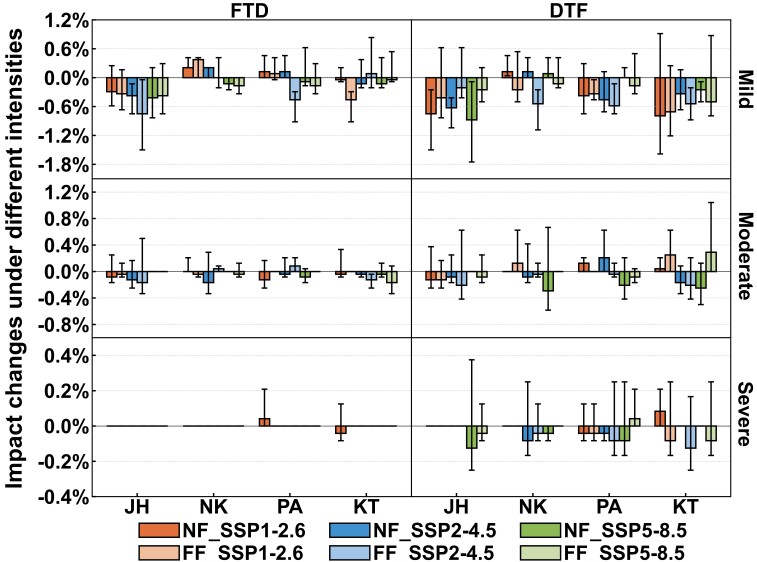


**Figure 8: Reservoir impacts on DFAA under different intensities, averaged across five GCMs and their ranges in the near future (2021-2060) and far future (2061-2100) periods under three SSPs. Here, JH, NK, PA, and KT respectively denote JingHong, Nong Khai, Pakse, and Kratie stations. NF and FF represent the near future period and the far future period. Please note that this figure shows how the reservoir affects DFAA events at different intensities. The probabilities of DFAA events at each intensity under the dammed scenario are presented in Table S4.**

While the reservoir's mitigation effect on FTD events is less pronounced than on DTF events (Fig. 7), it
demonstrates a commendable mitigation effect on moderate FTD, reducing their probability by -0.08%
to 0.17% (Fig. 8). This reduction represents -0.4 to 1 times the probability under the natural scenario.
This ratio surpasses the reservoir's mitigation effect on moderate DTF, where the probability is reduced
by -0.3% to 0.3% (Fig. 8), accounting for -0.70 to 1 times the natural probability. This highlights that the
reservoir exerts a more significant mitigating force on high-intensity FTD events compared to high-
frequency FTD events.
Reservoirs exhibit notable mitigating effects for DTF events across all three intensity levels. However,
their ability to alleviate moderate DTF is relatively weaker than that for mild DTF (Fig. 8), which differs
from the characteristic of FTD events. This implies that reservoirs possess a stronger capability to manage
high-frequency DTF events than higher-intensity events.

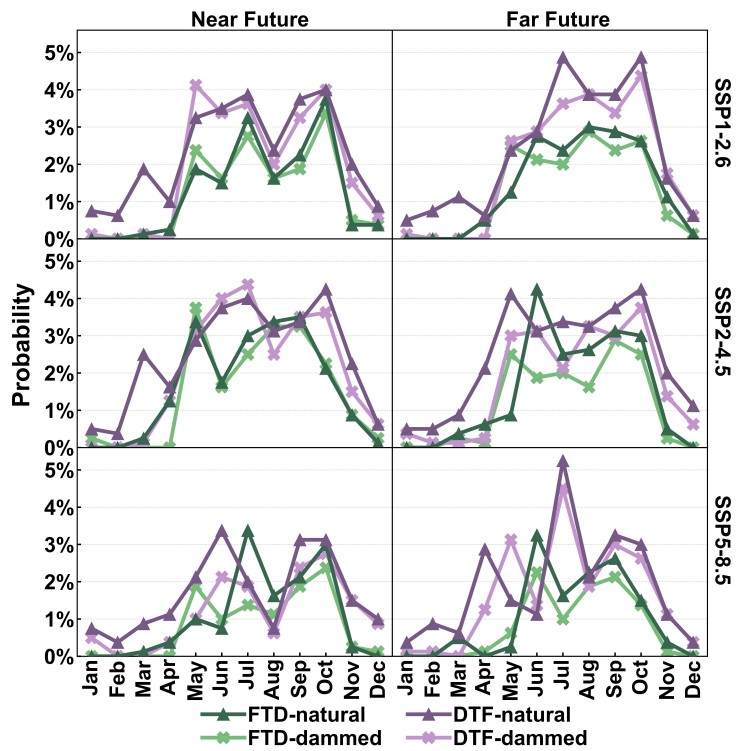

**Figure 9: Monthly DFAA probability averaged over four mainstream hydrological stations (i.e., JingHong, Nong Khai, Pakse, and Kratie stations) under natural and dammed scenarios for three SSPs during the near future (2021-2060) and far future (2061-2100) periods. Please note that the probabilities shown in this figure are averaged over 5 GCMs.**

DFAA often shows several monthly peaks under the natural scenario. This means some months have a higher DFAA probability than their neighbors. The multiple peaks are clearer in DTF than in FTD (Fig. 9). When averaging monthly DFAA over four mainstream hydrological stations, DTF shows three peaks under near-term SSP2-4.5 and far-term SSP5-8.5 scenarios, while FTD only shows two peaks in both cases. Reservoirs help to regulate DFAA by lowering and reducing peaks, with a stronger peak reduction effect anticipated in the near future for DTF (Fig. 9). In the far future, for FTD, especially under SSP1-2.6 and SSP2-4.5, reservoirs still alleviate peaks, though less so in terms of reducing their number. Reservoirs also lower DFAA probability during early and middle dry seasons (December to April) for both near and far futures, often 1% or less at most stations. Sometimes, such as the SSP2-4.5 scenario in the near future, reservoirs actually increase the probability of DFAA in May. This happens because helping during the dry season before May reduces the capacity of reservoirs for water regulation in May, making it hard to control DFAA risks that month. Reservoirs also shorten DFAA's monthly span. Instead of occurring throughout the year under the natural scenario, DFAA is to concentrated from May to October under the dammed scenario (Fig. 9). This allows the LMR Basin to focus DFAA policies and

actions on those months. As a result, riparian states can combine resources and coordinate their efforts
more efficiently to manage and respond to DFAA and related hazards.
**4. Discussion**
**4.1 Different characteristics of DTF and FTD events**
The distinct characteristics of DTF and FTD events have been identified by previous research. Shi et al.
(2021) found that FTD events predominate in the Wei River Basin. Wang et al. (2023) projected that in
the Poyang Lake Basin, the temporal spread of DTF events will expand in the future, while that of FTD
events will constrict. Ren et al. (2023) found that under SSP1-2.6 and SSP2-4.5 scenarios, the Huang-
Huai-Hai River Basin will experience more DTF events, whereas under SSP3-7.0 and SSP5-8.5 scenarios,
it will experience more FTD events. This study identifies differences between DTF and FTD events as
well, and further highlights the different characteristics of reservoirs' mitigating effects on these events.
The average probability of DTF across all periods is 2.1% under the natural scenario, which is
significantly higher than the 1.4% average for FTD (Fig. 5a). The probability of DTF consistently
exceeds that of FTD under three different intensities (Fig. 6). Additionally, DTF probabilities show a
significant increase in both the near and far future, averaging 0.23%, which exceeds the increase in FTD
probabilities, averaging 0.13% (Fig. 5a).
Compared with FTD events, reservoirs more effectively control DTF probabilities, significantly lowering
DTF risk in both dry and wet seasons (Fig. 7). The reason is that the timing of DTF's water regulation
matches the way reservoirs operate. At the start of DTF, reservoirs typically hold water at the storage
corresponding to the normal water level, which equates to 0.8 times the maximum storage (Eq. (20)).
Hence, reservoirs possess sufficient storage capacity to mitigate the drought conditions. In parallel, the
water release during the initial phase of the DTF reduced the water level, thereby meeting the storage
needs for sudden floods that occur later in the DTF. As a result, even if DTF events are frequent,
reservoirs can manage them well. Reservoirs especially succeed in reducing mild DTF events (Fig. 8).
However, they control moderate DTF events less effectively. In intense DTF cases, the rules for operating
reservoirs are not enough. For example, if a severe drought at DTF's beginning exceeds reservoir storage,
they cannot effectively relieve the extreme drought and thus fail to control such DTF events.
Although FTD is less likely than DTF, reservoirs control FTD less effectively, especially in the dry season

(Fig. 7). The problem is that when the FTD event occurs, reservoirs are generally maintained at their
target storage for the wet season. The storage corresponds to the flood control water level, which is 1.2
times the minimum storage capacity (Eq. (19)). Consequently, reservoirs, while fully meeting flood
control requirements at the start of FTD, struggle to maintain sufficient water storage to satisfy water
supply demands for the subsequent drought stage. If FTD occur frequently, reservoirs' control decreases
further. While reservoirs do little for mild FTD, they noticeably reduce moderate FTD (Fig. 8). This
means that, for rare but strong FTD events, reservoirs can help by storing water for later droughts.
However, if FTD is frequent, current reservoir operations do not help much. This difficulty in regulation
is what makes FTD a major challenge. It is encouraging, though, that FTD is expected to become less
common in most areas of the LMR Basin in the future (Fig. 5).
**4.2 The relationship between reservoirs' mitigation roles and their storage**
The reservoir systems provide enhanced mitigation efficiency against DFAA at JingHong and Kratie
compared to those at Nong Khai and Pakse (Fig. 7). Reservoir storage in the region above JingHong and
the Pakse to Kratie region is significantly larger than storage in the JingHong to Nong Khai and Nong
Khai to Pakse regions (Fig. 1c). Reservoirs' capacity to reduce total DFAA risk closely relates to the total
storage of mainstream and tributary reservoirs, consistently showing a positive correlation for DTF and
FTD events (Fig. 10a). These findings highlight reservoirs' multifaceted role in managing flood
prevention and drought resistance (Hecht et al., 2019; Hoang et al., 2019; Ly et al., 2023) while also
addressing sudden DFAA challenges. These results align with Feng et al.' s (2024) discovery that large
reservoirs significantly reduce drought and flood risks and corroborate Ehsani et al.' s (2017) conclusion
that increased dam dimensions can mitigate water resource vulnerability to climate uncertainties.
The positive correlation between total reservoir storage and the reduction of total DFAA risk indicates
that basins with larger total storage are better equipped to resist DFAA events. However, this study
examines only hydroelectric reservoirs in the LMR Basin and excludes other water storage facilities such
as irrigation reservoirs. In the LMR Basin, total storage of irrigation reservoirs is considerable. According
to the MRC, the Mekong Basin contains 1317 irrigation reservoirs, with total storage of about 17 billion
$m^3$ (MRC, 2018; LMC and MRC, 2023). This storage exceeds the total storage of reservoirs between
JingHong and Nong Khai stations (around 9.7 billion $m^3$). It is slightly lower than the storage between
Nong Khai and Pakse stations (approximately 22.1 billion $m^3$) (Figs. 1c and 10). Since reservoirs mitigate
extreme hydrological events regardless of their primary function (Brunner, 2021a; Ho and Ehret, 2025),
even irrigation reservoirs can play a beneficial role in addressing DFAA events. Fully utilizing irrigation
reservoirs and implementing coordinated operation of all reservoir types across the LMR Basin could
effectively lower DFAA risks and enhance the basin's resistance to these events.

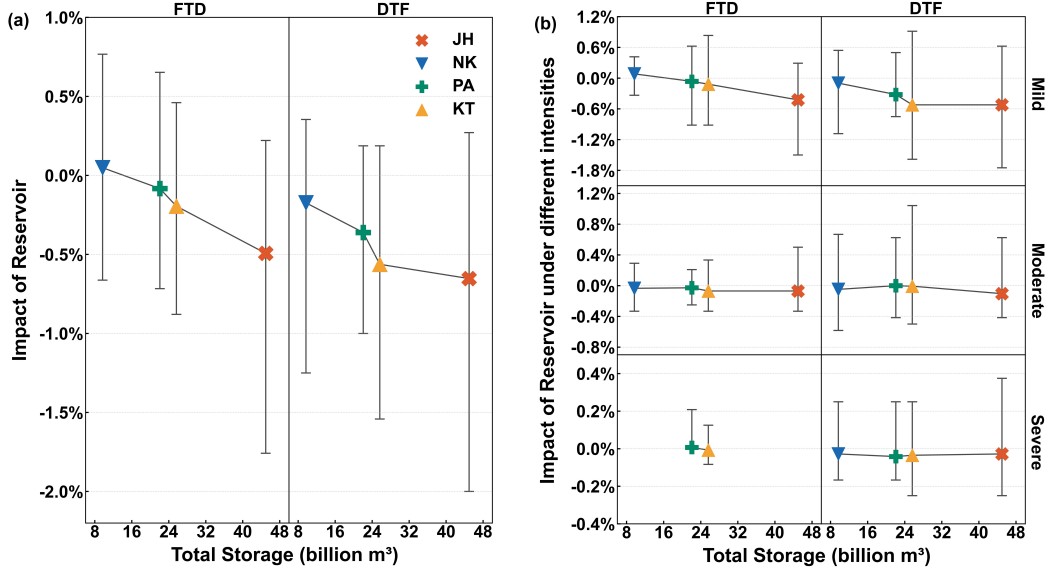


**Figure 10: The relationship between reservoirs' mitigation effects and their total storage. Symbol points**
**denote the average values for each station under three SSP scenarios during the near future (2021-2060) and**
**far future (2061-2100) periods, while error bars indicate the maximum and minimum values. (a) The impact**
**of reservoirs on the total probability of DFAA. (b) The impact of reservoirs on DFAA of different intensities.**
**Here, JH, NK, PA, and KT respectively denote JingHong, Nong Khai, Pakse, and Kratie stations. Please note**
**that, as JingHong and Nong Khai stations are not expected to experience severe FTD events in the future, the**
**relevant information has not been included in this figure.**
Both mild DTF and mild FTD show a positive correlation with total reservoir storage, consistent with
total DFAA events (Fig. 10b). In contrast, moderate and severe DFAA events do not strongly correlate
with reservoir storage (Fig. 10b). This implies that for moderate to severe DFAA events, increasing
reservoir storage capacity does not enhance the reservoirs' control capabilities. Therefore, refining
reservoir operation rules presents a more appropriate strategy to strengthen control of moderate and
severe DFAA events in the LMR Basin.
**4.3 Limitations of reservoir regulation rules**
The reservoir operation rule SOP adopted in this study is a commonly used method. Previous studies
have widely employed this method (Wang et al., 2017a; Yun et al., 2020). The SOP rule is proven
appropriate for hydrological modeling in large-scale basins such as the LMR Basin. It is also effective
for extended simulation periods in future hydrological assessments (Wang et al., 2017b; Yun et al., 2021a;
Yun et al., 2021b).
This study further improved the standard SOP operation rules by adding the general case and emergency
case (Fig. 2). This scheduling approach manages reservoir operations using real-time inflow data. It also
considers the operational year of each reservoir. As a result, the reservoir module developed in this study
is robust and adaptable. It reflects reservoir scheduling scenarios with high reliability.
Despite this, the study uses uniform operation rules for reservoirs of different storage scales within the
LMR Basin. It implements daily regulation for all reservoirs. The study does not use differentiated
regulation scales (daily, annual, or multi-annual) based on storage. It also does not consider unique
operation rules in different sub-basins. These simplifications may cause uncertainties in how reservoirs
mitigate effects. This is a limitation of the study.
**5. Conclusion**
This study adopts CMIP6 meteorological data, applying three SSP scenarios and five GCMs. It corrects
these data using the MBCn method. The study integrates the THREW distributed hydrological model
and the developed reservoir module. It describes DFAA through R-SDFAI, assessing mild, moderate, and
severe intensities. The study explores how reservoirs help reduce DFAA under the changing climate in
the LMR Basin. It examines three periods: history (1980-2014), near future (2021-2060), and far future
(2061-2100). The main findings are summarized below:
1. DFAA in the LMR Basin is dominated by DTF, with a mean probability of 2.1%. This is much higher
than the FTD probability of 1.4%. DTF remains higher than FTD at all intensity levels. The future
increase in DTF probability (average 0.23%) is also greater than the increase for FTD (average 0.13%).
Mild-intensity DFAA events are most common. They account for 58% to 90% of future DTF probability
and 75% to 100% of FTD probability. Both DTF and FTD present higher DFAA risk during the wet
season than the dry season.
2. Reservoirs manage DTF probability well, cutting DTF risks in both dry and wet seasons. However,
they have less influence over FTD risks, especially during dry-season FTD events. Limited capacity to
control FTD risks is a challenge. Reservoirs do better at managing high-frequency DTF and high-
intensity FTD events. They also cut down multi-peak DFAA events and reduce their monthly duration.

3. Reservoirs' ability to lower DFAA total risk is linked to their combined storage. Using large irrigation reservoirs within the LMR Basin can help withstand mild DFAA risks and overall events. To better handle moderate and severe DFAA events, reservoir operations need to be optimized.

This study gives new insights into how reservoirs help mitigate DFAA in the LMR Basin. It also aids water management for riparian countries. DFAA remains a serious challenge. This shows the need for LMR Basin countries to work together, build capacity against DFAA events, reduce climate change effects, and support sustainable development.

## Author contribution

**KZ:** Conceptualization; Data curation; Model development; Investigation; Methodology; Validation; Visualization; Writing - original draft; Writing - review & editing. **ZZ**: Writing - review & editing. **FT:** Conceptualization; Funding acquisition; Investigation; Methodology; Supervision; Writing - review & editing.

## Competing interests

At least one of the (co-)authors is a member of the editorial board of Hydrology and Earth System Sciences.

## Data availability

The hydrological data can be accessed and requested from the MRC Data Portal (https://portal.mrcmekong.org/home, last access: October 2025). Information related to dams is available on the Mekong Region Futures Institute (MERFI) website (https://www.merfi.org/mekong-region-dams-database, last access: October 2025). The raw CMIP6 data without correction is available at (https://esgf-node.llnl.gov/search/cmip6/, last access: October 2025). The MBCn algorithm can be accessed and implemented through an R package, which is available at (https://CRAN.R-project.org/package=MBC, last access: October 2025).

## Acknowledgment

This research was funded by the National Natural Science Foundation of China (U2442201,

723     51961125204).

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
