# Peer review of "Mitigating the Impact of Increased Drought-Flood"

_EGUsphere, 2025_

## Author Comment (AC1)

Response to Reviewer 1

1. **The introduction lacks sufficient discussion and comparison with recently published studies that examine the role of reservoir modules in hydrological modeling under climate projections.**

Thank you for your constructive comment. In the introduction part, we have discussed the role of reservoirs in mitigating hydrological impacts of climate change, such as controlling flood levels and reducing flood events. We will further elaborate on the functions of reservoir operation in addressing climate-related challenges in the revised version.

2. **Authors mentioned CMIP6 data collected from five GCMs, but only show the averaged meteorological data. Since each GCM may incorporate different assumptions and mechanisms for projecting climate variables, relying solely on the mean values could introduce bias or obscure important variability. If averaging is justified, please provide a clear rationale.**

We sincerely appreciate your comment. As each GCM possesses unique structure and assumptions, projections of climate change by a single GCM inherently possess uncertainties, which in turn introduce uncertainties in the simulation of hydrological outcomes (Kingston et al., 2011; Thompson et al., 2014). Thus, averaging across multiple GCMs is a crucial approach, as it minimizes model biases, eliminates outliers, reduces uncertainties, and ensures more robust and universally applicable outcomes (Lauri et al., 2012; Hoang et al., 2016; Wang et al., 2024; Yun et al., 2021b). This method has been extensively employed in prior studies (Dong et al., 2022; Li et al., 2021; Wang et al., 2022; Yun et al., 2021a). In Section 2.4, we have provided a concise overview of this method, and in the revised manuscript, we will expand on it with more comprehensive explanations. Furthermore, we will enhance the relevant results and provide a more thorough analysis of the extreme values in GCM outputs.

3. **Please list the equations to calculate the Standardized Runoff Index (SRI).**

Thank you for your comment. The formula for the Standard Runoff Index (SRI) is provided below and will be incorporated into the revised manuscript.

The probability density function that satisfies the Gamma distribution for runoff x at a given time period is:

$$g(x) = \frac{1}{\beta^{\alpha}\Gamma(\alpha)}x^{\alpha-1}e^{-\frac{x}{\beta}}, \ \ x > 0$$

where, $\alpha > 0$ and $\beta > 0$ are respectively the shape and scale parameters. $\hat{\alpha}$ and $\hat{\beta}$ are the optimal values of $\alpha$ and $\beta$, obtained according to the maximum likelihood estimation method. $\Gamma(\alpha)$ is the gamma function.

$$\hat{\alpha} = \frac{1}{4A}\left(1 + \sqrt{1 + \frac{4A}{3}}\right)$$

$$\hat{\beta} = \frac{\bar{x}}{\hat{\alpha}}$$

$$A = \ln(\bar{x}) - \frac{\sum \ln(x_i)}{n}$$

$$\Gamma(\alpha) = \int_0^\infty y^{\alpha-1} e^y \, dy$$

Where, $x_i$ is the sample of runoff sequence, $\bar{x}$ is averaged runoff, and $n$ is the length of runoff sequence.

Then the cumulative probability of runoff x is illustrated as follow.

$$G(x) = \int_0^x g(x)\,dx = \frac{1}{\hat{\beta}^{\hat{\alpha}}\Gamma(\hat{\alpha})}\int_0^x x^{\hat{\alpha}-1} e^{-\frac{x}{\hat{\beta}}}\,dx, \quad x > 0$$

**4. It is unclear that how the probability calculates in equation (22).**

We sincerely appreciate for your comment. The impact of reservoirs on DFAA probability in the certain period is quantified by subtracting the DFAA probability in that period under the natural scenario from that under the dammed scenario, as defined in Eq. (22). For instance, the impact of reservoir operation on DFAA during the near future can be assessed by finding the difference between the DFAA in the near future probability under the dammed scenario and that under the natural scenario.

Since this study focuses on DFAA events at the monthly scale, the probability of DFAA events during a specific period, as per Eq. (22), is determined by the ratio of months with DFAA events occurred ($|R - SDFAI| > 1$) to the total number of months in that period. More precisely, the proportion of months with DTF events ($R - SDFAI > 1$) to the total number of months signifies the probability of DTF events, whereas the proportion of months with FTD events ($R - SDFAI < -1$) to the total number of months indicates the probability of FTD events.

We will make the following adjustments to this formula in the revised version, and add detailed descriptions to enhance its clarity and precision.

$$P_{\text{Impact of Reservoirs},i,t} = P_{\text{Dammed},i,t} - P_{\text{Natural},i,t}$$

Where $P_{\text{Impact of Reservoirs},i,t}$ represents the impact of reservoirs on the probability of event t in period i. $P_{\text{Natural},i,t}$ denotes the probability of event t under the natural scenario in period i while

the $P_{Dammed,i,t}$ denotes the probability of event t under the dammed scenario in period i. Period i refers to the near future period and the far future period. Event t indicates the DTF events, FTD events and DFAA events.

$P_{Natural,i,t}$ and $P_{Dammed,i,t}$ described above are calculated by the following formulas.

$$P_{Natural,i,t} = \frac{M_{Natura,i,t}}{TM_i}$$

$$P_{Dammed,i,t} = \frac{M_{Dammed,i,t}}{TM_i}$$

Where $M_{Natura,i,t}$ denotes the number of months in which event t occurs in period i under the natural scenario. $M_{Dammed,i,t}$ denotes the number of months occurred event t occurs in period i under the dammed scenario. $TM_i$ refers to the total number of months in period i. Period i refers to the near future period and the far future period. Event t indicates the DTF events, FTD events and DFAA events.

5. **While the results show changes in indicator probabilities across different scenarios and time scales, the influence of reservoir operations on DFAA remains unclear. Are the operations temporally and spatially variable? Further clarification is necessary to understand the extent and mechanism of reservoir operations.**

Thanks for your comment. Reservoir operation rules remain consistent over time and space, as demonstrated in Eq. (2) - (21) within Section 2.4. The Standard Operation Policy hedging model is consistently applied to all reservoirs in the LMR Basin. The spatial distribution of reservoirs and their capacities is shown in Figs. 1a and 1c. Reservoirs mainly function as storage pools to mitigate DFAA events by controlling water storage and release. This function differently affects DTF and FTD events. During DTF events, reservoirs can release water during the drought phase and utilize low water levels to accommodate floodwaters later. However, managing FTD events presents challenges for reservoirs, as they must balance flood mitigation in the early phase with drought mitigation in the later phase. Therefore, we also further note in Section 4.2 that incorporating hydrological forecasts will improve the reservoir's ability to mitigate DFAA events. We will enhance the relevant sections in the revised version to improve readability and clarity.

6. **Are the reservoirs operations the dominant factor of DFAA events in the Lancang-Mekong River Basin? Please comment it.**

We appreciate your comment. According to our research findings, reservoir operation is not the dominant factor influencing DFAA events in the LMR Basin. In the natural scenario without reservoirs, DFAA will experience notable changes due to climate change, including increased annual DFAA risks under SSP1-2.6 and SSP2-4.5 scenarios, more significant increases in upstream FTD risks, and more pronounced increases in downstream DTF risks, as discussed in Section 3.3. These changes are entirely unaffected by reservoir operations. Furthermore, reservoirs significantly mitigate DFAA events, particularly by effectively reducing annual DTF risks, wet season's FTD risks, lowering the monthly probability peaks of DFAA, and decreasing the number

of peak events, as described in Section 3.4. Our analysis indicates that while reservoir operations can effectively reduce the probability of DFAA events under climate change, they are not the primary factor responsible for the increase in DFAA events. We will provide further clarification on this in the revised version.

**References**

Dong, Z., Liu, H., Baiyinbaoligao, Hu, H., Khan, M., Wen, J., Chen, L., Tian, F.: Future projection of seasonal drought characteristics using CMIP6 in the Lancang-Mekong River Basin. J. Hydrol. 610 https://doi.org/10.1016/j.jhydrol.2022.127815, 2022.

Hoang, L. P., Lauri, H., Kummu, M., Koponen, J., van Vliet, M. T. H., Supit, I., Leemans, R., Kabat, P., and Ludwig, F.: Mekong River flow and hydrological extremes under climate change, Hydrol. Earth Syst. Sci., 20, 3027–3041, https://doi.org/10.5194/hess-20-3027-2016, 2016.

Kingston, D. G., Thompson, J. R., and Kite, G.: Uncertainty in climate change projections of discharge for the Mekong River Basin, Hydrol. Earth Syst. Sci., 15, 1459–1471, https://doi.org/10.5194/hess-15-1459-2011, 2011.

Lauri, H., de Moel, H., Ward, P. J., Räsänen, T. A., Keskinen, M., and Kummu, M.: Future changes in Mekong River hydrology: impact of climate change and reservoir operation on discharge, Hydrol. Earth Syst. Sci., 16, 4603–4619, https://doi.org/10.5194/hess-16-4603-2012, 2012.

Li, Y., Lu, H., Yang, K., Wang, W., Tang, Q., Khem, S., Yang, F., Huang, Y.: Meteorological and hydrological droughts in Mekong river basin and surrounding areas under climate change. J. Hydrol.: Reg. Stud. 36, 100873. https://doi.org/10.1016/j.ejrh.2021.100873, 2021.

Thompson, J., Green, A., & Kingston, D: Potential evapotranspiration-related uncertainty in climate change impacts on river flow: An assessment for the Mekong River basin. Journal of Hydrology, 510, 259–279. https://doi.org/10.1016/j.jhydrol.2013.12.010, 2014.

Wang, A., Miao, Y., Kong, X., & Wu, H: Future changes in global runoff and runoff coefficient from CMIP6 multi-model simulation under SSP1-2.6 and SSP5-8.5 scenarios. Earth's Future, 10(12), e2022EF002910. https://doi.org/10.1029/2022EF002910, 2022.

Wang, C., Leisz, S., Li, L., Shi, X., Mao, J., Zheng, Y., and Chen, A.: Historical and projected future runoff over the Mekong River basin, Earth Syst. Dynam., 15, 75–90, https://doi.org/10.5194/esd-15-75-2024, 2024.

Yun, X., Tang, Q., Li, J., Lu, H., Zhang, L., Chen, D: Can reservoir regulation mitigate future climate change induced hydrological extremes in the lancang-Mekong River Basin? Sci. Total Environ. 785. https://doi.org/10.1016/j.scitotenv.2021.147322, 2021a.

Yun, X., Tang, Q., Sun, S., & Wang, J.: Reducing climate change induced flood at the cost of hydropower in the Lancang-Mekong River Basin. Geophysical Research Letters, 48, e2021GL094243. https://doi.org/10.1029/2021GL094243, 2021b.

---

## Author Comment (AC2)

Response to Reviewer 2

**1.  Line 18: It is recommended to describe the results using the conditions of the emission scenario rather than the version of the scenario.**

We appreciate your comment. In the revised version, we will replace SSP126, SSP245, and SSP585 with concerted mitigation efforts, limited mitigation efforts, and no mitigation efforts in the abstract section. Furthermore, we will update all references to SSP126, SSP245, and SSP585 in the manuscript, adopting the terms SSP1-2.6, SSP2-4.5, and SSP5-8.5. Additionally, we will provide a detailed explanation of these three emission scenarios in Section 2.2.

**2.  Line 36: Please supplement which secondary disasters.**

Thanks for your comment. The secondary hazards mentioned primarily include mountain floods, landslides, and debris flows. We will incorporate this information in the revised version.

**3.  Line 63: How did previous hydrological models simulate DFAA? How has the reservoir module progressed in hydrological models?**

We sincerely appreciate your comment. Existing studies primarily utilize specific indices, such as LDFAI (Long-cycle Drought-Flood Abrupt Alternation Index) and SDFAI (Short-cycle Drought-Flood Abrupt Alternation Index), to quantify DFAA events. These indices leverage precipitation and runoff data to characterize meteorological and hydrological DFAA events. Hydrological DFAA events specifically require the use of hydrological models. A discussion on these aspects is currently presented in Section 2.5 of the manuscript. In the revised version, we intend to relocate this discussion to the introduction part and further refine it. Additionally, we recognize the scarcity of existing research on DFAA events in the LMR Basin. To address this, we plan to incorporate a discussion on DFAA events in other basins within the introduction. Moreover, we aim to provide an enhanced review of the reservoir module within hydrological models.

**4.  Line 78: Where is the population data obtained?**

We appreciate your query. The population data cited is sourced from Sabo et al., 2017, and Luo et al., 2023, and we intend to include this reference in the revised version.

**5.  Line 100: Are there any other GCMs? Are only these five available, or do these five have better effects?**

We appreciate your comment. We selected the five GCMs that are widely applied and demonstrate robust performance in the LMR Basin. Their reliability has been confirmed by studies such as Li et al. 2021, Yun et al. 2021a, and Yun et al. 2021b.

**6.  Section 2.2: There need usage instructions for the data. For instance, if the precipitation and temperature of ERA5 are used to correct GCMs, then what is the potential evapotranspiration used for?**

Thank you for your comment. The evapotranspiration data of ERA5_Land dataset are utilized to derive the evapotranspiration data in the future period. The Van Peltetal method (Van Pelt et al., 2009) is implemented for this purpose. The detailed description of the methodology and its calculation formula will be included in the revised version. Additionally, we will enhance the description of other data-related content in the revised version.

7. **Section 2.3: As the core method of this section, the main formulas of MBG should be listed.**

We appreciate your suggestion. In the revised version, we will incorporate the two most critical formulas of the MBCn (Multivariate Bias Correction via N-dimensional Probability Density Function Transform) method (Cannon, 2018):

(1) Random Orthogonal Rotation:

$$\widetilde{X}_T^{[j]} = X_T^{[j]} R^{[j]}$$

$$\widetilde{X}_S^{[j]} = X_S^{[j]} R^{[j]}$$

$$\widetilde{X}_P^{[j]} = X_P^{[j]} R^{[j]}$$

This formula outlines the process of transforming historical observations $X_T^{[j]}$, historical climate

model simulations $X_S^{[j]}$, and climate model projections $X_P^{[j]}$ using a random orthogonal rotation

matrix $R^{[j]}$ during the j-th iteration. The rotated data are represented as $\widetilde{X}_T^{[j]}$, $\widetilde{X}_S^{[j]}$, and $\widetilde{X}_P^{[j]}$.

This procedure is pivotal for MBCn's multivariate joint distribution correction, as it transforms the original variable space into new random orientations. In contrast to conventional uni-variate correction approaches, MBCn employs a random orthogonal matrix to mix variables, thereby breaking their independence.

(2) Quantile Delta Mapping:

$$\Delta^{(n)[j]}(i) = \tilde{x}_P^{(n)[j]}(i) - F_S^{(n)[j]^{-1}}(F_P^{(n)[j]}(\tilde{x}_P^{(n)[j]}(i)))$$

$$\hat{x}_P^{(n)[j]}(i) = F_T^{(n)[j]^{-1}}(F_P^{(n)[j]}(\tilde{x}_P^{(n)[j]}(i))) + \Delta^{(n)[j]}(i)$$

The formula defines how quantile delta mapping is applied to n-th dimension of the rotated

climate model projection data $\tilde{x}_P^{(n)[j]}(i)$ within the rotated space of the j-th iteration. Here,

$\Delta^{(n)[j]}(i)$ represents the quantile difference between the historical climate model simulations and

climate model projections in the j-th iteration and the n-th dimension. $F_P^{(n)[j]}$ denotes the empirical cumulative distribution function for the rotated climate model projection data in the n-th dimension. $F_T^{(n)[j]^{-1}}$ and $F_S^{(n)[j]^{-1}}$ denote inverse Functions of the empirical cumulative distribution functions for the rotated historical observation data and historical climate model simulation data in the n-th dimension. This step preserves the trend of the climate model projection data throughout the correction process. The number of iterations is typically set to 10-30.

The MBCn algorithm performs multivariate joint distribution bias correction by iteratively applying the random orthogonal rotation and quantile delta mapping, while preserving the projected signals in the climate model. The rotation operation breaks dependencies between variables, enabling the quantile delta mapping of single variable to indirectly adjust multivariate correlations. The quantile delta mapping ensures the transmission of absolute or relative trends by computing quantile differences between the historical and projected periods of the climate model. The MBCn algorithm is accessible and implementable through an R package, available at: https://CRAN.R-project.org/package=MBC (last accessed at July 4, 2025).

8. **Section 2.4: Why are Formula 3 and Formula 8 repeated? Can so many simple formulas be explained in the main text? The principle of reservoir allocation is suggested to be shown in a schematic diagram because these formulas are both numerous and simple.**

Thanks for your comment. Eqs. (3) and (8) represent the water balance equations required during the initial phase and the normal phase of the reservoir, respectively. We exhibit both equations in order to provide a clearer and more comprehensive explanation of the operation rules for both phases. As per your suggestion, we will include a flowchart of reservoir operation in the revised version.

9. **Line 145: For the complex physical mechanisms in the model, there are no formulas at all? What are the equilibrium equations, geometrical relationships, and constitutive relationships in the model? The Nash efficiency coefficient is relatively less necessary to present.**

We appreciate your suggestion. In the revised version, we will provide a more comprehensive description to the THREW model to increase readers' understanding and knowledge of it.

10. **Section 2.4: The GCM model is spatially distributed grid data, and the reservoir here is a lumped water distribution. How can a simple lumped water distribution be regulated regionally?**

Thanks for your comment. As indicated in line 144, the THREW model performs spatial basin delineation based on Representative Elementary Watershed (REW). Within the LMR Basin, the THREW model delineates 651 REWs units and conducts runoff simulations based on these REW. The reservoir module also employs REW format for reservoir operation. For GCM data, to meet

the needs of hydrological simulation in the THREW model, we downscale the GCM data from grid scale to REW scale and utilize the downscale GCM data as meteorological input for the model. We will provide the detailed explanation of this aspect in the revised version.

11. **Line 215: I thought that the five GCMs used for simulation could mutually test the reliability, but here the average value was directly used without analyzing the sensitivity of the five models. GCMs' errors are not complementary. Some may be more accurate, while others have larger errors. A simple average value is of no help to the research.**

We appreciate your comment. Since each GCM has its unique structure and assumptions, leading to varied runoff results and DFAA characteristics, conducting mutual reliability verification among the five GCMs may not be appropriate. We average multiple GCMs because a single GCM's projection of climate change involves uncertainty, which affects hydrological simulation results (Kingston et al., 2011; Thompson et al., 2014). Considering the average of GCMs helps reduce model bias, prevent outliers, minimize uncertainty, and provide more robust results (Lauri et al., 2012; Hoang et al., 2016; Wang et al., 2024; Yun et al., 2021b). This method is commonly used in existing studies (Dong et al., 2022; Li et al., 2021; Wang et al., 2022; Yun et al., 2021a). Additionally, we will improve the results by adding elaborations of the extreme value in GCM outcomes in the revised version.

12. **Line 214: According to the abstract, "Reservoir operations reduce DFAA's intensity." It should be getting the intensity of the DFAA, why there is a probability, and how to quantify intensity.**

Thanks for your comment. In this work, we assess the risk of DFAA events by calculating their probability, but don't consider their intensity. We sincerely apologize for the inaccuracy terminology used in the abstract section. We will correct it in the revised version. We sincerely appreciate your correction.

13. **Please pay attention to the garbled characters that appear in lines 156, 242, and 243.**

Thank you for your reminder. We will focus on this issue and address it in the revised version.

14. **When many formulas are piled up and there are no corresponding textual descriptions, it is very difficult to know what the logic between them is. Here, it is necessary to select the most important ones from these formulas for listing and then describe the logic of the formulas. Furthermore, what's these methods' regional applicability? What are their advantages and limitations?**

Thank you for your suggestion. We will adjust the displayed equations in the revised version, , supplement their logical relationships, and emphasize their applicability and limitations.

15. **Section 3.1: Since the study originally used ERA5 for correction, it doesn't mean that being closer to ERA5 is accurate. ERA5 also has errors. It can only be said that after correction, the GCM is closer to ERA5, and this cannot be used as an accurate basis here. Even this**

**subsection can be transformed into a description of the spatiotemporal distribution of climate data.**

We appreciate your suggestion. In the revised version, we will modify the presentation of Section 3.1. However, we would like to note that correcting future meteorological data by reanalysis datasets or remote sensing datasets is a common practice. In the existing studies, for example, Hoang et al. (2016) utilized WATCH forcing data and APHRODITE datasets to correct the precipitation and temperature of GCM data. Ly et al. (2023) applied Global Precipitation Climatology Centre (GPCC) to correct the precipitation in the GCM data. Wang et al. (2021) employed precipitation data from the Climate Prediction Center (CPC) to correct the GCM data. Yun et al. (2021a) and (2021b) used the Global Meteorological Forcing (GMFD) dataset to correct ISMIP3b data. Therefore, we think it is reasonable to apply the ERA5 data to correct the bias of each GCM in the CMIP6 data.

16. **Section 3.2: The absence of reservoirs before 2009 and the existence of reservoirs after 2010 should be very important background. When the coupled reservoir module is used for DFAA simulation, it should be simulated in segments. For those after 2010, additional reservoirs should be added. What will the situation of reservoirs be like in future scenarios? This needs to be explained in the summary and subsequent sections of the DFAA results.**

Thanks for your comment. We would like to point out that "The statement "The absence of reservoirs before 2009 and the existence of reservoirs after 2010" is a general concept. However, in reality, reservoir construction and operation began before 2009 (Zhang et al., 2023). The earliest reservoirs in the reservoir data we utilize were constructed in 1965 and the latest in 2035. We considered the annual change of the reservoir storage in the reservoir module. For each new reservoir, its operation is initially conducted based on the operation rules during the initial phase. Once its storage reaches the minimum constraint, the new reservoir enters the normal phase and follows the rules of normal phase. The reservoir operation in future periods will also follow these patterns.

17. **Section 3.3: Shouldn't this probability be compared with the occurrence of a single disaster before? The probability can be calculated based on the time within a year.**

Thanks for your comment. Currently, there is a paucity of detailed statistics on individual flood and drought events in the LMR Basin, which presents challenges in determining the probability of DFAA events by the single disaster. Moreover, we adopt $|R - \text{SDFAI}| > 1$ as the criterion for the occurrence of DFAA events, which means that we identify DFAA events that are at least a transition between a mild hydrological drought event (standard runoff index $(\text{SRI}) < - 1$) and a mild hydrological wet event (standard runoff index $(\text{SRI}) > 1$) (Song et al., 2023). We plan to enhance this section in the revised version.

18. **Section 3.4: How will the future reservoir operation information be obtained?**

We appreciate your comment. The reservoir dataset we collected encompasses future planned reservoirs, with the latest one scheduled to begin operations in 2035. It's noted that the reservoir

storage is scheduled to increase significantly in the future period notable and the capacity of tributary reservoirs during this period is sizable, especially in downstream reservoirs, as shown in Fig. 1c.

19. **The discussion section should use more literature to support the causes and reliability of the results. Here, for example, in the first part of the discussion, except for the first sentence, which is cited, the rest is all about explaining the results.**

Thank you for your suggestion. We will provide additional discussions on different characteristics of DTF and FTD incorporate existing relevant studies to further substantiate our arguments.

20. **The second part of the discussion talked about the reservoir's ability to respond to DFAA. Here, in addition to considering the changes in the water volume of the reservoir, it is also necessary to consider how long the reservoir operation occurred before or after the disaster. The occurrence time of reservoir operation will have a timely impact on the specific disaster.**

Thanks for your comment. The second part of the discussion part centers on the role of hydrological forecasting in improving reservoirs' ability to respond to DFAA events. As highlighted in Section 4.1, the differing levels of control reservoirs exhibit over DTF and FTD risks are attributed to their constrained ability to fully harness storage when encountering completely unforeseen inflows. To address this challenge, we propose the integration of hydrological forecasting to enhance the reservoirs' mitigative capacity. The occurrence time of reservoir operation remains constant under the SOP-based module, while the inclusion of hydrological forecasting enables adaptive operation strategies informed by forecast data, which we will provide examples of in the revised manuscript.

21. **The discussion in the third part also rarely cites literature, and the utilization of the resilient storage should not be the focus of the discussion in this article. The focus is on the influence of the reservoir in the process of disaster simulation.**

We appreciate your comment. Given the alignment between reservoir mitigation effects and storage capacity distribution, our analysis in Section 4.3 highlights that irrigation reservoirs, alongside hydropower reservoirs, play a crucial role in diminishing the likelihood of DFAA events and mitigating flood and drought pressures in the LMR Basin. Moreover, the substantial total storage capacity of irrigation reservoirs is a key consideration. This finding is vital for policymakers and stakeholders, as it informs the development of a cohesive dispatch network that integrates both power and irrigation reservoirs, thereby fostering joint flood and drought prevention measures.

22. **In addition, the bar charts and line charts from Figure 4 to Figure 6 are all numbers that can be presented in a table, and the richness of the accompanying figures should be increased.**

Thanks for your comment. We plan to adjust Figs. 4 and 5 in the revised manuscript in response to your suggestion to display not only the average of five GCMs but also their maximum and minimum values, thereby enriching the figures' information and providing further clarification of DFAA events under different GCMs.

**References**

[revised manuscript text omitted]

---

## Author Response (AR1)

We would like to express our thanks for your constructive and valuable comments and suggestions on our manuscript, which has significantly improved it. We have revised the manuscript thoroughly based on all the comments. The reviewer's comments are enumerated. Our replies to each comment start with "Response".

In the revised manuscript, we added Dr. Zilong Zhao as a co-author, who contributed substantially to the revision, particularly in interpreting the results and revising the manuscript structure. All authors have agreed to this addition, and we believe that Dr. Zilong Zhao's contribution warrants his inclusion as an author. We appreciate your understanding and hope that this addition does not cause any inconvenience.

============================================================================

**Reviewer #1: This paper analysed the Drought-Flood Abrupt Alternation (DFAA) in the Lancang-Mekong River Basin under three climate projection scenarios from five Global Climate Models (GCMs) of CMIP6. Authors found that future DFAA trend varies widely in upstream and downstream and reservoirs operations can reduce DFAA's intensity, limit multiple peaks and shorten the monthly span. The paper is structured, however, there are some concerns.**

**Response:** We sincerely appreciate your positive comments and valuable insights on potential enhancements for our paper. Kindly find our detailed responses provided below.

1. **The introduction lacks sufficient discussion and comparison with recently published studies that examine the role of reservoir modules in hydrological modeling under climate projections.**

**Response:** Thank you for your constructive comment. We added a description of the reservoir module in hydrological modeling under climate change in lines 75 to 83 of the revised version. It reads as follows:

Research has shown that reservoirs play a crucial role in preventing extensive damages during the wet season and in minimizing low-flow occurrences in LMR Basin (Arias et al., 2014; Räsänen et al., 2012; Dang et al., 2024). The integration of a coupled reservoir module within the hydrological model is a widely adopted approach for evaluating reservoir impacts under changing climate. Wang et al. (2017b) utilized this approach to show that reservoir operation can minimize flood intensity and lower flood occurrence rates. Yun et al. (2021a; 2021b) demonstrated that, despite a trade-off in hydroelectric benefits, reservoir management can substantially alleviate extreme drought and wet hydrological events in LMR Basin. These studies collectively indicated that reservoirs represent a practical solution for addressing the impacts of climate change.

2. **Authors mentioned CMIP6 data collected from five GCMs, but only show the averaged meteorological data. Since each GCM may incorporate different assumptions and mechanisms for projecting climate variables, relying solely on the mean values could introduce bias or obscure important variability. If averaging is justified, please provide a clear rationale.**

**Response:** We sincerely appreciate your comment. Each GCM generally operates under distinct assumptions and mechanisms, leading to potential uncertainties when relying on a single GCM simulation. Our main objective in averaging five GCM outputs is to reduce the uncertainties associated with GCM projections. In the earlier submission, our explanation on this point was inadequate. We expanded this explanation within Section 2.6 in the revised manuscript. Furthermore, as you underscored, the variability among GCMs requires attention. We incorporated an examination of GCM result variations in the revised manuscript. The specific

elaborations on this aspect are provided in lines 368 to 377 of the revised manuscript, which reads as below:

As each GCM possesses unique structure and assumptions, projections of climate change by a single GCM inherently possess uncertainties, which in turn introduce uncertainties in the simulation of hydrological outcomes (Kingston et al., 2011; Thompson et al., 2014). Thus, averaging across multiple GCMs is a crucial approach, as it minimizes model biases, eliminates outliers, reduces uncertainties, and ensures more robust and universally applicable outcomes (Lauri et al., 2012; Hoang et al., 2016; Hecht et al., 2019; Wang et al., 2024; Yun et al., 2021b). This method has been extensively employed in prior studies (Dong et al., 2022; Li et al., 2021; Wang et al., 2022; Yun et al., 2021a). Therefore, this research determines the average DFAA probability from five GCMs to lessen the uncertainty in their predictions and assesses the fluctuation in these probabilities across the models to demonstrate their variability.

Additionally, we revised Figs. 5 and 6, as well as the descriptions in Sections 3.3 and 3.4, to provide a more comprehensive depiction and elaboration of the variability in GCM results. The amended versions of Figs.5 and 6 are shown below for your review, and the corresponding updated content can be found in Sections 3.3 and 3.4 of the revised manuscript.

[Figure]

Figure 5: DFAA under natural scenario. Here, JH, NK, PA, and KT respectively denote JingHong, Nong Khai, Pakse, and Kratie stations. (a) Seasonal probability of DFAA averaged across five GCMs during history (1980-2014), near future (2021-2060) and far future (2061-2100) periods, as well as under three SSPs. The annual probability is half of the sum of wet and dry season probabilities. (b) The annual change in DFAA probability averaged across five GCMs and their ranges in the near and far future periods with respect to history period under three SSPs. (c) The seasonal change in DFAA probability averaged across five GCMs and their ranges in the near and far future periods with respect to history period during wet and dry seasons under three SSPs.

[Figure]

**Figure 6: Reservoir impacts on DFAA during near future (2021-2060) and far future (2061-2100) under three SSPs. Here, JH, NK, PA, and KT denote JingHong, Nong Khai, Pakse, and Kratie stations, respectively. (a) The annual reservoir impacts averaged across five GCMs and their ranges. (b) The seasonal reservoir impacts in wet and dry seasons averaged across five GCMs and their ranges.**

**3. Please list the equations to calculate the Standardized Runoff Index (SRI).**

**Response:** Thank you for your comment. We added the formula for SRI indicator in the Section 2.5 of the revised manuscript, i.e., Eqs. (32) to (37) in lines 326 to 339.

**4. It is unclear that how the probability calculates in equation (22).**

**Response:** We sincerely appreciate your comment. The impact of reservoirs on DFAA probability in a certain period is quantified by subtracting the DFAA probability in that period under the natural scenario from that under the dammed scenario. For instance, the impact of reservoir operation on DFAA during the near future can be assessed by finding the difference between the DFAA probability in the near future under the dammed scenario and that under the natural scenario.

Since this study focuses on DFAA events at the monthly scale, the probability of DFAA events during a specific period is determined by the ratio of months with DFAA events ($|R - SDFAI| > 1$) to the total number of months in that period. More precisely, the proportion of months with DTF events ($R - SDFAI > 1$) to the total number of months represents the probability of DTF events, whereas the proportion of months with FTD events ($R - SDFAI < -1$) to the total number of months indicates the probability of FTD events.

We adjusted this calculation formula in the revised version and added detailed descriptions to enhance their clarity and precision. Please refer to Eqs. (38) to (40) in the revised manuscript.

**5. While the results show changes in indicator probabilities across different scenarios and time scales, the influence of reservoir operations on DFAA remains unclear. Are the operations temporally and spatially variable? Further clarification is necessary to understand the extent and mechanism of reservoir operations.**

**Response:** Thanks for your comment. The reservoir operation rule utilized in this paper remains consistent over time and space. The Standard Operation Policy hedging model is consistently applied to all reservoirs in LMR Basin. The spatial distribution of reservoirs and their capacities is shown in Figs. 1a and 1c. Reservoirs mainly function as storage pools to mitigate DFAA events by controlling water storage and release. This function differently affects DTF and FTD events. During DTF events, reservoirs can release water during the drought phase and utilize low water levels to accommodate floodwaters later. However, managing FTD events presents challenges for reservoirs, as they must balance flood mitigation in the early phase against drought mitigation in the later phase. Therefore, we also further note in Section 4.2 that incorporating hydrological forecasts will improve the reservoir's ability to mitigate DFAA events.

In the revised manuscript, we added a flowchart for the reservoir operation module, i.e., Fig. 2 and improved the overview of the reservoir module. Please refer to lines 249 to 262 in the revised version, and it reads as follows.

This study extends the THREW model through the development of a reservoir management module that can be incorporated into it. This module contains detailed data on 122 reservoirs in the basin, with operational years ranging from 1965 to 2035. Configuring the module's activation enables the integrated THREW model to simulate natural runoff without considering reservoirs, and dammed runoff with reservoirs considered.

The reservoir operation rules are consistent over time and space, with each reservoir following the same operation rules and starting scheduling according to its respective operational year. The reservoir module conducts daily-scale reservoir operation based on sub-basins. Each reservoir is allocated to the corresponding sub-basin according to its location information. The cumulative reservoir storage over multiple years for each sub-basin is calculated and serves as an input condition for the reservoir module. The module consists of two phases: the initial phase and the normal phase. The constraints of the normal phase are further divided into general and emergency cases. Both cases share the same reservoir operation rules, but their constraints differ, with the emergency case featuring more flexible constraints. The reservoir module's flowchart is depicted in Fig. 2.

**6. Are the reservoirs operations the dominant factor of DFAA events in the Lancang-Mekong River Basin? Please comment it.**

**Response:** We appreciate your comment. According to our research findings, reservoir operation is not the dominant factor influencing DFAA events in the LMR Basin. In the natural scenario without reservoirs, DFAA will experience notable changes due to climate change, including increased annual DFAA risks under SSP1-2.6 and SSP2-4.5 scenarios, more significant increases in upstream FTD risks, and more pronounced increases in downstream DTF risks, as discussed in Section 3.3. These changes are entirely unaffected by reservoir operations. Furthermore, reservoirs significantly mitigate DFAA events, particularly by effectively reducing annual DTF risks, wet season's FTD risks, lowering the monthly probability peaks of DFAA, and decreasing the number of peak events, as described in Section 3.4. Our analysis indicates that while reservoir operations can effectively reduce the probability of DFAA events under climate change, they are not the primary factor responsible for the increase in DFAA events.

**Reviewer #2: This study evaluated the impacts of climate change and reservoir operations on Drought-Flood Abrupt Alternation (DFAA) using five Global Climate Models (GCMs) from the Coupled Model Intercomparison Project Phase 6 (CMIP6) in the Lancang-Mekong River Basin. The authors have contributed to the understanding of future DFAA. But there are still some issues that need to be clarified.**

Response: Thank you for your positive feedback and comments to further improve the paper. Please see our detailed responses below.

1. **Line 18: It is recommended to describe the results using the conditions of the emission scenario rather than the version of the scenario.**

Response: We appreciate your comment. In the revised version, we will replace SSP126, SSP245, and SSP585 with low-, medium-, and high-emission scenarios in the abstract section. Furthermore, we will update all references to SSP126, SSP245, and SSP585 with replacements in the revised manuscript using the terms SSP1-2.6, SSP2-4.5, and SSP5-8.5, including those in Figs. 5 to 7.

2. **Line 36: Please supplement which secondary disasters.**

Response: Thanks for your comment. The secondary hazards mentioned primarily include flash floods, landslides, and mudslides. We incorporated this information in lines 35 to 37 of the revised version.

3. **Line 63: How did previous hydrological models simulate DFAA? How has the reservoir module progressed in hydrological models?**

Response: We sincerely appreciate your comment. Existing studies primarily utilize specific indices, such as LDFAI (Long-cycle Drought-Flood Abrupt Alternation Index) and SDFAI (Short-cycle Drought-Flood Abrupt Alternation Index), to quantify DFAA events. These indices leverage precipitation and runoff data to characterize meteorological and hydrological DFAA events. In the revised manuscript, we added a description of the DFAA quantification methods in the introduction section, see lines 38 to 48, which reads as below.

Employing indices to characterize DFAA events is a common quantitative method. Since Wu et al. (2006) proposed the precipitation-based long-cycle drought-flood abrupt alternation index (LDFAI) to quantitatively characterize the long-term DFAA of wet season, LDFAI has been widely adopted (Ren et al., 2023; Shi et al., 2021; Yang et al., 2022; Yang et al., 2019). Zhang et al. (2012) proposed the one-month interval short-cycle drought-flood abrupt alternation index (SDFAI) based on LDFAI to characterize the short-term DFAA of wet season, and expanded the application from precipitation to runoff. SDFAI has been extensively applied in various fields such as hydrology, meteorology, ecology, and agriculture (Zhao et al., 2022; Lei et al., 2022; Yang et al., 2019; Zhang et al., 2019). Song et al. (2023) further refined the

SDFAI index and developed the Revised Short-cycle Drought-Flood Abrupt Alteration Index (R-SDFAI), which is calculated based on the Standardized Runoff Index (SRI) and designed to characterize short-term DFAA.

Additionally, we recognize the scarcity of existing research on DFAA events in the LMR Basin. To address this, we incorporated a discussion on DFAA events in other basins within the introduction section. Please refer to the lines 49 to 56 in the revised manuscript.

It has been observed that the intensity and frequency of DFAA events demonstrate a global increasing trend (Yang et al., 2022; Chen et al., 2024). However, regional differences are notable. Shan et al. (2018) observed that the scope of DFAA events in the Yangtze River mid-lower reaches has expanded since the 1960s, with both frequency and intensity increasing annually. Zhang et al. (2012) found that while droughts and floods in the Huai River Basin have increased, DFAA events have become less frequent. For future projections, Zhao et al. (2022) indicated that DFAA events in the Han River Basin will experience an upward trend in both frequency and intensity. Yang et al. (2019) reported that in the Hetao region, the number and frequency of DFAA events will diminish.

Moreover, we provided an enhanced review of the reservoir module within hydrological models in the lines 75 to 83 of the revised manuscript.

Research has shown that reservoirs play a crucial role in preventing extensive damages during the wet season and in minimizing low-flow occurrences in LMR Basin (Arias et al., 2014; Räsänen et al., 2012; Dang et al., 2024). The integration of a coupled reservoir module within the hydrological model is a widely adopted approach for evaluating reservoir impacts under changing climate. Wang et al. (2017b) utilized this approach to show that reservoir operation can minimize flood intensity and lower flood occurrence rates. Yun et al. (2021a; 2021b) demonstrated that, despite a trade-off in hydroelectric benefits, reservoir management can substantially alleviate extreme drought and wet hydrological events in LMR Basin. These studies collectively indicated that reservoirs represent a practical solution for addressing the impacts of climate change.

**4. Line 78: Where is the population data obtained?**

**Response:** We appreciate your query. The population data cited is sourced from Sabo et al., 2017, and Luo et al., 2023. We added these two references to the line 98 of the revised manuscript.

References:

Sabo, J. L., Puhi, A., Holtgrieve, G. W., Elliott, V., Arias, M. E., Ngor, B. P., Räsänen, T. A., Nam, S.: Designing river flows to improve food security futures in the lower Mekong Basin. Science 358 (6368). https://doi.org/10.1126/science.aao1053, 2017.

Luo, X., Luo, X., Ji, X., Ming, W., Wang, L., Xiao, X., Xu, J., Liu, Y., Li, Y.:
    Meteorological and hydrological droughts in the Lancang-Mekong River Basin:
    spatiotemporal patterns and propagation. Atmospheric Research 293, 106913.
    https://doi.org/10.1016/j.atmosres.2023.106913, 2023.

**5.  Line 100: Are there any other GCMs? Are only these five available, or do these five have better effects?**

**Response:** We appreciate your comment. We selected the five GCMs that are widely applied and demonstrate robust performance in the LMR Basin. Their reliability has been confirmed by studies such as Li et al. 2021, Yun et al. 2021a, and Yun et al. 2021b. We added an explanation in lines 125 to 127 of the modified version, as detailed below.

Five GCMs (Global Climate Models) with wide utilization and proven performance in LMR Basin are applied in this study (Li et al. 2021; Yun et al., 2021a; Yun et al., 2021b), i.e., GFDL-ESM4, IPSL-CM6A-LR, MPI-ESM1-2-HR, MRI-ESM2-0, and UKESM1-0-LL.

References:

Li, Y., Lu, H., Yang, K., Wang, W., Tang, Q., Khem, S., Yang, F., Huang, Y.:
    Meteorological and hydrological droughts in Mekong River Basin and
    surrounding areas under climate change. J. Hydrol.: Reg. Stud. 36, 100873.
    https://doi.org/10.1016/j.ejrh.2021.100873, 2021.

Yun, X., Tang, Q., Li, J., Lu, H., Zhang, L., Chen, D: Can reservoir regulation
    mitigate future climate change induced hydrological extremes in the
    Lancang-Mekong River Basin? Sci. Total Environ. 785.
    https://doi.org/10.1016/j.scitotenv.2021.147322, 2021a.

Yun, X., Tang, Q., Sun, S., & Wang, J.: Reducing climate change induced flood at the
    cost of hydropower in the Lancang-Mekong River Basin. Geophysical Research
    Letters, 48, e2021GL094243. https://doi.org/10.1029/2021GL094243, 2021b.

**6.  Section 2.2: There need usage instructions for the data. For instance, if the precipitation and temperature of ERA5 are used to correct GCMs, then what is the potential evapotranspiration used for?**

**Response:** Thank you for your comment. The evapotranspiration data of ERA5_Land dataset are utilized to derive the evapotranspiration data in the future period. The method proposed by Van Pelt et al. (2009) is implemented for this purpose. The calculation formula of the method has been included in Eq. (3) of the revised version.

Reference:

Van Pelt, S. C., Kabat, P., ter Maat, H. W., van den Hurk, B. J. J. M., and Weerts, A.
    H.: Discharge simulations performed with a hydrological model using bias

corrected regional climate model input, Hydrol. Earth Syst. Sci., 13, 2387–2397, https://doi.org/10.5194/hess-13-2387-2009, 2009.

7. **Section 2.3: As the core method of this section, the main formulas of MBG should be listed.**

**Response:** We appreciate your suggestion. We incorporated the two most critical formulas of the MBCn (Multivariate Bias Correction via N-dimensional Probability Density Function Transform) method in the revised manuscript: random orthogonal rotation and quantile delta mapping, as shown in Eqs. (1) and (2) on lines 164 to 185 of the revised manuscript.

8. **Section 2.4: Why are Formula 3 and Formula 8 repeated? Can so many simple formulas be explained in the main text? The principle of reservoir allocation is suggested to be shown in a schematic diagram because these formulas are both numerous and simple.**

**Response:** Thanks for your comment. In the revised manuscript we removed Eq. (8) of the previous manuscript and retained Eq. (3) of the previous manuscript, i.e., Eq. (9) in the revised manuscript. Additionally, we incorporated a flowchart illustrating reservoir scheduling, designated as Fig. 2, as shown below.

[Figure]

Figure 2: Flowchart of the constructed reservoir module.

9. **Line 145: For the complex physical mechanisms in the model, there are no formulas at all? What are the equilibrium equations, geometrical relationships, and constitutive relationships in the model? The Nash efficiency coefficient is relatively less necessary to present.**

**Response:** We appreciate your suggestion. In the revised manuscript, we enhanced the description of the THREW model and incorporated the fundamental mass balance,

momentum balance, and heat balance equations underlying the THREW model, which are presented as Eqs. (4) to (6) in the revised version. Please refer to Section 2.4 of the revised version. Additionally, to ensure the completeness of the formula presentation, we preserved the description of the NSE formula.

**10. Section 2.4: The GCM model is spatially distributed grid data, and the reservoir here is a lumped water distribution. How can a simple lumped water distribution be regulated regionally?**

**Response:** Thanks for your comment. The THREW model performs spatial basin delineation based on Representative Elementary Watershed (REW). Within the LMR Basin, the THREW model delineates 651 REWs units and conducts runoff simulations based on these REW. The reservoir module also employs REW format for reservoir operation. For GCM data, to meet the needs of hydrological simulation in the THREW model, we downscale the GCM data from grid scale to REW scale and utilize the downscale GCM data as meteorological input for the model. We added a description of GCM downscaling in lines 345 to 347 of Section 2.6, in the revised version. It reads as follows.

The meteorological data from five selected GCMs under three SSPs are downscaled from grid scale to REW scale and served as meteorological inputs for the THREW model.

**11. Line 215: I thought that the five GCMs used for simulation could mutually test the reliability, but here the average value was directly used without analyzing the sensitivity of the five models. GCMs' errors are not complementary. Some may be more accurate, while others have larger errors. A simple average value is of no help to the research.**

**Response:** We appreciate your comment. Our rationale for averaging the outputs of five GCMs is to reduce the uncertainty stemming from individual GCM simulations. We apologize that our explanation of this approach was incomplete in the initial manuscript. In the revised manuscript, we provided more detailed explanation of this aspect in Section 2.6. Furthermore, we concur with your point that relying solely on the average value is inadequate. To address this, we incorporated an analysis of the range of GCM outputs, emphasizing the variability among different GCM in the revised version. For further details, please refer to lines 368 to 377 in the revised manuscript, which reads as follows.

As each GCM possesses unique structure and assumptions, projections of climate change by a single GCM inherently possess uncertainties, which in turn introduce uncertainties in the simulation of hydrological outcomes (Kingston et al., 2011; Thompson et al., 2014). Thus, averaging across multiple GCMs is a crucial approach, as it minimizes model biases, eliminates outliers, reduces uncertainties, and ensures more robust and universally applicable outcomes (Lauri et al., 2012; Hoang et al., 2016; Hecht et al., 2019; Wang et al., 2024; Yun et al., 2021b). This method has been

extensively employed in prior studies (Dong et al., 2022; Li et al., 2021; Wang et al., 2022; Yun et al., 2021a). Therefore, this research determines the average DFAA probability from five GCMs to lessen the uncertainty in their predictions and assesses the fluctuation in these probabilities across the models to demonstrate their variability.

Moreover, we revised Figs. 5 and 6, as well as the descriptions in Sections 3.3 and 3.4, to better depict the variability in GCM outputs. The updated figures are listed below and corresponding text are included in the revised manuscript for your reference.

[Figure]

Figure 5: DFAA under natural scenario. Here, JH, NK, PA, and KT respectively denote JingHong, Nong Khai, Pakse, and Kratie stations. (a) Seasonal probability of DFAA averaged across five GCMs during history (1980-2014), near future (2021-2060) and far future (2061-2100) periods, as well as under three SSPs. The annual probability is half of the sum of wet and dry season probabilities. (b) The annual change in DFAA probability averaged across five GCMs and their ranges in the near and far future periods with respect to history period under three SSPs. (c) The seasonal change in DFAA probability averaged across five GCMs and their ranges in the near and far future periods with respect to history period during wet and dry seasons under three SSPs.

[Figure]

Figure 6: Reservoir impacts on DFAA during near future (2021-2060) and far future (2061-2100) under three SSPs. Here, JH, NK, PA, and KT denote JingHong, Nong Khai, Pakse, and Kratie stations,

respectively. (a) The annual reservoir impacts averaged across five GCMs and their ranges. (b) The seasonal reservoir impacts in wet and dry seasons averaged across five GCMs and their ranges.

12. **Line 214: According to the abstract, "Reservoir operations reduce DFAA's intensity." It should be getting the intensity of the DFAA, why there is a probability, and how to quantify intensity.**

**Response:** Thanks for your comment. In this work, we assess the risk of DFAA events by calculating their probability, but do not consider their intensity. We sincerely apologize for the inaccuracy terminology used in the abstract section. We have corrected it in the revised manuscript and would like to share our sincere appreciation for your correction.

13. **Please pay attention to the garbled characters that appear in lines 156, 242, and 243.**

**Response:** Thank you for your reminder. We have fixed this issue in the revised manuscript.

14. **When many formulas are piled up and there are no corresponding textual descriptions, it is very difficult to know what the logic between them is. Here, it is necessary to select the most important ones from these formulas for listing and then describe the logic of the formulas. Furthermore, what's these methods' regional applicability? What are their advantages and limitations?**

**Response:** Thank you for your suggestion. We adjusted the displayed equations in the revised manuscript, added logical relationships between formulas, and focused on their explanation and illustration.

15. **Section 3.1: Since the study originally used ERA5 for correction, it doesn't mean that being closer to ERA5 is accurate. ERA5 also has errors. It can only be said that after correction, the GCM is closer to ERA5, and this cannot be used as an accurate basis here. Even this subsection can be transformed into a description of the spatiotemporal distribution of climate data.**

**Response:** We appreciate your suggestion. In the revised manuscript, we modified the presentation in lines 380 to 385 of Section 3.1, and emphasized that the accuracy of CMIP6 data is grounded in a comparison with the ERA5_Land data. The revisions in section 3.1 are listed below.

From both regional and seasonal perspectives, the uncorrected raw CMIP6 data exhibits significant discrepancies with ERA5_Land data during history period (1980-2014). When compared with ERA5_Land data for history period, the uncorrected raw CMIP6 data reveals an average annual precipitation bias of 1800 mm

and an average daily temperature of 12 (Figs. 3b and 3e). These notable inconsistencies underscore that hydrological modeling using uncorrected raw CMIP6 data would incur considerable inaccuracies.

Moreover, we would like to note that correcting future meteorological data by reanalysis datasets or remote sensing datasets is a common practice. In the existing studies, for example, Hoang et al. (2016) utilized WATCH forcing data and APHRODITE datasets to correct the precipitation and temperature of GCM data. Ly et al. (2023) applied Global Precipitation Climatology Centre (GPCC) to correct the precipitation in the GCM data. Wang et al. (2021) employed precipitation data from the Climate Prediction Center (CPC) to correct the GCM data. Yun et al. (2021a) and (2021b) used the Global Meteorological Forcing (GMFD) dataset to correct ISMIP3b data. Therefore, we think it is reasonable to apply the ERA5 data to correct the bias of each GCM in the CMIP6 data.

References:

Hoang, L. P., Lauri, H., Kummu, M., Koponen, J., van Vliet, M. T. H., Supit, I., Leemans, R., Kabat, P., and Ludwig, F.: Mekong River flow and hydrological extremes under climate change, Hydrol. Earth Syst. Sci., 20, 3027–3041, https://doi.org/10.5194/hess-20-3027-2016, 2016.

Ly, S., Sayama, T. & Try, S.: Integrated impact assessment of climate change and hydropower operation on streamflow and inundation in the lower Mekong Basin. Prog Earth Planet Sci 10, 55. https://doi.org/10.1186/s40645-023-00586-8, 2023.

Wang, S., Zhang, L., She, D., Wang, G., Zhang, Q.: Future projections of flooding characteristics in the Lancang-Mekong River Basin under climate change. J. Hydrol. 602. https://doi.org/10.1016/j.jhydrol.2021.126778, 2021.

Yun, X., Tang, Q., Li, J., Lu, H., Zhang, L., Chen, D: Can reservoir regulation mitigate future climate change induced hydrological extremes in the Lancang-Mekong River Basin? Sci. Total Environ. 785. https://doi.org/10.1016/j.scitotenv.2021.147322, 2021a.

Yun, X., Tang, Q., Sun, S., & Wang, J.: Reducing climate change induced flood at the cost of hydropower in the Lancang-Mekong River Basin. Geophysical Research Letters, 48, e2021GL094243. https://doi.org/10.1029/2021GL094243, 2021b.

16. **Section 3.2: The absence of reservoirs before 2009 and the existence of reservoirs after 2010 should be very important background. When the coupled reservoir module is used for DFAA simulation, it should be simulated in segments. For those after 2010, additional reservoirs should be added. What will the situation of reservoirs be like in future scenarios? This needs to be explained in the summary and subsequent sections of the DFAA results.**

**Response:** Thanks for your comment. We would like to point out that the statement regarding "the absence of reservoirs before 2009 and the existence of reservoirs after

2010" is a general concept. However, in reality, reservoir construction and operation commenced prior to 2009 (Zhang et al., 2023). The earliest reservoirs in the reservoir data we utilize were constructed in 1965 and the latest in 2035. We considered the annual change of the reservoir storage in the reservoir module. For each new reservoir, its operation is initially conducted based on the operation rules during the initial phase. Once its storage reaches the minimum constraint, the new reservoir enters the normal phase and follows the rules of normal phase. The reservoir operation in future periods will also follow these patterns. We added the description in Section 2.4 of the revised manuscript that each reservoir starts scheduling based on their operational year in the reservoir module, as detailed below.

This module contains detailed data on 122 reservoirs in the basin, with operational years ranging from 1965 to 2035. ...... The reservoir operation rules are consistent over time and space, with each reservoir following the same operation rules and starting scheduling according to its respective operational year.

Reference:

Zhang, K., Morovati, K., Tian, F., Yu, L., Liu, B., Olivares, M.A.: Regional contributions of climate change and human activities to altered flow of the Lancang-Mekong river. J. Hydrol.: Reg. Stud. 50, 101535. https://doi.org/10.1016/j.ejrh.2023.101535, 2023.

17. **Section 3.3: Shouldn't this probability be compared with the occurrence of a single disaster before? The probability can be calculated based on the time within a year.**

**Response:** Thanks for your comment. Currently, there is a paucity of detailed statistics on individual flood and drought events in the LMR Basin, which presents challenges in determining the probability of DFAA events by the single disaster.

Moreover, we adopt $|R - SDFAI| > 1$ as the criterion for the occurrence of DFAA events, which means that DFAA events we identify are at least rapid transitions between mild hydrological drought events (standard runoff index (SRI) $<- 1$) and a mild hydrological wet events (standard runoff index (SRI) $> 1$) (Song et al., 2023). We enhanced this part in lines 340 to 342 of the revised manuscript, as detailed below.

The threshold for R-SDFAI to recognize DFAA events is $\pm 1$, which indicates that the identified DFAA event is at least an abrupt transition between a mild hydrological drought event (SRI $<- 1$) and a mild hydrological wet event (SRI $> 1$) (Song et al., 2023).

Reference:

Song, X., Lei, X., Ma, R., Hou, J., Liu, W.: Spatiotemporal variation and multivariate controls of short-cycle drought–flood abrupt alteration: A case in the Qinling-Daba Mountains of China. International Journal of Climatology, 43(10), 4756–4769, https://doi.org/10.1002/joc.8115, 2023.

**18. Section 3.4: How will the future reservoir operation information be obtained?**

**Response:** We appreciate your comment. The reservoir dataset we collected includes future planned reservoirs, the latest of which will operate from 2035. It is noted that the reservoir storage is scheduled to increase significantly in the future period and the capacity of tributary reservoirs during this period is sizable, especially in downstream reservoirs, as shown in Fig. 1c.

**19. The discussion section should use more literature to support the causes and reliability of the results. Here, for example, in the first part of the discussion, except for the first sentence, which is cited, the rest is all about explaining the results.**

**Response:** Thank you for your suggestion. We provided additional discussions on different characteristics of DTF and FTD, and incorporated existing relevant studies in lines of 534 to 539 of the revised manuscript. We hope this addition could further substantiate our arguments, and it reads as follows.

The distinct characteristics of DTF and FTD events have been identified by previous research. Shi et al. (2021) found that FTD events are predominant in the Wei River Basin. Wang et al. (2023) projected that in the Poyang Lake Basin, the temporal spread of DTF events will expand in future, while that of FTD events will constrict. Ren et al. (2023) found that under SSP1-2.6 and SSP2-4.5 scenarios, the Huang-Huai-Hai River Basin will experience more DTF events, but under SSP3-7.0 and SSP5-8.5 scenarios, it will experience more FTD events.

**20. The second part of the discussion talked about the reservoir's ability to respond to DFAA. Here, in addition to considering the changes in the water volume of the reservoir, it is also necessary to consider how long the reservoir operation occurred before or after the disaster. The occurrence time of reservoir operation will have a timely impact on the specific disaster.**

**Response:** Thanks for your comment. In Section 4.2, our primary focus is on the role of hydrological forecasting in enhancing the reservoir's ability to manage DFAA events. The differing control reservoirs exhibit over DTF and FTD risks is primarily due to the reservoir's limited ability to fully regulate when encountering unforeseen inflows, as described in Section 4.1. Hence, we suggest combining hydrological forecasting to strengthen the reservoir's mitigation functions in Section 4.2.

Within the reservoir module, the occurrence time of reservoir operation remains unchanged. After its construction, the reservoir is subject to daily scheduling based on the incoming river flow. Nonetheless, when hydrological forecasts are factored in, the scheduling approach of the reservoir is altered according to the forecast information. We added a more detailed discussion on this aspect in the revised version at lines 566 to 574, as detailed below.

Hydrological forecasts provide insights into runoff and disaster situations, enabling the adaptation of reservoirs' current and future operational procedures. This adjustment can maximize reservoirs' water management efficiency, effectively counteracting flood-induced drought (FTD) and drought-induced flood (DTF). For instance, when a flood is occurring and hydrological forecasts predict an impending drought, reservoirs' operational methods should be modified to both reserve adequate storage capacity for the next flood event and maximize water retention to counteract the subsequent drought. Likewise, if hydrological forecasts indicate that a flood will strike after the current drought, reservoir management will transition from maximizing water storage to ensuring water availability during the drought while also setting aside adequate storage capacity for the upcoming flood event.

21. **The discussion in the third part also rarely cites literature, and the utilization of the resilient storage should not be the focus of the discussion in this article. The focus is on the influence of the reservoir in the process of disaster simulation.**

**Response:** We appreciate your comment. We have supplemented the literature references in Section 4.3, which reads as follows.

...... This finding emphasizes a strong connection between reservoir storage and its mitigation potential on DFAA. It aligns with Ehsani et al. (2017), who suggested that expanding dam dimensions can offset the vulnerability of water resources to climate uncertainties, and Feng et al. (2024), whose study highlighted the effectiveness of large reservoirs in mitigating drought and flood risks. ...... The existing research has pointed out that the mitigating effect of reservoirs on extreme hydrological events is independent of their main purpose. Even when their main purpose isn't directly tied to mitigating such events, they can still offer significant benefits (Brunner, 2021a; Ho et al., 2025). ......

Additionally, given the identified alignment between reservoir mitigation effects and storage capacity distribution, what we would like to emphasize in Section 4.3 is that irrigation reservoirs, alongside hydroelectric reservoirs, play a crucial role in diminishing the likelihood of DFAA events and mitigating flood and drought pressures in the LMR Basin. Notably, the total storage of these irrigation reservoirs is considerable. This insight is vital for policymakers and stakeholders, as it contributes to the development of the coordinated system for hydroelectric and irrigation reservoirs across the entire basin and promotes joint flood and drought mitigation initiatives.

22. **In addition, the bar charts and line charts from Figure 4 to Figure 6 are all numbers that can be presented in a table, and the richness of the accompanying figures should be increased.**

**Response:** Thanks for your comment. We updated these two figures (Figs. 5 and 6 in the revised version) in the revised manuscript. These revised figures present the

average values of five GCMs alongside the ranges of their calculation results. Moreover, we provided additional details about the variability among GCMs in Sections 3.3 and 3.4 of the revised manuscript. The updated figures are as follows.

[Figure]

**Figure 5: DFAA under natural scenario. Here, JH, NK, PA, and KT respectively denote JingHong, Nong Khai, Pakse, and Kratie stations. (a) Seasonal probability of DFAA averaged across five GCMs during history (1980-2014), near future (2021-2060) and far future (2061-2100) periods, as well as under three SSPs. The annual probability is half of the sum of wet and dry season probabilities. (b) The annual change in DFAA probability averaged across five GCMs and their ranges in the near and far future periods with respect to history period under three SSPs. (c) The seasonal change in DFAA probability averaged across five GCMs and their ranges in the near and far future periods with respect to history period during wet and dry seasons under three SSPs.**

[Figure]

**Figure 6: Reservoir impacts on DFAA during near future (2021-2060) and far future (2061-2100) under three SSPs. Here, JH, NK, PA, and KT denote JingHong, Nong Khai, Pakse, and Kratie stations, respectively. (a) The annual reservoir impacts averaged across five GCMs and their ranges. (b) The seasonal reservoir impacts in wet and dry seasons averaged across five GCMs and their ranges.**

---

## Author Response (AR3)

**Editor:**

**There seems to be a technical issue with the file upload — the equations are not rendering correctly and are unreadable. Please check and re-upload a file where all content is presented properly.**

**Response:** We sincerely appreciate you pointing out that technical issue with the file upload. We have identified and resolved the formatting inconsistencies that caused the equation to render incorrectly. We have uploaded the revised files, which display all content correctly. We apologise for any inconvenience caused and appreciate your patience and support.

**We have received the second round of reviewers' comments on your revised manuscript. While some improvements have been made, major concerns regarding the clarity, methodological justification, and support of key conclusions still require your careful attention.**

**Given the potential scientific significance of your work, I'd like to offer one final opportunity for revision. Please note that acceptance/rejection of your manuscript will depend entirely on your ability to fully address all remaining concerns in this round.**

**Response:** We sincerely appreciate your constructive comments and the precious chance to improve our paper. We revised the paper extensively in response to your feedback and that of the second-round reviewers, making targeted adjustments to the issues you highlighted.

Clarity: The manuscript has undergone substantial rephrasing and restructuring to ensure precise and logical expression. We believe the revised manuscript could foster clearer communication of our findings.

Methodological justification: We supplemented the intensity classification of DFAA events and analyzed the probabilities of events with different intensities in future periods. This enables an exploration of the reservoir's mitigating capacity from both intensity and frequency (reflected through probability) perspectives.

Support of key conclusions: We restructured the discussion section and removed content that was not closely related. We emphasized the distinct characteristics of FTD and DTF, explored the relationship between reservoir mitigation role and capacity, and introduced additional data to reinforce persuasiveness. Moreover, we added the limitation section, highlighting the paper's shortcomings in reservoir regulation.

In addition, with regard to the annual DFAA probability being half the sum of the dry season and wet season DFAA probabilities, we adjusted the inappropriate expressions and illustrations in the revised version. We are deeply grateful for your patient guidance and support. We hope this modification will meet your expectations.

**Reviewer #1:**

**Thanks for authors' efforts on replying to the comments and making revisions. My comments are as follows.**

**Response:** We sincerely appreciate your invaluable time and effort in reviewing our paper and providing us with your insightful comments and suggestions. Your valuable insights are of great significance to our paper's improvement, and we are grateful for your thoughtful feedback. We have carefully considered each of your comments and hereby report the following responses.

1.  **The primary concern of this paper is that it remains unclear how climate change impacts the LMR basin. For instance, statements such as "FTD is more challenging though DTF is more probable to occur" and "Reservoir operations reduce DFAA's risks" in the abstract are confusing and lack clarity. Moreover, the study relies solely on one metric—probability—to evaluate the impacts of climate change and reservoir operations. Yet, as noted in the references, several studies have already investigated climate change impacts on the Lancang-Mekong River basin using additional metrics such as duration, intensity, and severity. It would strengthen the paper if the authors could incorporate comparisons with these studies in the discussion section.**

**Response:** Thank you for your detailed review of our research!

(1) Your insightful comments on the expression have alerted us to potential shortcomings. We recognize that the introduction may lack clarity in logic or strength in conclusions. Therefore, in the revised version, we substituted the statements 'Reservoir operations reduce DFAA's risks' and 'FTD is more challenging though DTF is more probable to occur' in the introduction with the revised expressions found in lines 15 to 22 of the revised manuscript, as shown below. We believe that the revised expressions are more precise and effectively convey our research results.

The findings reveal that DFAA in the LMR Basin is primarily dominated by DTF (drought to flood), with probabilities of DTF exceeding those of FTD (flood to drought) at mild, moderate, and severe intensity levels. The increase in DTF probability for future periods is also significantly higher than that of FTD. Mild DTF and mild FTD account for 58% to 90% and 75% to 100% of their total probability in the future, making the mild-intensity events the most frequent DFAA. Reservoirs play a significant role in reducing DTF risks during both dry and wet seasons, though their effectiveness in controlling FTD risks, particularly during the dry season, is relatively weaker.

(2) You pointed out that our study only evaluates the impacts of climate change and reservoir operation using probability as a single metric, whereas existing studies also consider frequency metrics. We acknowledge that this was a limitation we had not sufficiently addressed. Therefore, in the revised manuscript, we introduced the intensities of DFAA events, analyzing their trends at mild, moderate, and severe levels

under climate change, as well as the reservoir regulation's role in mitigation. The specific details are as follows.

In the revised manuscript, we added Table 2 and Figures 6, 8, and 10 to illustrate the intensity characteristics of DFAA under climate change and reservoir operation. The intensity classification of DFAA events is outlined in Table 2 and on lines 332 to 344. The analysis of various intensity events under climate change is presented in Figure 6 and lines 476 to 499. Figure 8 and lines 533 to 554 explain the mitigating effects of reservoirs on various intensity events. Figure 10 and lines 615 to 652 discuss the relation between reservoirs' regulation capacity and their storage. To enhance comprehension, Figures 6, 8, and 10 are listed below.

[Figure]

**Figure 6: Annual probability of DFAA at different intensities under the natural scenario, averaged across five GCMs and their ranges in the near future (2021-2060) and far future (2061-2100) periods under three SSPs. Here, JH, NK, PA, and KT respectively denote JingHong, Nong Khai, Pakse, and Kratie stations.**

[Figure]

**Figure 8: Reservoir impacts on DFAA under different intensities, averaged across five GCMs and their ranges in the near future (2021-2060) and far future (2061-2100) periods under three SSPs. Here, JH, NK, PA, and KT respectively denote JingHong, Nong Khai, Pakse, and Kratie stations.**

[Figure]

**Figure 10: The relationship between reservoirs' mitigation effects and their total storage. Symbol points denote the average values for each station under three SSP scenarios during the near future (2021-2060) and far future (2061-2100) periods, while error bars indicate the maximum and minimum values. Here, JH, NK, PA, and KT respectively denote JingHong, Nong Khai, Pakse, and Kratie stations. (a) The impact of reservoirs on the total probability of DFAA. (b) The impact of reservoirs on DFAA of different intensities. Please note that, as Jinghong and Nong Khai stations are not expected to experience severe FTD in the future, the relevant information has not been included in the figure.**

(3) We appreciate your comment to supplement our discussion with existing studies, and we have carefully considered it. However, we would like to clarify that, as noted in lines 78 to 80 of the revised manuscript, to our knowledge, there are very few existing studies on DFAA events in the LMR Basin. Consequently, we reviewed existing literature on DFAA in other basins, as referenced in lines 42 to 49 of the introduction section in the revised manuscript. Given the significant differences in hydrological characteristics and geographical locations between study regions, it is challenging to conduct comparative analyses of specific conclusions. Moreover, to our knowledge, existing research has scarcely addressed reservoir regulation's impact on DFAA events, both in the LMR Basin and in other regions. Therefore, we were unable to compare our findings on reservoir mitigation effects with existing studies.

Nevertheless, we have striven to integrate discussions with previous research within our manuscript. In section 4.1 of the discussion, we underscored the similarities between our research and prior studies. Consistent with previous relevant research, our research found that FTD and DTF events display distinct characteristics, as explained in lines 578 to 585 of the revised manuscript, listed below.

The distinct characteristics of DTF and FTD events have been identified by previous research. Shi et al. (2021) found that FTD events predominate in the Wei River Basin. Wang et al. (2023) projected that in the Poyang Lake Basin, the temporal spread of DTF events will expand in the future, while that of FTD events will constrict. Ren et al. (2023) found that under SSP1-2.6 and SSP2-4.5 scenarios, the Huang-Huai-Hai River Basin will experience more DTF events, whereas under SSP3-7.0 and SSP5-8.5 scenarios, it will experience more FTD events. This study identifies differences between DTF and FTD events as well, and further highlights the different characteristics of reservoirs' mitigating effects on these events.

2. **Regarding reservoir operations, only a single operation rule (stated in line 254) is applied across multiple years, including both near- and far-future scenarios. Using one fixed rule to assess reservoir performance over such diverse temporal scales is not persuasive. A more nuanced approach that accounts for changing conditions is needed.**

**Response:** We appreciate your comment regarding reservoir operation rules.

We would like to note that applying consistent reservoir operation rules across multiple years or study periods is a widely adopted methodology (Lauri et al., 2012; Ly et al., 2023; Piman et al., 2013; Wang et al., 2017; Yun et al., 2021a; Yun et al., 2021b).

For instance, Piman et al. (2013) implemented the HEC-ResSim reservoir operation model to assess the influence of reservoir construction on the 3S basin under diverse future scenarios. Lauri et al. (2012) and Ly et al. (2023) utilized specific nonlinear reservoir operation methods to investigate the impacts of reservoirs and climate change on the flow regimes of the LMR Basin over the next 20-30 years, and the effects of climate change and hydroelectric operations on the downstream flow of the Mekong River and the flooding of Tonle Sap Lake in the near, medium, and long term respectively. Wang (2017) incorporated SOP as the reservoir operation rule, applying it across multiple future study periods (near, medium, and long term) to investigate the role of reservoir construction in reducing future flood risks in the LMR Basin. Yun (2021a) and Yun (2021b) divided future periods into near and far future, both utilizing SOP as the reservoir operation rule. Their studies respectively analyzed the effects of reservoir operations on extreme drought and wet events, as well as the trade-offs between hydroelectric generation and flood protection benefits under climate change scenarios.

The examples provided consider the impact of reservoirs on future hydrological conditions and employed fixed reservoir operation methods. Thus, we believe it is reasonable to maintain the same regulation method for both near and far periods.

We understand your comment that different years require different reservoir operations, as operations should adapt to varying inflows. We would like to highlight that our operations do account for inflow variations and adjust accordingly. We acknowledge that this manuscript was not sufficiently emphasized in the previous

draft. Therefore, we strengthened this explanation in the revised version. The details can be found in lines 248 to 249 of the revised manuscript, as shown below.

Strategies are adapted in response to inflow fluctuations and administered on a daily scale. Each reservoir is assigned based on location.

References:

Lauri, H., de Moel, H., Ward, P., Räsänen, T., Keskinen, M., and Kummu, M.: Future changes in Mekong River hydrology: impact of climate change and reservoir operation on discharge, Hydrol. Earth Syst. Sci., 16, 4603–4619, https://doi.org/10.5194/hess-16-4603-2012, 2012.

Ly, S., Sayama, T., Try, S.: Integrated impact assessment of climate change and hydropower operation on streamflow and inundation in the lower Mekong Basin. Prog Earth Planet Sci 10, 55, https://doi.org/10.1186/s40645-023-00586-8, 2023.

Piman, T., Cochrane, T., Arias, M., Green, A., Dat, N.: Assessment of flow changes from hydropower development and operations in Sekong, Sesan, and Srepok rivers of the Mekong basin. J. Water Resour. Plan. Manag. 139, 723–732. https://doi.org/10.1061/(ASCE)WR.1943-5452.0000286, 2013.

Wang, W., Lu, H., Leung, L. R., Li, H.-Y., Zhao, J., Tian, F., Yang, K., Sothea, K.: Dam construction in Lancang-Mekong River Basin could mitigate future flood risk from warming-induced intensified rainfall. Geophysical Research Letters, 44, 10,378–10,386. https://doi.org/10.1002/2017GL075037, 2017.

Yun, X., Tang, Q., Li, J., Lu, H., Zhang, L., Chen, D.: Can reservoir regulation mitigate future climate change induced hydrological extremes in the Lancang-Mekong River Basin? Sci. Total Environ. 785, https://doi.org/10.1016/j.scitotenv.2021.147322, 2021a.

Yun, X., Tang, Q., Sun, S., Wang, J.: Reducing climate change induced flood at the cost of hydropower in the Lancang-Mekong River Basin. Geophysical Research Letters, 48, e2021GL094243, https://doi.org/10.1029/2021GL094243, 2021b.

3. **The discussion section makes it difficult to clearly compare the study's findings with previous work, and several conclusions are presented without sufficient support from the results. For example, what explains the statement "DTF and FTD exhibit quite different characteristics, in that DTF is more frequent but FTD is more challenging" (line 532)? Similarly, where in the results section can readers find evidence for "Hydrological forecasting technology enhances the potential of reservoirs" (line 560)? Likewise, what results support the claim that "The mitigation effect of reservoirs on DFAA risk is closely associated with storage distribution of mainstream and tributary reservoirs (line 579)"? These points need to be more explicitly linked to the presented findings.**

**Response:** We express our gratitude for your thorough review and insightful comment.

In the revised version, we implemented various revisions to address your concerns, such as refining the discussion section with enhanced data support, providing additional explanations, revising the expression for greater clarity, and removing less relevant content, thereby enhancing the paper's overall quality and coherence.

(1) The statement "DTF and FTD exhibit quite different characteristics, in that DTF is more frequent but FTD is more challenging" was derived from the results section and further elaborated in the three paragraphs following it. We recognize that the expression in the previous version was not sufficiently clear. Therefore, we refined the explanation and structure of Section 4.1 in the revised manuscript to enhance clarity. The revised statement can be found in lines 586 to 613 of the revised manuscript, and the principal changes are outlined as follows.

The average probability of DTF across all periods is 2.1% under the natural scenario, which is significantly higher than the 1.4% average for FTD (Fig. 5a). The probability of DTF consistently exceeds that of FTD under three different intensities (Fig. 6). ...... Compared with FTD events, reservoirs more effectively control DTF probabilities, significantly lowering DTF risk in both dry and wet seasons (Fig. 7). ...... Although FTD is less likely than DTF, reservoirs control FTD less effectively, especially in the dry season (Fig. 7). ...... This difficulty in regulation is what makes FTD a major challenge.......

(2) We appreciate your comment that the section on 'Hydrological forecasting technology enhances the potential of reservoirs' is weakly connected to the paper's main argument. We agree with it. This part was initially included as a future prospect in the discussion section. Upon reflection, we believe the revised manuscript is sufficiently comprehensive, and removing the part pertaining to hydrological forecasting would not compromise the paper's overall structure and completeness. Thus, we decided to remove the part, i.e., Section 4.2 of the previous version. This deletion has enhanced the paper's clarity and focus. We are deeply grateful for your valuable feedback, which has been crucial in refining our research.

(3) The statement regarding 'The mitigation effect of reservoirs on DFAA risk is closely associated with storage distribution of mainstream and tributary reservoirs' is based on Figures 1c and 6 from the previous manuscript. We acknowledge that the previous manuscript lacked a direct comparison between these two aspects, leading to clarity issues. To address this, we incorporated Figure 10 into the revised version, which effectively demonstrates the connection between reservoir storage and its mitigation impacts on DFAA risks across different intensities. Moreover, we enhanced the section with additional explanations and discussions to make it more readable and logically structured. The revised statement is presented in lines 615 to 652 and Figure 10 of the revised manuscript, with the main points listed as follows.

The reservoir systems provide enhanced mitigation efficiency against DFAA at JingHong and Kratie compared to those at Nong Khai and Pakse (Fig. 7). Reservoir storage in the region above JingHong and the Pakse to Kratie region is significantly larger than storage in the JingHong to Nong Khai and Nong Khai to Pakse regions (Fig. 1c). Reservoirs' capacity to reduce total DFAA risk closely relates to the total

storage of mainstream and tributary reservoirs, consistently showing a positive correlation for DTF and FTD events (Fig. 10a). ...... The positive correlation between total reservoir storage and the reduction of total DFAA risk indicates that basins with larger total storage are better equipped to resist DFAA events. ...... Both mild DTF and mild FTD show a positive correlation with total reservoir storage, consistent with total DFAA events (Fig. 10b). In contrast, moderate and severe DFAA events do not strongly correlate with reservoir storage (Fig. 10b). ......

4. **The overall quality of English expression in the paper should be improved to enhance clarity and readability.**

**Response:** We appreciate your attention to the language expression in the paper.

We recognize potential for improvement in this aspect and have revised it for greater clarity and professionalism. We optimized sentence structures, polished the language, and ensured more precise, consistent English throughout. These revisions will better convey the research results and improve the manuscript's overall readability. Please kindly review the revised manuscript.

**Reviewer #2:**

**This paper investigates the Drought-Flood Abrupt Alternation (DFAA) phenomenon in the Lancang-Mekong Basin under climate change and reservoir operation. The results conclude that climate change increases the risk of such DFAA, while reservoir operation plays a mitigating role, which aligns with the current broad consensus. Overall, the topic is relevant and addresses regional concerns, and the paper is well-structured. However, I have several concerns that the authors should explain and incorporate into their revision.**

**Response:** Your detailed review of our research is greatly appreciated. We appreciate your constructive comments on our manuscript, which have greatly helped us to improve the quality of our work. We believe that the revised version of our paper will be more rigorous and clear, effectively conveying our research findings. Below, we address each of your comments in detail.

1. **The reservoir operation rules are based on the SOP, without distinguishing the capacities of reservoirs, e.g., annual regulation for large reservoirs & daily regulation for smaller ones. Moreover, since most mainstream reservoirs in the upper Mekong are primarily for hydropower, constraints such as installed capacity and firm power should be considered. Additionally, it is worth exploring whether adjusting reservoir operation rules could improve their functionality in mitigating transitions from floods to droughts and vice versa.**

**Response:** We sincerely appreciate your insightful comment on our research!

(1) Your comment regarding the lack of differentiation in reservoir regulation scales is valuable. However, as the regulatory time scales and specific operation rules for individual reservoirs within the basin are largely inaccessible, our analysis was constrained to examining only one regulation scale: daily regulation under the SOP rule. We recognize that this limitation exists and, as such, added a section on limitations in the revised manuscript. The revised content is included in lines 664 to 668 of the revised draft, with the key modifications detailed below.

Despite this, the study uses uniform operation rules for reservoirs of different storage scales within the LMR Basin. It implements daily regulation for all reservoirs. The study does not use differentiated regulation scales (daily, annual, or multi-annual) based on storage. It also does not consider unique operation rules in different sub-basins. These simplifications may cause uncertainties in how reservoirs mitigate effects. This is a limitation of the study.

(2) We agree with your perspective that upstream reservoirs should consider constraints such as installed capacity and firm power. However, since we focused on the reservoir's regulation capacity rather than its power generation function, we did not include power-generation-related constraints such as installed capacity and firm power, as you mentioned. Nevertheless, we recognize the significance of your suggestion and will carefully consider it in future research related to reservoir power

generation. Thank you once again for your valuable insight and comment.

(3) We appreciate your suggestion to explore whether adjusting reservoir operation rules can enhance their capacity in mitigating DFAA events. We appreciate your insightful perspective and thank you for your constructive feedback. However, we believe that the current scope of our study and its findings are extensive and comprehensive. We are committed to exploring this subject further in future research.

2. **The structure of the Introduction section requires reconsideration. For instance, the quantitative indicators are presented in the second paragraph, which disrupts the coherence of the background and regional issue description.**

**Response:** We appreciate your suggestion on the structure of the introduction section.

We have revised and relocated the quantitative indicator description to section 2.5, as shown in lines 297 to 303. This change improves the logical flow and structure of the introduction. Thank you for your input.

3. **The historical period is set from 1980 to 2014, yet the hydrological model simulations extend to 2020. Given that the current year is 2025, the near-future period should ideally start from 2021, and the timeline needs updating accordingly.**

**Response:** Thank you for your attention to the timeline of this research.

The future meteorological data used here is based on the CMIP6 data set, which includes historical data (meteorological data from 1961 to 2014 for various GCMs) and future projections (meteorological data from 2015 to 2100 for various GCMs under different emission scenarios). Since the historical portion of the CMIP6 data set only extends to 2014, this study defines the history period as ending in 2014. Additionally, to maintain consistency in the length of the near future and far future periods, the near-future in this study is defined as 2021 to 2060, while the far-future is set from 2061 to 2100.

As you mentioned, the validation period for the hydrological model is set from 2010 to 2020, with an extension beyond 2014. This is because major reservoirs in the LMR Basin were predominantly built around 2010 (Zhang et al., 2023; Morovati et al., 2024). The Nuozhadu reservoir, distinguished by its substantial storage of 21.7 billion m3, started operations in 2014 (MERFI, 2024). The validation period is intended to assess the model's performance following reservoir construction. Restricting the validation period to 2014 would result in an insufficient time frame and fail to adequately reflect the regulatory capacities of mega dams, thus not fulfilling the original purpose of the validation period. Therefore, the validation period is extended to 2020 in this study.

References:

MERFI: Dataset on the Dams of the Greater Mekong. Bangkok, Mekong Region Futures Institute, https://www.merfi.org/mekong-region-dams-database (last access: March 2025), 2024.

Morovati, K., Zhang, K., Shi, L., Pokhrel, Y., Wu, M., Someth, P., Ly, S., and Tian, F.: On the cause of large daily river flow fluctuations in the Mekong River, Hydrol. Earth Syst. Sci., 28, 5133–5147, https://doi.org/10.5194/hess-28-5133-2024, 2024.

Zhang, K., Morovati, K., Tian, F., Yu, L., Liu, B., Olivares, M.A.: Regional contributions of climate change and human activities to altered flow of the Lancang-Mekong river. J. Hydrol.: Reg. Stud. 50, 101535, https://doi.org/10.1016/j.ejrh.2023.101535, 2023.

**4. The Results section currently focuses on describing phenomena but lacks interpretation of the underlying causes of these changes. This part needs strengthening, particularly for notable phenomena that may attract attention. For example:**

**(1) Lines 440-441: Why do high-emission scenarios often lead to lower risks of DFAA?**

**Response:** We appreciate your suggestion and inquiry.

The decrease in DFAA risk under the high-emission scenario is mainly because droughts and floods tend to occur together rather than in succession. As shown by Dong et al. (2022), the SPI index indicates decreased fluctuations in the SSP5-8.5 scenario compared to SSP1-2.6 and SSP2-4.5 scenarios, as illustrated in the figure below. This suggests a lower probability of rapid transitions between drought and flood events under the SSP5-8.5 scenario.

In the revised manuscript, we included an explanation of this phenomenon, as outlined in lines 446 to 448, which reads as follows.

Such a pattern is attributable to the enhanced tendency for flood and drought events in the LMR Basin to cluster rather than alternate under the SSP5-8.5 scenario (Dong et al., 2022).

References:

Dong, Z., Liu, H., Baiyinbaoligao, Hu, H., Khan, M., Wen, J., Chen, L., Tian, F.: Future projection of seasonal drought characteristics using CMIP6 in the Lancang-Mekong River Basin. J. Hydrol. 610, https://doi.org/10.1016/j.jhydrol.2022.127815, 2022.

[Figure]

Figure. Dynamics of future annual (a), dry season (b), wet season (c) SPI of the LMR Basin (Dong et al., 2022).

**(2)    Figure 7: Why do both drought-to-flood and flood-to-drought risks appear to intensify in May? Could this be due to the temporal-spatial trade-off effects of reservoir operation?**

**Response:** Thank you for your question.

As depicted in Figure 7 (Figure 9 in the revised manuscript), under the SSP2-4.5 scenario, reservoir management increased the probability of DFAA in May in the near future. This is mainly because the reservoir successfully mitigates DFAA probabilities during the preceding dry season (December to April), thereby reducing the available reservoir storage for DFAA risk regulation in May. Consequently, this diminishes the reservoir's ability to control DFAA risks in May and, in some scenarios, amplifies the risk.

We provided additional clarifications in lines 568 to 571 of the revised manuscript, as detailed below.

Sometimes, such as the SSP2-4.5 scenario in the near future, reservoirs actually increase the probability of DFAA in May. This happens because helping during the dry season before May reduces the capacity of reservoirs for water regulation in May, making it hard to control DFAA risks that month.

**5.   It is recommended to enhance the interpretation of reservoir effects from the perspective of operational processes and mechanisms.**

**Response:** We sincerely appreciate your valuable comment.

In the prior version of the manuscript, we had provided an explanation of the reservoir's role from the perspectives of operational processes and mechanisms in

Section 4.1. However, we acknowledge that our previous explanation may not have been entirely clear or thorough. To address this, we strengthened our discussion of this section in the revised manuscript. We explained the reasons for the reservoir's differing control capacities over DTF and FTD events and further explored the potential mechanisms underlying the varying mitigation effects of the reservoir on DFAA events of different intensities, as outlined in lines 591 to 613. The revised text is provided below.

Compared with FTD events, reservoirs more effectively control DTF probabilities, significantly lowering DTF risk in both dry and wet seasons (Fig. 7). The reason is that the timing of DTF's water regulation matches the way reservoirs operate. At the start of DTF, reservoirs typically hold water at the storage corresponding to the normal water level, which equates to 0.8 times the maximum storage (Eq. (20)). Hence, reservoirs possess sufficient storage capacity to mitigate the drought conditions. In parallel, the water release during the initial phase of the DTF reduced the water level, thereby meeting the storage needs for sudden floods that occur later in the DTF. As a result, even if DTF events are frequent, reservoirs can manage them well. Reservoirs especially succeed in reducing mild DTF events (Fig. 8). However, they control moderate DTF events less effectively. In intense DTF cases, the rules for operating reservoirs are not enough. For example, if a severe drought at DTF's beginning exceeds reservoir storage, they cannot effectively relieve the extreme drought and thus fail to control such DTF events.

Although FTD is less likely than DTF, reservoirs control FTD less effectively, especially in the dry season (Fig. 7). The problem is that when the FTD event occurs, reservoirs are generally maintained at their target storage for the wet season. The storage corresponds to the flood control water level, which is 1.2 times the minimum storage capacity (Eq. (19)). Consequently, reservoirs, while fully meeting flood control requirements at the start of FTD, struggle to maintain sufficient water storage to satisfy water supply demands for the subsequent drought stage. If FTD happens often, the reservoir's control decreases further. While reservoirs do little for mild FTD, they noticeably reduce moderate FTD (Fig. 8). This means that, for rare but strong FTD events, reservoirs can help by storing water for later droughts. However, if FTD is frequent, current reservoir operations do not help much. This difficulty in regulation is what makes FTD a major challenge. It is encouraging, though, that FTD is expected to become less common in most areas of the LMR Basin in the future (Fig. 5).

These revisions aim to provide a more thorough explanation of the reservoir's operational processes and mechanisms, thereby enhancing the scientific credibility and persuasiveness of the paper. Thank you once again for your insightful review and guidance.

6. **There are issues with notation, such as the use of "j" to represent both iterations and sub-regions. Additionally, please ensure that variables are italicized appropriately.**

**Response:** Thank you for your valuable comment on the notation and variables.

We have thoroughly examined and implemented the revisions. In the revised version, we introduced '$l$' to represent the number of iterations, as shown in Equations 1 and 2 and lines 159 to 176, while retaining '$j$' to denote sub-basins. Additionally, we have checked the formatting of variables, applying italics to each, as shown in Equations 1 to 40 and the relevant explanations in the revised manuscript. We believe these modifications could improve the paper's clarity and professionalism. Thank you once again for your feedback.

---

## Author Response (AR4)

**Editor:**

**The MS is still subject to revisions and further reviewed by editor and referees.**

**Response:** We are grateful to the editor and the reviewers for their constructive and detailed feedback, which has been invaluable in improving our work. We have carefully revised the manuscript accordingly, primarily focusing on enhancing the manuscript's clarity and providing a more thorough elaboration on the validity of our methodology.

In terms of manuscript presentation, we have made targeted adjustments to the display of figures, improving the clarity of those that were previously unclear. In the revised manuscript, we have optimized the color configuration logic for figures, simplified originally complex multi-panel figures by splitting them, and adopted a combined presentation of figures and tables to showcase research outcomes more clearly. We have also provided the original data on the simulated DFAA possibilities under natural and reservoir scenarios in the Supplementary File for readers' reference and comparison. Additionally, we aligned all paragraphs to full justification and unified the indentation of all equation numbers.

In terms of methodological validity, we have supplemented the hydrological parameters involved in calibrating the THREW model, as well as their value ranges, as presented in Table 2 of the revised manuscript. Moreover, we have elaborated on the effectiveness of the SOP operation rule in reflecting the impact of reservoirs on extreme drought and wet hydrological events We have also provided a rationale in the Supplementary File for adopting the Gamma distribution to assess DFAA events.

To enhance the readability of our response, we use black text for the feedback from editors and reviewers, blue text for our response to the comments, and red text for citations from the revised manuscript.

We trust that the revisions made to the revised manuscript will address the concerns raised by the editors and reviewers regarding the previous draft. We hold the conviction that their expert recommendations will substantially improve the quality of the paper. We would like to take this opportunity to express our sincere gratitude to the editors and reviewers again.

**Reviewer #1:**

**This manuscript focuses on the Lancang–Mekong River Basin, an important transboundary watershed, and investigates the scientific issue of drought–flood abrupt alternation under climate change, with particular emphasis on the regulatory role of the reservoir operation. The topic is interesting, but the presentation of the content still requires improvement. My comments are as follows.**

**Response:** We greatly appreciate your constructive comments, which are invaluable in helping us to further refine the manuscript. Please find our detailed response to your comments below.

1. **Some figures in the manuscript are densely packed with very small text, which makes them difficult to read. Moreover, many numerical values mentioned in the text are difficult to locate within the figures. In addition, when the text contains large amounts of numerical values, it becomes challenging for readers to grasp the main points the authors are trying to convey. Overall, I recommend presenting key results in tables, improving figure layouts, and, if needed, splitting complex figures to clearly highlight the main findings and enhance overall readability.**

**Response:** Thank you very much for your comment on figure display.

In the revised manuscript, we have fully incorporated this feedback. Specifically, we enlarged the text in the figures, added guidelines and harmonized the color scheme for the different seasons and periods. To enhance readability and facilitate information extraction, we also distinguished the legend colors for climate change and human activity impact. Additionally, the originally complex Figure 5 from the previous manuscript has been split into Table 4 and Figure 5 in the revised manuscript, providing a more organized and accessible layout. The revised Table 4 and Figure 5 are displayed as follows.

**Table 4: The year-round DFAA probability averaged across five GCMs during each period under the natural scenario.**

| Natural | Station | History | Near Future | | | Far Future | | |
|---------|---------|---------|----------|----------|----------|----------|----------|----------|
| | | | SSP1-2.6 | SSP2-4.5 | SSP5-8.5 | SSP1-2.6 | SSP2-4.5 | SSP5-8.5 |
| DTF | JingHong | 1.67% | 2.04% | 1.71% | 1.63% | 1.67% | 1.75% | 1.21% |
| | Nong Khai | 1.52% | 1.71% | 2.08% | 1.17% | 1.96% | 2.25% | 1.71% |
| | Pakse | 2.24% | 2.38% | 3.13% | 1.83% | 2.67% | 2.75% | 2.04% |
| | Kratie | 2.33% | 3.17% | 2.83% | 2.08% | 3.04% | 2.92% | 2.54% |
| FTD | JingHong | 0.72% | 0.83% | 1.17% | 0.63% | 0.79% | 1.25% | 0.54% |
| | Nong Khai | 1.10% | 1.25% | 1.42% | 0.71% | 1.13% | 1.12% | 0.67% |
| | Pakse | 2.10% | 1.33% | 2.04% | 1.54% | 1.58% | 1.71% | 1.17% |
| | Kratie | 1.86% | 1.71% | 1.92% | 1.33% | 2.04% | 1.87% | 1.75% |

[Figure]

**Figure 5: DFAA under the natural scenario. (a) The annual change in DFAA probability averaged across five GCMs and their ranges in the near and far future periods with respect to the history period under three SSPs. (b) The seasonal change in DFAA probability averaged across five GCMs and their ranges in the near and far future periods with respect to the history period during wet and dry seasons under three SSPs. Here, JH, NK, PA, and KT respectively denote JingHong, Nong Khai, Pakse, and Kratie stations. NF and FF represent the near future period and the far future period. 1-2.6, 2-4.5 and 5-8.5 respectively denote SSP1-2.6, SSP2-4.5, and SSP 5-8.5 scenarios. Please note that this figure illustrates variations in DFAA events under climate change. The annual and seasonal probabilities of DFAA under the natural scenario are presented in Table 4 and Table S1, respectively.**

To visually highlight the differences between DTF and FTD events and the evolving patterns of the two DFAA event types over various periods, we have maintained the use of figures in the revised manuscript. Concurrently, we have listed the probabilities of DFAA events under both natural and dammed scenarios for each period in Tables S1 to S4 of the Supplementary File, thereby clarifying key numerical values and facilitating readers' access and citation. Tables S1 to S4 are presented as follows:

**Table S1: The seasonal probability of DFAA under the natural scenario, averaged across five GCMs, during the history period (1980-2014), the near future (2021-2060), and the far future (2061-2100), as well as under three SSPs.**

| Natural | Station | History | Near Future | | | Far Future | | |
|---|---|---|---|---|---|---|---|---|
| | | | SSP1-2.6 | SSP2-4.5 | SSP5-8.5 | SSP1-2.6 | SSP2-4.5 | SSP5-8.5 |
| Wet season | | | | | | | | |
| | JingHong | 2.10% | 2.50% | 1.92% | 1.83% | 1.92% | 2.17% | 1.17% |
| DTF | Nong Khai | 2.00% | 2.25% | 2.83% | 1.75% | 3.00% | 3.00% | 2.33% |
| | Pakse | 3.81% | 3.42% | 4.58% | 2.58% | 4.50% | 3.75% | 3.00% |
| | Kratie | 3.71% | 4.83% | 4.50% | 3.08% | 5.25% | 4.25% | 4.08% |

| | Station | History | Near Future | | | Far Future | | |
|---|---|---|---|---|---|---|---|---|
| | | | SSP1-2.6 | SSP2-4.5 | SSP5-8.5 | SSP1-2.6 | SSP2-4.5 | SSP5-8.5 |
| FTD | JingHong | 0.95% | 1.08% | 1.50% | 0.67% | 1.33% | 2.17% | 0.83% |
| | Nong Khai | 1.62% | 1.92% | 1.92% | 1.25% | 1.92% | 2.08% | 1.25% |
| | Pakse | 3.52% | 2.25% | 3.17% | 3.00% | 2.92% | 3.08% | 2.25% |
| | Kratie | 3.14% | 3.25% | 3.17% | 2.50% | 3.67% | 3.33% | 3.42% |
| Dry season | | | | | | | | |
| DTF | JingHong | 1.24% | 1.58% | 1.50% | 1.42% | 1.42% | 1.33% | 1.25% |
| | Nong Khai | 1.05% | 1.17% | 1.33% | 0.58% | 0.92% | 1.50% | 1.08% |
| | Pakse | 0.67% | 1.33% | 1.67% | 1.08% | 0.83% | 1.75% | 1.08% |
| | Kratie | 0.96% | 1.50% | 1.17% | 1.08% | 0.83% | 1.58% | 1.00% |
| FTD | JingHong | 0.48% | 0.58% | 0.83% | 0.58% | 0.25% | 0.33% | 0.25% |
| | Nong Khai | 0.57% | 0.58% | 0.92% | 0.17% | 0.33% | 0.17% | 0.08% |
| | Pakse | 0.67% | 0.42% | 0.92% | 0.08% | 0.25% | 0.33% | 0.08% |
| | Kratie | 0.57% | 0.17% | 0.67% | 0.17% | 0.42% | 0.42% | 0.08% |

**Table S2: The DFAA probability at different intensities under the natural scenario, averaged across five GCMs, during the history period (1980-2014), the near future (2021-2060), and the far future (2061-2100), as well as under three SSPs.**

| Natural | Station | History | Near Future | | | Far Future | | |
|---|---|---|---|---|---|---|---|---|
| | | | SSP1-2.6 | SSP2-4.5 | SSP5-8.5 | SSP1-2.6 | SSP2-4.5 | SSP5-8.5 |
| Mild events | | | | | | | | |
| DTF | JingHong | 1.39% | 1.63% | 1.29% | 1.38% | 1.38% | 1.46% | 1.08% |
| | Nong Khai | 1.29% | 1.29% | 1.21% | 0.71% | 1.67% | 1.75% | 1.38% |
| | Pakse | 1.71% | 1.67% | 2.29% | 1.33% | 2.13% | 2.00% | 1.46% |
| | Kratie | 1.39% | 2.21% | 1.88% | 1.46% | 2.38% | 2.04% | 1.79% |
| FTD | JingHong | 0.52% | 0.75% | 1.00% | 0.63% | 0.75% | 1.08% | 0.54% |
| | Nong Khai | 1.00% | 1.08% | 1.25% | 0.67% | 1.00% | 1.00% | 0.54% |
| | Pakse | 1.90% | 1.00% | 1.67% | 1.21% | 1.42% | 1.50% | 1.00% |
| | Kratie | 1.53% | 1.46% | 1.67% | 1.29% | 1.83% | 1.46% | 1.46% |
| Moderate events | | | | | | | | |
| DTF | JingHong | 0.19% | 0.33% | 0.42% | 0.13% | 0.21% | 0.25% | 0.08% |
| | Nong Khai | 0.19% | 0.29% | 0.67% | 0.33% | 0.29% | 0.42% | 0.29% |
| | Pakse | 0.38% | 0.42% | 0.46% | 0.29% | 0.42% | 0.46% | 0.42% |
| | Kratie | 0.76% | 0.67% | 0.58% | 0.50% | 0.50% | 0.75% | 0.42% |
| FTD | JingHong | 0.05% | 0.08% | 0.17% | 0.00% | 0.04% | 0.17% | 0.00% |
| | Nong Khai | 0.14% | 0.17% | 0.17% | 0.04% | 0.13% | 0.13% | 0.13% |
| | Pakse | 0.10% | 0.33% | 0.29% | 0.33% | 0.17% | 0.21% | 0.13% |
| | Kratie | 0.33% | 0.21% | 0.21% | 0.04% | 0.21% | 0.42% | 0.29% |
| Severe events | | | | | | | | |
| DTF | JingHong | 0.08% | 0.08% | 0.00% | 0.13% | 0.08% | 0.04% | 0.04% |
| | Nong Khai | 0.33% | 0.13% | 0.21% | 0.13% | 0.00% | 0.08% | 0.04% |
| | Pakse | 0.67% | 0.29% | 0.38% | 0.21% | 0.13% | 0.29% | 0.17% |
| | Kratie | 0.67% | 0.29% | 0.38% | 0.13% | 0.17% | 0.13% | 0.33% |

| | | | | | | | | |
|---|---|---|---|---|---|---|---|---|
| | JingHong | 0.00% | 0.00% | 0.00% | 0.00% | 0.00% | 0.00% | 0.00% |
| FTD | Nong Khai | 0.00% | 0.00% | 0.00% | 0.00% | 0.00% | 0.00% | 0.00% |
| | Pakse | 0.10% | 0.00% | 0.08% | 0.00% | 0.00% | 0.00% | 0.04% |
| | Kratie | 0.10% | 0.04% | 0.04% | 0.00% | 0.00% | 0.00% | 0.00% |

**Table S3: The year-round and seasonal probability of DFAA under the dammed scenario, averaged across five GCMs, during the near future (2021-2060) and the far future (2061-2100), as well as under three SSPs.**

| Dammed | Station | Near Future | | | Far Future | | |
|---|---|---|---|---|---|---|---|
| | | SSP1-2.6 | SSP2-4.5 | SSP5-8.5 | SSP1-2.6 | SSP2-4.5 | SSP5-8.5 |
| Year-round | | | | | | | |
| DTF | JingHong | 1.17% | 1.00% | 0.63% | 1.13% | 1.33% | 0.83% |
| | Nong Khai | 1.83% | 2.04% | 0.92% | 1.83% | 1.63% | 1.58% |
| | Pakse | 2.08% | 2.83% | 1.54% | 2.29% | 2.04% | 1.83% |
| | Kratie | 2.50% | 2.33% | 1.58% | 2.50% | 2.04% | 2.25% |
| FTD | JingHong | 0.46% | 0.67% | 0.21% | 0.42% | 0.33% | 0.17% |
| | Nong Khai | 1.46% | 1.46% | 0.58% | 1.46% | 1.17% | 0.46% |
| | Pakse | 1.38% | 2.13% | 1.37% | 1.67% | 1.33% | 1.00% |
| | Kratie | 1.58% | 1.75% | 1.17% | 1.58% | 1.83% | 1.54% |
| Wet season | | | | | | | |
| DTF | JingHong | 1.92% | 1.75% | 1.00% | 1.83% | 1.75% | 0.83% |
| | Nong Khai | 2.67% | 2.75% | 1.25% | 2.58% | 2.33% | 2.08% |
| | Pakse | 3.17% | 4.33% | 2.58% | 4.08% | 3.33% | 2.83% |
| | Kratie | 4.08% | 4.17% | 2.67% | 4.75% | 3.67% | 3.92% |
| FTD | JingHong | 0.75% | 1.17% | 0.42% | 0.75% | 0.67% | 0.33% |
| | Nong Khai | 1.83% | 1.58% | 0.58% | 2.08% | 1.17% | 0.83% |
| | Pakse | 2.42% | 3.25% | 2.25% | 2.67% | 2.42% | 1.83% |
| | Kratie | 2.83% | 3.17% | 2.08% | 2.92% | 3.17% | 2.83% |
| Dry season | | | | | | | |
| DTF | JingHong | 0.42% | 0.25% | 0.25% | 0.42% | 0.92% | 0.83% |
| | Nong Khai | 1.00% | 1.33% | 0.58% | 1.08% | 0.92% | 1.08% |
| | Pakse | 1.00% | 1.33% | 0.50% | 0.50% | 0.75% | 0.83% |
| | Kratie | 0.92% | 0.50% | 0.50% | 0.25% | 0.42% | 0.58% |
| FTD | JingHong | 0.17% | 0.17% | 0.00% | 0.08% | 0.00% | 0.00% |
| | Nong Khai | 1.08% | 1.33% | 0.58% | 0.83% | 1.17% | 0.08% |
| | Pakse | 0.33% | 1.00% | 0.50% | 0.67% | 0.25% | 0.17% |
| | Kratie | 0.33% | 0.33% | 0.25% | 0.25% | 0.50% | 0.25% |

**Table S4: The DFAA probability at different intensities under the dammed scenario, averaged across five GCMs, during the near future (2021-2060) and the far future (2061-2100), as well as under three SSPs.**

| Dammed | Station | Near Future | Far Future |
|---|---|---|---|

|  |  | SSP1-2.6 | SSP2-4.5 | SSP5-8.5 | SSP1-2.6 | SSP2-4.5 | SSP5-8.5 |
|---|---|---|---|---|---|---|---|
| **Mild events** | | | | | | | |
| DTF | JingHong | 0.88% | 0.67% | 0.50% | 0.96% | 1.25% | 0.83% |
| | Nong Khai | 1.42% | 1.33% | 0.79% | 1.42% | 1.21% | 1.25% |
| | Pakse | 1.29% | 1.83% | 1.33% | 1.79% | 1.42% | 1.29% |
| | Kratie | 1.42% | 1.54% | 1.21% | 1.67% | 1.50% | 1.29% |
| FTD | JingHong | 0.46% | 0.63% | 0.21% | 0.42% | 0.33% | 0.17% |
| | Nong Khai | 1.29% | 1.46% | 0.54% | 1.38% | 1.00% | 0.38% |
| | Pakse | 1.13% | 1.79% | 1.12% | 1.50% | 1.04% | 0.83% |
| | Kratie | 1.42% | 1.54% | 1.17% | 1.37% | 1.54% | 1.42% |
| **Moderate events** | | | | | | | |
| DTF | JingHong | 0.21% | 0.33% | 0.13% | 0.08% | 0.04% | 0.00% |
| | Nong Khai | 0.29% | 0.58% | 0.04% | 0.42% | 0.38% | 0.29% |
| | Pakse | 0.54% | 0.67% | 0.08% | 0.42% | 0.42% | 0.33% |
| | Kratie | 0.71% | 0.42% | 0.25% | 0.75% | 0.54% | 0.71% |
| FTD | JingHong | 0.00% | 0.04% | 0.00% | 0.00% | 0.00% | 0.00% |
| | Nong Khai | 0.17% | 0.00% | 0.04% | 0.08% | 0.17% | 0.08% |
| | Pakse | 0.21% | 0.25% | 0.25% | 0.17% | 0.29% | 0.13% |
| | Kratie | 0.17% | 0.17% | 0.00% | 0.21% | 0.29% | 0.13% |
| **Severe events** | | | | | | | |
| DTF | JingHong | 0.08% | 0.00% | 0.00% | 0.08% | 0.04% | 0.00% |
| | Nong Khai | 0.13% | 0.12% | 0.08% | 0.00% | 0.04% | 0.04% |
| | Pakse | 0.25% | 0.33% | 0.12% | 0.08% | 0.21% | 0.21% |
| | Kratie | 0.38% | 0.38% | 0.13% | 0.08% | 0.00% | 0.25% |
| FTD | JingHong | 0.00% | 0.00% | 0.00% | 0.00% | 0.00% | 0.00% |
| | Nong Khai | 0.00% | 0.00% | 0.00% | 0.00% | 0.00% | 0.00% |
| | Pakse | 0.04% | 0.08% | 0.00% | 0.00% | 0.00% | 0.04% |
| | Kratie | 0.00% | 0.04% | 0.00% | 0.00% | 0.00% | 0.00% |

**2. Section 3.4 presents extensive numerical simulation results illustrating the impact of reservoirs on drought–flood abrupt alternation. However, the representation remains largely descriptive and does not explain the underlying mechanisms, which reduces the depth and persuasiveness of the conclusions.**

**Response:** Thank you for your thorough review of the manuscript. We sincerely appreciate your insightful comment.

While drafting the manuscript, we structured the presentation of results and findings to guide readers from fundamental observations to deeper insights. Section 3 (Results) emphasizes a descriptive account of the findings, whereas Section 4 (Discussion) concentrates on a mechanistic interpretation of those results.

In Section 3, the descriptive elaboration serves to present the research results in an accessible manner, helping readers recognize the overall trends in DFAA events under the changing climate and fostering a factual and conceptual appreciation of reservoirs' potential role in mitigating these events.

Further interpretation of the underlying mechanism is provided in Section 4. Section 4.1 thoroughly examines the characteristics of DTF and FTD events, building on the information in Sections 3.3 and 3.4. It also clarifies why reservoirs exert different levels of control over these two types of DFAA events.

To help readers better grasp the connection between Section 3.4 and Section 4.1, we have added explanatory sentences in lines 509 to 512 of Section 3.4, which are listed below:

The distinct controlling role of reservoirs on DTF risk versus FTD risk is associated with the consistency between these two types of DFAA events and the logic of reservoir operation. Section 4.1 will delve into the mechanistic details.

We believe that through the descriptive elaboration in Section 3 and the mechanistic discussion in Section 4, readers will gain a comprehensive understanding and profound insight into two essential issues: the variation in the occurrence probability of DFAA events under climate change, and the contribution of reservoir operation to mitigating the impact of climate change on these events.

3. **Supplement to Comment 1. The presentation of Figure 3 also has issues. While it is useful for the authors to compare the effects of bias correction, panels c and f appear almost blank. I recommend improving the colorbar of the figure. In addition, the manuscript states that the original data have a precipitation bias of 1800 mm and a temperature bias of 12 ℃, which are reduced to 120 mm and 0.2 ℃, respectively, after correction. However, these specific values do not seem to be directly accessible in the figure. The authors might consider adding a table to present the key numerical values or including appropriate text annotations within the figure to help readers better understand the results.**

**Response:** Thank you for your detailed revision comments, which will help us enhance the intuitive expressiveness of the manuscript and raise its quality level.

In the previous version of the manuscript, we maintained identical color bars for the pre- and post-correction plots to provide readers with a direct visual comparison of the significant correction effect, and to prevent visual confusion caused by different color bars.

We appreciate your recommendation to refine the color bars. Accordingly, we have modified the color bars in sub-figures (c) and (f) of the revised manuscript. In addition, we have added labels to all sub-figures (a) through (f) to indicate the locations of the maximum and minimum values, along with their respective numerical values. We hope

that these revisions make Figure 3 more informative. The revised Figure 3 is shown below.

[Figure]

**Figure 3: Averaged meteorological data of 5 GCMs for the history period (1980-2014). Here, 5 GCMs are corrected separately. The red and blue star symbols respectively indicate the locations of the maximum and minimum values in (a) to (f). (a) to (c) present the spatial distribution of precipitation based on respectively ERA5_Land, raw CMIP6 (raw CMIP6 minus ERA5_Land) and bias-corrected CMIP6 (bias-corrected CMIP6 minus ERA5_Land). (d) to (f) illustrate the spatial distribution of temperature based on ERA5_Land, raw CMIP6 (raw CMIP6 minus ERA5_Land) and bias-corrected CMIP6 (bias-corrected CMIP6 minus ERA5_Land). (g) shows seasonal cycles of temperature and precipitation from ERA5_Land, raw and bias-corrected CMIP6, as well as their corresponding range.**

4. **The manuscript exhibits inconsistent formatting. For example, Section 2.3 uses left-aligned text, whereas other sections are justified, and the indentation of equation numbers on the right is also inconsistent. I recommend that the authors standardize the formatting throughout the manuscript to improve readability and professionalism.**

**Response:** We sincerely appreciate your attention to the formatting issues. Your rigor and thoroughness have greatly contributed to improving the manuscript quality.

The previous manuscript did have problems with its layout. This was our oversight. We are very sorry about this.

In the revised manuscript, we have carefully reviewed the entire manuscript to ensure full justification throughout. We have also corrected the indentation of each equation number to ensure consistent formatting across the manuscript.

We believe that the formatting of the revised manuscript has improved significantly. We would like to express our gratitude once again for your valuable comments.

5. **In lines 230–237, regarding the model calibration section, I recommend that the authors provide a list of the parameters to be calibrated along with their respective value ranges, so that readers can better understand the basis and scope of the model adjustments.**

**Response:** Many thanks for raising the question on model calibration.

We have included the hydrological parameters used for THREW model calibration in Table 2 of the revised manuscript, along with explanations of the parameters and their value ranges.

This table is also listed below for your reference.

**Table 2: Calibrated hydrological parameters and their ranges.**

| Parameter | Explanation | Range |
|---|---|---|
| kv | Fraction of potential transpiration rate over potential evaporation | 0-10 |
| nt | Roughness of slope | 0-2 |
| KKA | Exponential coefficient in subsurface runoff calculations | 0-100 |
| nr | Roughness of river channel | 0-1 |
| KKD | Linear coefficient in subsurface runoff calculation | 0-1 |
| B | Shape coefficient | 0-1 |
| WM | Average water storage capacity (m) | 0-5 |
| K | Storage factor in Muskingum Method | 0-1 |
| X | Flow ratio factor in Muskingum Method | 0-0.5 |

**Reviewer #2:**

**Overall, I think the authors have made a valuable contribution and have responded constructively to most of the comments from previous reviewers. The study addresses an important and challenging question, and the manuscript is now clearer in both structure and narrative.**

**Speaking from my own experience working on climate-change impact assessments and hydrological simulations, I am very aware that this chain-type modelling framework (i.e., from GCM to hydrological model to impact indicator) requires a balance between simplicity and effectiveness. The methods need to be simple enough to keep the overall uncertainty under control, yet effective enough to capture the key mechanisms without losing physical or engineering reality. From this perspective, I have two comments for the authors' consideration.**

**Response:** We would like to express our sincere gratitude for your supportive comments on our research. Your constructive suggestions concerning the modeling framework will substantially enhance the reliability of our research methodology. The following provides our detailed responses to your specific recommendations.

1. **First, the integration of a standard operating policy (SOP) with the hydrological model is, in my view, methodologically simple. However, it remains unclear whether this setup is also effective in capturing the key mechanisms of reservoir regulation under flood and drought conditions, and thus the essential dynamics of DFAA. The current validation focuses primarily on simulated streamflow. For a study focusing on DFAA behaviors, I would strongly encourage the authors to go one step further and validate the simulated historical R-SDFAI series themselves, rather than only the streamflow. This would greatly enhance the reliability of the modelling chain.**

**Response:** We are grateful for your comments concerning the validation of the simulation approach.

Our adoption of SOP operation rules to simulate reservoirs' operation and their impacts for DFAA events was guided by a prudent review of existing research. As demonstrated in the extant literature, the SOP rules effectively capture flood and drought events under reservoir operation and perform satisfactorily in the LMR Basin, which constitutes the core justification for adopting SOP in our study. For instance, Wang et al. (2017) used an SOP-based reservoir model to investigate the effects of reservoir regulation on flood frequency curves across the United States. Yun et al. (2020) utilized the VIC model integrated with SOP rules to evaluate changes in flood scale and frequency due to reservoir operations in the LMR Basin during period from 2008 to 2016. Yun et al. (2021a) assessed future extreme dry and wet events under reservoir regulation in the same basin using the VIC-SOP framework. Yun et al. (2021b) investigated trade-offs between hydropower benefits and flood control in the LMR Basin using the SOP-integrated VIC model.

We have added an explanation regarding the effectiveness of the SOP operation rules in Section 2.4, lines 272 to 274, as detailed below:

The SOP operating policy is proven to effectively capture floods and droughts under reservoir regulation (Wang et al., 2017a; Yun et al., 2020; 2021a; 2021b).

Furthermore, the THREW hydrological model has demonstrated its capacity to reliably simulate extreme wet events in natural scenarios, i.e., in the absence of the reservoir module being coupled. This assertion is corroborated by Hou et al. (2021), who utilized the THREW model to generate representative flood hydrographs at multiple recurrence intervals for major control stations in the LMR Basin.

As outlined in lines 52 to 54 of the introduction, the LMR Basin has witnessed multiple dry-season floods and wet-season droughts in recent years, thereby underscoring the basin's potential for DFAA events. However, substantiated reports or documented instances of such specific events in the LMR Basin are exceptionally scarce. It appears that they are entirely deficient, at least to our knowledge. This gap arises due to the fact that DFAA events represent an emerging field of study, with minimal prior investigation or reporting from this standpoint. In the LMR Basin, principal organizations such as the Mekong River Commission and governmental institutions have not yet incorporated DFAA monitoring into their reporting. They typically highlight the most significant annual flood and drought events rather than individual occurrences. This scarcity significantly complicates the task of validating historical DFAA occurrences and the performance of the R-SDFAI metric.

Despite these constraints, the R-SDFAI indicator remains a reliable tool for DFAA assessment. This confidence is based on the SRI's established capacity to quantify runoff deviations and represent extreme dry/wet episodes (Shukla and Wood, 2008). The R-SDFAI further enhances this by quantifying the transition intensity between SRI-derived drought and wet states to identify and probabilistically assess DFAA events (Song et al., 2023). This theoretically grounded process enables the R-SDFAI to consistently capture transitions from flood to drought events and from drought to flood events.

Reference:

Hou, S., Tian, F., Lu, Y., Ni, G., Lu, H., Liu, H., Wei, J.: Potential role of coordinated operation of transboundary multi-reservoir system to reduce flood risk in the Lancang-Mekong River basin. Advances in Water Science, 32(1), 68-78. https://doi.org/10.14042/j.cnki.32.1309.2021.01.007, 2021. (in Chinese).

Shukla, S., and Wood, A.W.: Use of a standardized runoff index for characterizing hydrologic drought. Geophys. Res. Lett. 35 (2). https://doi.org/10.1029/2007gl032487, 2008.

Song, X., Lei, X., Ma, R., Hou, J., Liu, W.: Spatiotemporal variation and multivariate controls of short-cycle drought–flood abrupt alteration: A case in the Qinling-Daba Mountains of China. International Journal of Climatology, 43(10),

4756–4769, https://doi.org/10.1002/joc.8115, 2023.

Wang, W., Li, H. Y., Leung, L. R., Yigzaw, W., Zhao, J., Lu, H., Deng, Z., Demisie, Y., Blöschl, G.: Nonlinear filtering effects of reservoirs on flood frequency curves at the regional scale, Water Resour. Res., 53, 8277–8292, https://doi.org/10.1002/2017WR020871, 2017.

Yun, X., Tang, Q., Wang, J., Liu, X., Zhang, Y., Lu, H., Wang, Y., Zhang, L., Chen, D.: Impacts of climate change and reservoir operation on streamflow and flood characteristics in the Lancang-Mekong River Basin. J. Hydrol. 590, 125472, https://doi.org/10.1016/j.jhydrol.2020.125472, 2020.

Yun, X., Tang, Q., Li, J., Lu, H., Zhang, L., Chen, D.: Can reservoir regulation mitigate future climate change induced hydrological extremes in the Lancang-Mekong River Basin? Sci. Total Environ. 785, https://doi.org/10.1016/j.scitotenv.2021.147322, 2021a.

Yun, X., Tang, Q., Sun, S., Wang, J.: Reducing climate change induced flood at the cost of hydropower in the Lancang-Mekong River Basin. Geophysical Research Letters, 48, e2021GL094243, https://doi.org/10.1029/2021GL094243, 2021b.

**2. Second, compared with more traditional DFAA metrics, R-SDFAI can be considered simple as it only uses streamflow as inputs. However, some additional steps are needed to better demonstrate its effectiveness. For instance, the computation of R-SDFAI assumes a Gamma distribution for streamflow; this assumption should be supported with a statistical test for key stations and periods. Additionally, when calculating R-SDFAI in the historical period, it would be helpful to cross-check the identified major DFAA events against those documented in paper, newsletters or covered by media that led to notable socio-economic impacts. This would strengthen the claim that R-SDFAI has real engineering relevance and can inform future reservoir operation and planning.**

**Response:** Thank you very much for your insightful suggestion concerning the validation of the R-SDFAI indicator. This recommendation has substantially enhanced the reliability of the methodological approach presented in the manuscript.

The use of indicators based on the Gamma distribution assumption to examine extreme events in the LMR Basin is a common approach. For instance, Dong et al. (2022) applied Gamma-based SPI and SPEI to analyze meteorological droughts in the basin, while Li et al. (2021) employed Gamma-based SPI and SRI to explore the linkage between meteorological and hydrological droughts in the same region.

In addition, we acknowledge the importance of investigating the hypothesis that runoff data follow a Gamma distribution. Consequently, we evaluated the runoff distribution at key mainstream hydrological stations of the LMR Basin using simulated values from the THREW model for the calibration period (2000–2009). The findings indicate that the simulated runoff follows a Gamma distribution. This

provides a robust basis for using the Gamma distribution to calculate the SRI and R-SDFAI indicators. The verification process for the simulated runoff distribution has been appended to Appendix 1 of the Supplementary File, which reads as follows:

This study investigates the runoff distribution at principal mainstream hydrological stations in the Lancang-Mekong River (LMR) Basin using simulated outputs from the THREW (Tsinghua Representative Elementary Watershed) model over its calibration period (2000–2009). An evaluation of five common statistical distributions is conducted. The models under consideration are the Gamma, Log-Normal, Weibull, Generalized Extreme Value (GEV), and Log-logistic (see Fig. S1). The analysis demonstrates that the simulated runoff at the LMR Basin's four mainstream stations is most accurately represented by the Gamma distribution.

[Figure]

**Figure S1: Distribution of simulated runoff at four major mainstream hydrological stations during the calibration period (2000-2009).**

Furthermore, the Akaike Information Criterion (AIC) (Akaike, 1974) is employed in this study to identify the distribution that most accurately reflected simulated runoff in the calibration period. The AIC method is a widely utilized approach for conducting relative comparisons among multiple candidate distributions. The distribution that corresponds to the minimum AIC value is regarded as the optimal one. The calculation formula for AIC is provided in Eq. S1.

$$AIC = 2k + n\ln\left(\frac{SSR}{n}\right) \tag{S1}$$

Where, $k$ is the number of parameters $n$ is the number of data sequences, and *SSR* denotes the sum of squared residuals.

The AIC values for five commonly used distributions and the empirical distribution (derived from the histogram in Fig. S1) are calculated based on the simulated runoff at four major hydrological stations during the calibration period. The results are presented in Fig. S2. It can be observed that for all four major hydrological stations, the Gamma distribution provides the closest match. Therefore, under the assumption that runoff conforms to a Gamma distribution, employing the Gamma-based R-SDFAI index to evaluate Drought-Flood Abrupt Alternation (DFAA) events in the LMR Basin is a justifiable undertaking.

[Figure]

**Figure S2: AIC values of five common distributions and the empirical distribution at four mainsrteam hydrological stations.**

Furthermore, in lines 319 to 322 of the revised manuscript, we have elucidated the gamma-distributed nature of the simulated runoff, offering further support for the use of the gamma distribution in the index computation. The added text is presented below:

The runoff simulated by the THREW model for the LMR Basin conforms to a Gamma distribution, as detailed in Appendix 1 of the Supplementary File. Hence, the Gamma distribution is adopted to derive the SRI index.

Regarding your suggestion for cross-checking the identified major DFAA events against those documented in papers, newsletters, or covered by media, as noted in our response to Question 1, the various reports and publications currently available to us lack records or descriptions of historical DFAA events in the LMR Basin. As a result, it is challenging for us to verify the simulation results against such events. Nevertheless, as indicated in lines 77 to 83 of the manuscript, the absence of prior scientific and media attention to DFAA events in the LMR Basin highlights the

importance of our work. Our work adopts the novel perspective of "DFAA (drought-flood abrupt alternation)", revealing the trends of this distinctive extreme hydrological event under climate change in the LMR Basin. Furthermore, the study explores the potential of reservoir operations to mitigate the identified impacts. The findings of this study offer novel perspectives and profound insights to basin managers and relevant stakeholders.

Reference:

Dong, Z., Liu, H., Baiyinbaoligao, Hu, H., Khan, M., Wen, J., Chen, L., Tian, F.: Future projection of seasonal drought characteristics using CMIP6 in the Lancang-Mekong River Basin. J. Hydrol. 610, https://doi.org/10.1016/j.jhydrol.2022.127815, 2022.

Li, Y., Lu, H., Yang, K., Wang, W., Tang, Q., Khem, S., Yang, F., Huang, Y.: Meteorological and hydrological droughts in Mekong River Basin and surrounding areas under climate change, J. Hydrol.: Reg. Stud. 36, 100873, https://doi.org/10.1016/j.ejrh.2021.100873, 2021.

---

## Author Response (AR5)

We would like to express our gratitude to the editor and reviewers for acknowledging our work and for their diligent work on the manuscript. Their support and expertise have been instrumental in the manuscript's acceptance.

In this stage, we made some refinements to enhance the manuscript's expression. The specific modifications are listed below:

| Lines | Original Manuscript | Revised Manuscript |
|---|---|---|
| 11 | such as reservoirs | such as reservoir operations |
| 39 | DFAA, particularly DTF, is... | DFAA events, particularly DTF events, are... |
| 50 | The Lancang-Mekong River (LMR) Basin, as... | The Lancang-Mekong River (LMR), as... |
| 52 | the basin … | the LMR Basin … |
| 53 | wet season droughts account for about 40% of annual drought … | wet-season droughts account for about 40% of annual droughts … |
| 67 | Research highlights … | Researches highlight … |
| 160-162 | $X_T^{[l]}$, $X_S^{[l]}$, $X_P^{[l]}$, $R^{[l]}$, $\tilde{X}_T^{[l]}$, $\tilde{X}_S^{[l]}$, $\tilde{X}_P^{[l]}$ | Added the bold display for variables: $\boldsymbol{X}_T^{[l]}$, $\boldsymbol{X}_S^{[l]}$, $\boldsymbol{X}_P^{[l]}$, $\boldsymbol{R}^{[l]}$, $\tilde{\boldsymbol{X}}_T^{[l]}$, $\tilde{\boldsymbol{X}}_S^{[l]}$, $\tilde{\boldsymbol{X}}_P^{[l]}$ |
| 254 | Fig.2 | Fig. 2 |
| 299 | LDFAI, proposed by Wu et al. (2006) … | Long-cycle droughts-floods abrupt alternation index (LDFAI), proposed by Wu et al. (2006) … |
| 302-303 | Zhang et al. (2012) introduced the one-month interval SDFAI, … | Zhang et al. (2012) introduced the one-month interval SDFAI (short-cycle droughts-floods abrupt alternation index), … |
| 364-365; 371-372 | Period $i$ refers to the near future and far future periods. Event $e$ indicates the DTF, FTD, and DFAA events. | Period $i$ refers to near future or far future. Event $e$ indicates DTF, FTD, or DFAA. |
| 454 | Under SSP5-8.5 scenario, … | Under the SSP5-8.5 scenario, … |
| 462 | while for Nong Khai, Pakse, and Kratie, it is … | while for Nong Khai, Pakse, and Kratie, the changes are … |
| 466 | Under three climate models, … | Under three climate scenarios, … |
| 568 | Reservoir exhibits notable mitigating effects … | Reservoirs exhibit notable mitigating effects … |
| 581 | Reservoirs help regulate DFAA … | Reservoirs help to regulate DFAA … |
| 623 | …, reservoirs' control decrease further. | …, reservoirs' control decreases further. |

Once again, we extend our heartfelt thanks to the editors and reviewers for the precious time invested in reviewing the manuscript, the insightful feedback shared, and the outstanding guidance provided. We sincerely appreciate their efforts!